# An exactly solvable model for emergence and scaling laws in the multitask sparse parity problem

Yoonsoo Nam*[a], Nayara Fonseca*[a], Seok Hyeong Lee[b], Chris Mingard[a c], and Ard A. Louis[a]

[a]Rudolf Peierls Centre for Theoretical Physics, University of Oxford
[b]Center for Quantum Structures in Modules and Spaces, Seoul National University
[c]Physical and Theoretical Chemistry Laboratory, University of Oxford

## Abstract

Deep learning models can exhibit what appears to be a sudden ability to solve a new problem as training time, training data, or model size increases, a phenomenon known as emergence. In this paper, we present a framework where each new ability (a skill) is represented as a basis function. We solve a simple multi-linear model in this skill-basis, finding analytic expressions for the emergence of new skills, as well as for scaling laws of the loss with training time, data size, model size, and optimal compute. We compare our detailed calculations to direct simulations of a two-layer neural network trained on multitask sparse parity, where the tasks in the dataset are distributed according to a power-law. Our simple model captures, using a single fit parameter, the sigmoidal emergence of multiple new skills as training time, data size or model size increases in the neural network.

## 1 Introduction

*Emergence* in large language models (LLMs) has attracted a lot of recent attention [1–4]. It motivates the costly drive to train ever larger models on ever larger datasets, in the hope that new skills will emerge. While the concept of emergence has been critiqued on the grounds that the sharpness of the transition to acquiring a new skill may be sensitive to the measure being used [5], the observation that important new skills are learned for larger models raises many challenging questions: when the skills emerge and what drives the emergence. These questions are complicated by difficulties in formally defining skills or capabilities [6], and by our general limited understanding of the internal representations of deep neural networks [7].

Another widely observed property of deep learning models is that the loss improves predictably as a power-law in the number of data points or the number of model parameters or simply in the amount of compute thrown at a problem. These neural scaling laws [8, 9] have been widely observed across different architectures and datasets [10–16]. While the scaling exponents can depend on these factors, the general phenomena of scaling appear to be remarkably robust. This raises many interesting questions such as: What causes the near-universal scaling behavior? How does the continuous scaling of the loss relate to the discontinuous emergence of new skills?

A challenge in answering the questions raised by the phenomena of emergence and scaling laws arises from the enormous scale and expense of training cutting-edge modern LLMs, which are optimized for commercial applications, and not for answering scientific questions about how they work. One way that progress can be made is to study simpler dataset/architecture combinations that are more tractable. The current paper is inspired in part by recent work in this direction that proposed studying emergence in learning the sparse parity problem [17, 18], which is easy to define, but known to be

---

*These authors contributed equally; {yoonsoo.nam,nayara.fonsecadesa}@physics.ox.ac.uk.

38th Conference on Neural Information Processing Systems (NeurIPS 2024).

computationally hard. In particular, Michaud et al. [18] introduce the multiple unique sparse parity problem – where tasks are distributed in the data through a power-law distribution of frequencies – as a proxy for studying emergence and neural scaling in LLMs. For this data set, the authors empirically measure the scaling laws of a 2-layer multilayer perceptron (MLP) as a function of training steps ($T$), parameters ($N$), and training samples ($D$). Based on their quanta model of abrupt skill acquisition, they schematically derive neural scaling laws as a sum of emergences of new skills. However, no link was established between the neural network dynamics and the quanta model.

In this paper, we introduce an analytically tractable model by defining a basis of orthogonal functions for the multitask sparse parity problem. Each basis function corresponds to a skill that can be learned, and their respective frequencies are distributed following a power-law with exponent $\alpha + 1$. We then propose a simple multilinear expansion in these orthogonal functions that introduces a layered structure reminiscent of neural networks (NNs) and gives rise to the stage-like training dynamics [19]. With our simple model, we can analytically calculate full scaling laws, including pre-factors, as a function of data exponents $\alpha$, $T$, $D$, $N$, and optimal compute $C$. Our simple model can, with just one parameter calibrated to the emergence of the first skill, predict the ordered emergence of multiple skills in a 2-layer MLP. We summarize our contributions as follows:

1. *Skills as basis functions.* We establish a framework for investigating emergence by representing skills as orthogonal functions that form a basis in function space (Section 2). We apply our methods to controlled experiments on the multitask sparse parity dataset.

2. *Multilinear model.* We propose an analytically tractable model that is expanded in the basis of skill functions, and is multilinear with respect to its parameters so that it possesses a layerwise structure (Section 3). The multilinear nature of the model produces non-linear dynamics, and the orthogonal basis decouples the dynamics of each skill.

3. *Scaling laws.* We derive scaling laws for our multilinear model, including the prefactor constants, which relate the model's performance to training time ($T$), dataset size ($D$), number of parameters ($N$), and optimal compute ($C = N \times T$), see Section 4. We show that the scaling exponents for these factors are $-\alpha/(\alpha+1)$, $-\alpha/(\alpha+1)$, $-\alpha$, $-\alpha/(\alpha+2)$, respectively, where $\alpha + 1$ is the exponent of the power-law input data.

4. *Predicting emergence.* We demonstrate that our multilinear model captures the skill emergence of an MLP with 2 layers for varying training time, dataset size, and number of trainable parameters. Our results show that the multilinear model, calibrated only on the first skill, can predict the emergence of subsequent skills in the 2-layer MLP, see Fig. 1 and Section 5. We obtain an equivalent result on the time emergence for a transformer architecture (Fig. 4).

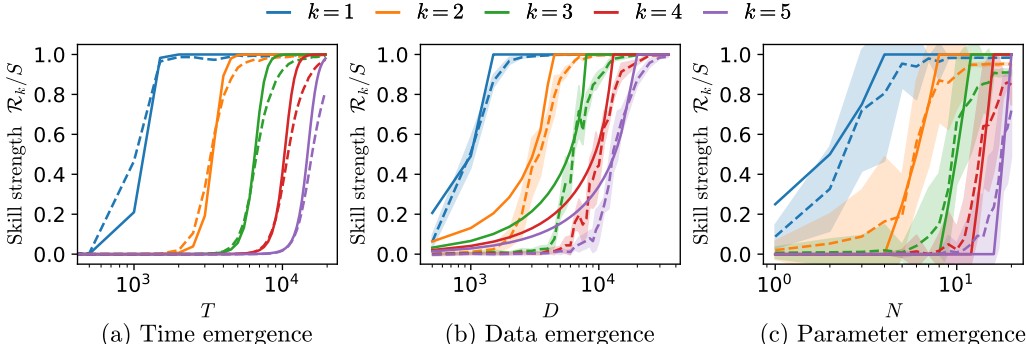

(a) Time emergence      (b) Data emergence      (c) Parameter emergence

Figure 1: **Predicting emergence.** The skill strength $\mathcal{R}_k$, defined as the $k^{th}$ coefficient if a model is expanded in the basis of the skill functions ($g_k$), measures how well the $k^{\text{th}}$ skill is learned, and is plotted against (a) time $T$, (b) data set size $D$, and (c) number of parameters $N$ (width of the hidden layer). $\mathcal{R}_k$ is normalized by the target scale $S$ such that $\mathcal{R}_k/S = 1$ means zero skill loss. The dashed lines show the abrupt growth – emergence – of 5 skills for a 2-layer MLP (Appendix K) trained on the multitask sparse parity problem with data power-law exponent $\alpha = 0.6$ (shaded area indicate 1-standard deviation over at least 10 runs). Solid lines are the predictions (Eqs. (14), (17) and (21), respectively) from our multilinear model calibrated on the first skill (blue) only.

Table 1: **Multitask sparse parity dataset and skill basis functions.** The control bits are $n_s$-dimensional one-hot vectors encoding specific parity tasks, indexed in the first column. The frequency of the distinct parity tasks follows a rank-frequency distribution with an inverse power law relation (Eq. (1)). The skill bits are binary strings with $m = 3$ relevant sparse bits (highlighted in colors) with their locations varying by skill. The $y$ column shows the target scale $S$ multiplied by the parity computed from the relevant bit set $M(i, x)$. The last columns show the values of the skill basis functions $g_k(i, x)$, defined in Eq. (2).

| Skill idx $(I)$ | Control bits | Skill bits $(X)$ | $y$ | $M(i,x)$ | $g_1(i,x)$ | $g_2(i,x)$ | ... | $g_{n_s}(i,x)$ |
|---|---|---|---|---|---|---|---|---|
| 1 | 1000000 | 110110000100 | $S$ | [1,1,0] | 1 | 0 | ... | 0 |
| 1 | 1000000 | 100101010001 | $-S$ | [0,1,0] | $-1$ | 0 | ... | 0 |
| $\vdots$ | $\vdots$ | $\vdots$ | $\vdots$ | $\vdots$ | $\vdots$ | $\vdots$ | $\vdots$ | $\vdots$ |
| 2 | 0100000 | 001001011011 | $-S$ | [0,0,1] | 0 | $-1$ | ... | 0 |
| $\vdots$ | $\vdots$ | $\vdots$ | $\vdots$ | $\vdots$ | $\vdots$ | $\vdots$ | $\vdots$ | $\vdots$ |
| $n_s$ | 0000001 | 001010100110 | $-S$ | [1,1,1] | 0 | 0 | ... | $-1$ |

## 2 Setup

In this section, we define the multitask sparse parity problem under the mean-squared error (MSE) loss. We represent skills as orthogonal functions and measure their strength in a model by calculating the linear correlation between the model output and the skill basis functions. For a comprehensive list of notations, refer to the **glossary** in Appendix A. Our code is also available online.[1]

**Multitask sparse parity problem.** In the sparse parity problem, $n_b$ skill bits are presented to the model. The target function is a parity function applied to a fixed subset of the input bits. The model must detect the relevant $m < n_b$ sparse bits and return the parity function on this subset ($M(i, x)$), see Table 1). Michaud et al. [18] introduced the **multitask** sparse parity problem by introducing $n_s$ unique sparse parity variants – or skills – with different sparse bits (for a representation, see Table 1). Each skill is represented in the $n_s$ control bits as a one-hot string, and the model must solve the specific sparse parity task indicated by the control bits (for more details, see Appendix B.1).

The $n_s$ skills (random variable $I \in \{1, 2, \ldots, n_s\}$) follow a power law distribution $\mathcal{P}_s$, and the skill bits (random variable $X \in \{0, 1\}^{n_b}$) are uniformly distributed. Because $\mathcal{P}_s$ and $\mathcal{P}_b$ are independent, the input distribution $\mathcal{P}(I, X)$ follows a product of two distributions:

$$\mathcal{P}_s(I = i) := \frac{i^{-(\alpha+1)}}{\sum_j^{n_s} j^{-(\alpha+1)}}, \qquad \mathcal{P}_b(X = x) := 2^{-n_b}, \qquad \mathcal{P}(I, X) := \mathcal{P}_s(I)\mathcal{P}_b(X). \quad (1)$$

We denote $A = \left(\sum_{j=1}^{n_s} j^{-(\alpha+1)}\right)^{-1}$ so that $\mathcal{P}_s(i) := A i^{-(\alpha+1)}$.

**Skill basis functions.** We represent the $k^{th}$ skill as a function $g_k : \{0, 1\}^{n_s+n_b} \to \{-1, 0, 1\}$ that returns the parity ($\{-1, 1\}$) on the $k^{th}$ skill's sparse bits if $i = k$, but returns 0 if the control bit mismatches that of the $k^{th}$ skill ($i \neq k$):

$$g_k(i, x) := \begin{cases} (-1)^{\sum_j M_j(i,x)} & \text{if } i = k \\ 0 & \text{otherwise} \end{cases}, \quad (2)$$

where $M : \{0, 1\}^{n_s+n_b} \to \{0, 1\}^m$ is the map that selects the relevant sparse bits for the $i^{th}$ skill (Table 1) and $M_j(i, x)$ is the $j^{\text{th}}$ entry of $M(i, x)$. Note that different skill functions have 0 correlation as the supports of skills functions are **mutually exclusive**:

$$g_k(i, x)g_{k'}(i, x) = \delta_{i,k}\delta_{k,k'}. \quad (3)$$

---

[1]https://github.com/yoonsoonam119/Skill_Eigenmode.git

**The target function.** The target function is a sum over $n_s$ skill functions multiplied by a target scale $S$:

$$f^*(i, x) := S \sum_{k=1}^{n_s} g_k(i, x). \tag{4}$$

The target scale $S$ is the norm of the target function ($\mathbf{E}_{I,X}[f^*(I, X)f^*(I, X)] = S^2$). Note that the skill functions serve as 'features' or countable basis for describing the target function as in Hutter [20].

**MSE loss.** We use MSE loss for analytic tractability:

$$\mathcal{L} := \frac{1}{2}\mathbf{E}_{X,I}\left[(f^*(I, X) - f(I, X))^2\right], \tag{5}$$

where $f$ is the function expressed by a given model. We define the skill loss $\mathcal{L}_k$ as the loss when only the $k^{th}$ skill is given, which can be weighted by their skill frequencies to express the total loss:

$$\mathcal{L}_k := \frac{1}{2}\mathbf{E}_X\left[(f^*(I = k, X) - f(I = k, X))^2\right], \qquad \mathcal{L} = \sum_{k=1}^{n_s} \mathcal{P}_s(I = k)\mathcal{L}_k. \tag{6}$$

**Skill strength.** The skill strength or the linear correlation between the $k^{th}$ skill ($g_k$) and a function expressed by the model at time $T$ ($f_T$) is

$$\mathcal{R}_k(T) := \mathbf{E}_X\left[g_k(I = k, X)f_T(I = k, X)\right]. \tag{7}$$

The skill strength $\mathcal{R}_k$ is the $k^{th}$ coefficient if a model is expanded in the basis of the skill functions ($g_k$). The skill strength, like the test loss, can be accurately approximated by a sum (see Appendix K.3). The skill loss $\mathcal{L}_k$ (Eq. (6)) can be expressed by the skill strength and the norm of the learned function for $I = k$:

$$\mathcal{L}_k(T) = \frac{1}{2}\left(S^2 + \mathbf{E}_X\left[f_T(I = k, X)^2\right] - 2S\mathcal{R}_k(f_T)\right). \tag{8}$$

The skill loss becomes 0 if and only if $f_T(I = k, X) = Sg_k(I = k, X)$.

**Experimental setting.** We use a 2-layer MLP that receives the $n_s + n_b$ bits as inputs and outputs a scalar ($\{0, 1\}^{n_s + n_b} \to \mathbb{R}$). In most of the experiments, the NN is trained with stochastic gradient descent (SGD) with width 1000, using $n_s = 5$, $m = 3$, and $n_b = 32$, unless otherwise stated. A decoder transformer is also used for the time emergent experiments. See Appendix K for details.

## 3 Multilinear model

We propose a simple multilinear model – multilinear with respect to the parameters – with the first $N$ most frequent skill functions $g_k(i, x)$ as the basis functions (features):

$$f_T(i, x; a, b) = \sum_{k=1}^{N} a_k(T)b_k(T)g_k(i, x), \tag{9}$$

where $a, b \in \mathbb{R}^N$ are the parameters. The model has built-in skill functions $g_k$ – which transform control bits and skill bits into the parity outputs of each skill – so the model only needs to scale the parameters to $a_k b_k = S$.

The multilinear structure (product of $a_k, b_k$) is analogous to the layered structure of NNs and results in emergent dynamics (Fig. 1(a)) different from a linear model with the same basis functions (Appendix H). A similar model has been studied by Saxe et al. [19] in the context of linear neural networks (Appendix B.2).

For the multilinear model, note that $a_k(T)b_k(T)$ is the skill strength $\mathcal{R}_k$ (Eq. (7)) and the skill loss (Eq. (6)) is a function of $S$ and $\mathcal{R}_k$ only:

$$a_k(T)b_k(T) = \mathcal{R}_k(T), \qquad \mathcal{L}_k(T) = \frac{1}{2}(S - \mathcal{R}_k(T))^2. \tag{10}$$

Assuming that we are training the model on $D$ samples from $\mathcal{P}(I, X)$, the empirical loss decomposes into a sum of empirical skill losses because $g_k$'s supports are mutually exclusive. This **decouples** the dynamics of each skill ($\mathcal{R}_k(T)$), which is analytically solvable under gradient flow (Appendix C.1).

$$\mathcal{L}^{(D)}(T) = \frac{1}{2D} \sum_{k=1}^{n_s} d_k (S - \mathcal{R}_k(T))^2, \qquad \frac{\mathcal{R}_k(T)}{S} = \frac{1}{1 + \left(\frac{S}{\mathcal{R}_k(0)} - 1\right) e^{-2\eta \frac{d_k}{D} ST}}, \qquad (11)$$

where $d_k$ is the number of samples of the $k^{th}$ skill (i.e., number of samples $(i, x)$ with $g_k(i, x) \neq 0$), $\eta$ is the learning rate, and $0 < \mathcal{R}_k(0) < S$ is the skill strength at initialization.

## 4 Scaling laws

Recent literature has extensively explored scaling laws; see Section 7 for an overview. In this section, we derive the scaling laws of our multilinear model (Section 3) for time ($T$), data ($D$), parameters ($N$) and optimal compute ($C$). We define compute as $C := T \times N$ [21].

Table 2 shows our analytical scaling laws including their prefactor constants (Appendix J) and Fig. 2 compares the simulation of our model with our scaling law predictions. For the scaling law exponents, we achieve the same exponent as in Hutter [20] for $D$ and in Michaud et al. [18] for $T, D$, and $N$. Assuming $0 < \alpha < 1$, the exponents are consistent with the small power-law exponents reported in large-scale experiments, see, e.g., [9, 14, 22].

Using Eqs. (6), (10) and (11), we derive the loss as a function of time ($T$), data ($D$), parameters ($N$), and the number of observations for each skill $[d_1, \cdots, d_{n_s}]$:

$$\mathcal{L} = \frac{S^2}{2} \sum_{k=1}^{N} \mathcal{P}_s(k) \frac{1}{\left(1 + \left(\frac{S}{\mathcal{R}_k(0)} - 1\right)^{-1} e^{2\eta \frac{d_k}{D} ST}\right)^2} + \frac{S^2}{2} \sum_{k=N+1}^{n_s} \mathcal{P}_s(k). \qquad (12)$$

Under suitable assumptions (e.g., for the $T$ scaling law, we take $D, N \to \infty$ and $d_k/D \to \mathcal{P}_s(k)$), we can use Eq. (12) to derive the scaling laws. For $T, D$, and $N$, we used Eq. (11) – decoupled dynamics induced the basis functions $g_k$ – to decouple the evolution of each skill loss:

1. For the time scaling law, each $\mathcal{L}_k$ shares the same dynamics with $T$ scaled by $\mathcal{P}_s(k)$.

2. For the data scaling law, each $\mathcal{L}_k$ depends only on the observation the $k^{th}$ skill ($d_k > 0$).

3. For the parameter scaling law, each $\mathcal{L}_k$ depends on whether the model has $g_k$ as a basis function.

For the optimal compute scaling law, we show in Corollary 4 (Appendix J) that the optimal tradeoff between $T$ and $N$ for given $C$ is when $T$ is large enough to fit the $N^{th}$ skill (Fig. 3). In Appendix J, we show **rigorous** derivations of all scaling laws, including the prefactors, error bounds, and conditions (e.g., how large $N$ must be compared to $T$ to be treated as infinity). For simplified derivations for the exponents only, see Appendix E. For an intuitive derivation (stage-like training) and connection to Michaud et al. [18], see Appendix D.

## 5 Predicting emergence

The literature on emergence has rapidly expanded lately; for a review of these developments, see Section 7. In this section, we analyze the emergence of a 2-layer NN (Section 2) and discuss to what degree the emergence in NNs can be described with our model. At initialization, NNs **lack** the information about the data and must 'discover' each $g_k$. To take this effect into account in our model, we add an extra parameter which we calibrate (fit) on an NN trained on one skill ($n_s = 1$) system and use it to predict the emergence of subsequent skills for the $n_s = 5$ setup (Fig. 1).

### 5.1 Time emergence

In our multilinear model, the layerwise structure – the product of parameters $a_k b_k$ – leads to a sigmoidal saturation where an update of one layer hastens the update of the other layer. Feature

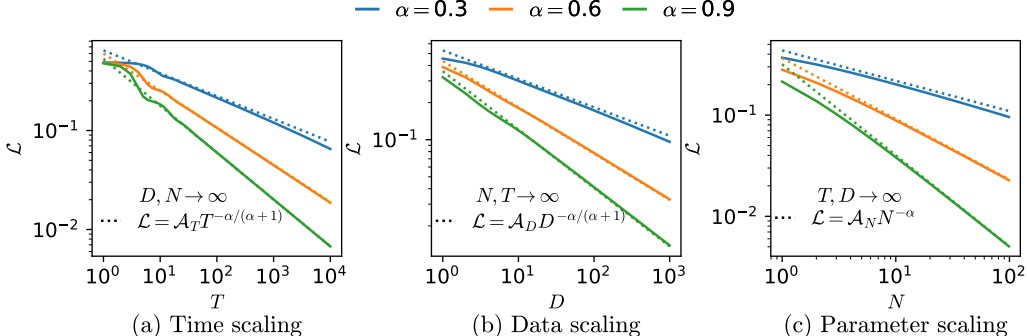

|  | $\alpha = 0.3$ | $\alpha = 0.6$ | $\alpha = 0.9$ |

(a) Time scaling      (b) Data scaling      (c) Parameter scaling

Figure 2: **Scaling laws.** The learning curve ($\mathcal{L}$ is the MSE loss) of the multilinear model (solid) and the theoretical power-law (dotted) for (a) time $T$, (b) data $D$, and (c) parameters $N$. Lower left legends show the condition (top) and the scaling law (bottom) where $\alpha + 1$ is the exponent of the power-law input data (Eq. (1)). See the appendices for 1) rigorous derivations of the theoretical scaling laws including the exponents, prefactors (e.g., $\mathcal{A}_N$ for $\mathcal{L} = \mathcal{A}_N N^{-\alpha}$), and conditions (Appendix J); 2) simplified derivations of the exponent only (Appendix E); 3) details of the experiment (Appendix K.4).

Table 2: **Summary of the scaling laws for the multilinear model.** The leftmost column indicates the bottleneck resource while the next two columns are the conditions for the 'large resources' – large enough to be treated as infinity. The fourth column is the bottleneck resource's scaling law exponent for the loss. The last two columns show the statement for the prefactor constant and the scaling law (with the assumptions and explicit error terms) in Appendix J.

| Bottleneck | Condition 1 | Condition 2 | Exponent | Prefactor | Scaling law |
|---|---|---|---|---|---|
| Time ($T$) | $D \gg NT^2, T^3$ | $N^{\alpha+1} \gg T$ | $-\alpha/(\alpha+1)$ | Thm.4 | Thms.2,3 |
| Data ($D$) | $T \gg D(\log D)^{1+\epsilon}$ | $N^{\alpha+1} \gg D$ | $-\alpha/(\alpha+1)$ | Thm.5 | Thm.5 |
| Parameter ($N$) | $D \gg T^3$ | $N^{\alpha+1} = o(T)$ | $-\alpha$ | Thm.1 | Thm.1 |
| Compute ($C$) | $D \gg T^3$ | $N^{\alpha+1} \approx T$ | $-\alpha/(\alpha+2)$ | Cor. 5 | Cor. 4 |

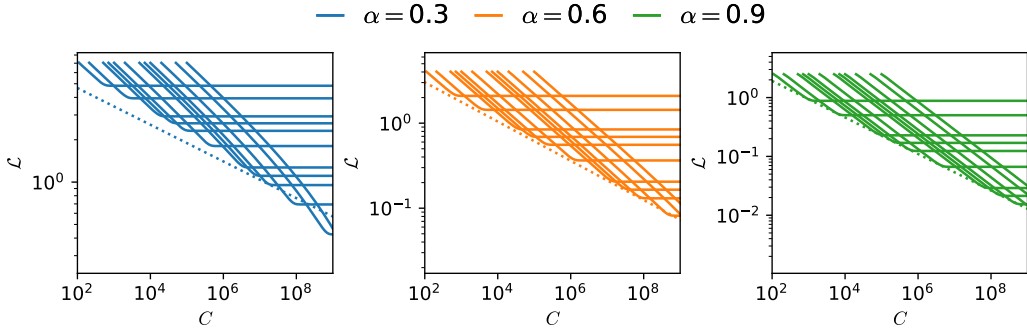

|  | $\alpha = 0.3$ | $\alpha = 0.6$ | $\alpha = 0.9$ |

Figure 3: **Scaling law for optimal compute.** The solid lines are the learning curves of the multilinear model as a function of compute $C = T \times N$ with varying parameters $N$ from $10^1$ (top plateau) to $10^4$ (bottom plateau). The dotted lines are optimal compute scaling laws with exponent $-\alpha/(\alpha+2)$ (Appendix E.4) and calculated prefactor constants (Appendix J). See Appendix K.4 for details of the experiment. For a given $C$, we achieve the optimal tradeoff when $T$ is large enough to fit all $N$ skills (i.e. when the solid lines plateau). For the case $\alpha = 0.3$, the optimal $C$ for the model decays faster than the power-law, see Appendix E.1.

learning dynamics in a 2-layer MLP shares the positive feedback between the layers but require a non-trivial update of parameters to express $g_k$.

**Extended model.** Given that feature learning, though nonlinear, involves parameter updates, we compensate for the additional delay in feature-learning by multiplying $g_k$ by a calibration constant $0 < \mathcal{B} < 1$:

$$f_T(i, x; a, b) = \sum_{k=1}^{N} a_k(T) b_k(T) \mathcal{B} g_k(i, x), \qquad 0 < \mathcal{B} < 1. \tag{13}$$

The calibration constant $\mathcal{B}$ rescales the dynamics in $T$ (Eq. (11)):

$$\frac{\mathcal{R}_k(T)}{S} = \frac{1}{1 + \left(\frac{S}{\mathcal{R}_k(0)} - 1\right) e^{-2\eta \mathcal{P}_s(k) \mathcal{B}^2 S T}}, \tag{14}$$

where $d_k/D \to \mathcal{P}_s(k)$ because we assume $D \to \infty$. We observe that $\mathcal{B}^2 = 1/22$ fits the NN trained on one skill (see Fig. 11 in Appendix I), and the calibrated model predicts emergence in the $n_s = 5$ system (Fig. 1(a)): suggesting that the dynamics of feature-learning $g_k$ in 2-layers NNs is similar to that of parameter learning ($a_k b_k$) in a simple multilinear model. For further intuition of the extended model, see an example of time emergence in an NN in Appendix G.

## 5.2 Data point emergence

Our multilinear model can learn the $k^{th}$ skill with a single observation of the skill because the skill functions $g_k$ are built in (see Corollary 1 in Appendix C.2). NNs, without the fixed basis functions, must 'discover' each $g_k$, which requires multiple samples from the $k^{th}$ skill.

**Extended model.** To make our model a $D_c$-shot learner, we extend it by replacing $g_k$ with the $e_{k,l}$ basis:

$$f_T(i, x; a, B) = \sum_{k=1}^{N} a_k(T) \sum_{l=1}^{D_c} B_{k,l}(T) e_{k,l}(i, x), \tag{15}$$

where the matrix $B \in \mathbb{R}^{N \times D_c}$ is an extension of $b \in \mathbb{R}^N$ in Eq. (9), $D_c$ is a fixed scalar, and $e_{k,l}(i, x) : \{0,1\}^{n_s + n_b} \to \mathbb{R}$ are functions with the following properties:

$$\mathbf{E}_{X|I=k}[e_{k,l} e_{k,l'}] = \delta_{ll'}, \quad e_{k,l}(I \neq k, x) = 0, \quad \sum_{l=1}^{D_c} \frac{1}{\sqrt{D_c}} e_{k,l} = g_k. \tag{16}$$

The first property states that $e_k$'s, when $I = k$, are orthonormal in $X$. The second property asserts that, similar to $g_k$ (Eq. (2)), $e_{k,l}$ is non-zero only when $I = k$, and fitting of the $k^{th}$ skill only occurs among $e_{k,l}$'s, keeping the skills *decoupled*. The third property states that $g_k$ can be expressed using $e_{k,l}$'s.

For the $k^{th}$ skill, the extended model overfits $g_k$ when there are fewer observations ($d_k$) than the dimension of the $e_{k,l}$ basis ($D_c$), and fits $g_k$ when $d_k \geq D_c$, making our model a $D_c$ shot learner.

***$D_c$ shot learner.*** *If we initialize the extended model in Eq. (15) with sufficiently small initialization and if the conditions in Eq. (16) are satisfied, then the skill strength after training ($T \to \infty$) on $D$ datapoints is*

$$\mathcal{R}_k(\infty) = \begin{cases} S\left(1 - \sqrt{1 - d_k/D_c}\right) & : d_k < D_c \\ S & : d_k \geq D_c. \end{cases} \tag{17}$$

*The number $d_k$ is the number of samples in the training set for the $k^{th}$ skill (i.e., datapoints with $g_k(i, x) \neq 0$).*

**Proof** See Appendix F.3. ∎

Using Eq. (17), we can calculate the emergence of $\mathcal{R}_k/S$ as a function of $D$. Note that Eq. (17) is similar to the model in Michaud et al. [18] in that, to learn a skill, the model requires a certain number of samples from the skill.

The derivation of Eq. (17) follows trivially from the dynamics of the extended model (Eq. (15)) and well-known results in linear/kernel regression [23–27]. To be more specific, the model finds the minimum norm solution as if we performed ridgeless regression on $g_k$ with basis functions $[e_{k,1}, \cdots e_{k,D_c}]$. See Appendix F.3 for details.

We observe that $D_c = 800$ approximates the data emergence for the $n_s = 1$ system (see Fig. 11 in Appendix I) and also the emergence for $n_s = 5$ system (Fig. 1(b)), suggesting that the NN discovers $g_k$ when it observes $D_c$ samples from the $k^{th}$ skill.

## 5.3 Parameter emergence

Since our multilinear model has $g_k$'s as the basis functions, it requires only one basis function (2 parameters) to express a skill (see Corollary 2 in Appendix C.3). A 2-layer NN cannot express a skill with a single hidden node (i.e., a hidden layer with width 1); it requires multiple hidden nodes to express a single skill.

**Extended model.** To compensate for the need for multiple hidden nodes in expressing one skill, we extend our model similarly to Eq. (15). Because the number of parameters is now a bottleneck, we ensure the model has $N$ basis functions ($e_{k,l}$'s):

$$f_T(i, x; a, B) = \sum_{k=1}^{q-1} \sum_{l=1}^{N_c} a_k(T) B_{k,l}(T) e_{k,l}(i, x) + \sum_{l'=1}^{r} a_q(T) B_{q,l'}(T) e_{q,l'}(i, x), \qquad (18)$$

where $N_c$ is the number of basis functions needed to express a skill, quotient $q$ is $\lfloor (N-1)/N_c \rfloor + 1$ and remainder $r$ is such that $(q-1)N_c + r = N$. In short, the $N$ basis functions are

$$[e_{1,1}, \cdots, e_{1,N_c}, \quad e_{2,1}, \cdots, e_{2,N_c} \quad \cdots \quad e_{q,1}, \cdots, e_{q,r}]. \qquad (19)$$

Similar to Eq. (16), the basis functions satisfy the following properties

$$\mathbf{E}_{X|I=k}[e_{k,l} e_{k,l'}] = \delta_{ll'}, \quad e_{k,l}(I \neq k, x) = 0, \quad \sum_{l=1}^{N_c} \frac{1}{\sqrt{N_c}} e_{k,l} = g_k. \qquad (20)$$

$N_c$ **basis functions for a skill.** *For the extended model in Eq. (18), the skill strength at $T, D \to \infty$ for a given $N$ becomes*

$$\mathcal{R}_k(\infty) = \begin{cases} 0 & : k > q \\ S \frac{r}{N_c} & : k = q \\ S & : k < q . \end{cases} \qquad (21)$$

**Proof** See Appendix F.4. ∎

The model can express the $k^{\text{th}}$ skill based on the number of available basis functions for the given skill (Eq. (21)). For example, skills with $k < q$ have all $N_c$ basis functions $[e_{k,1}, \cdots, e_{k,N_c}]$ to express the $k^{\text{th}}$ skill (Eq. (20)), while for $k = q$, only $r$ of the $N_c$ basis functions are available.

We observe that $N_c = 4$ fits the parameter emergence for the $n_s = 1$ system (see Fig. 11 in Appendix I) and also the emergence for the $n_s = 5$ system (Fig. 1(c)), suggesting that the NN requires 4 nodes in expressing $g_k$. The results also suggest that an NN, while lacking the ordering of basis functions (Eq. (19)), prefers to use the hidden neuron in fitting more frequent skills. The 'preference' toward frequent skills agrees with Fig. 1(a) where the NN learns more frequent skills first. Note that for the parameter emergence experiment, Adam [28] was used, instead of SGD, to increase the chance of escaping the near-flat saddle points induced by an insufficient number of parameters.

## 5.4 Time emergence in a transformer

To test whether our conceptual framework extends to other architectures, we perform a time emergence experiment with a transformer (Fig. 4). Note that the emergent time $\tau_{emerge}$ – when the skill strength is sufficiently larger than 0 – follows the same power-law relationship as Eq. (11): $\tau_{emerge}(k) \propto k^{\alpha+1}$ (see Fig. 6 in Appendix D for a discussion on emergent time). This suggests that, in the multitask sparse parity setup, other architectures may follow similar decoupled dynamics (Eq. (11)) and the consequent scaling laws (Section 4) and emergence (Section 5). An in-depth study of these findings across different architectures is left for future work.

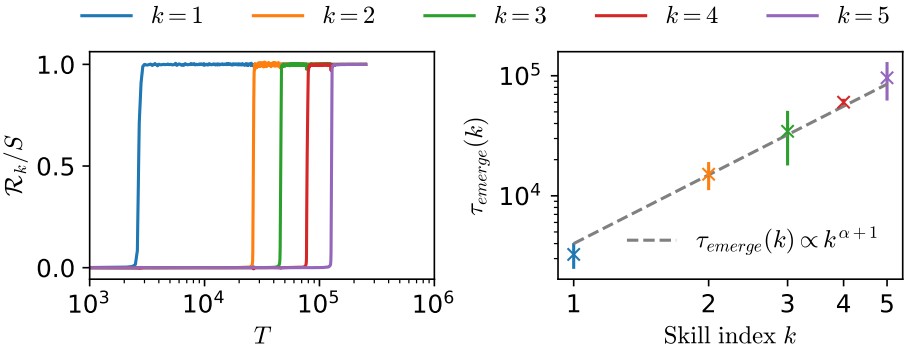

Figure 4: **Transformer on multitask sparse parity task.** We trained a transformer on the multitask sparse parity task with $\alpha = 0.9$; see Appendix K for details. **Left:** An example of the time emergence (measued in steps) for the transformer in the $n_s = 5$ setup. See Appendix I for enlarged plots showing the saturation of each skill in linear scale. **Right:** The $k^{th}$ skill's emergent time $\tau_{emerge}(k)$ (i.e. $\mathcal{R}_k(\tau_{emerge}(k))/S = 0.05$) as a function of $k$ (error bars indicate 1-standard deviation over 5 runs). The emergent times follow a power law of $k^{\alpha+1}$, following the same relationship in the multilinear model (Eq. (11)).

## 5.5 Limitations of the multilinear model

The strength of our extended multilinear model comes from the decoupled dynamics for each skill: leading to the prediction of the time, data, and parameter emergence with a single calibration. The weakness of our model is that it simplifies the more complex dynamics of NNs.

**Time emergence.** We note that the NN and the multilinear model emerge at similar instances, but the NN takes longer to saturate fully. This is because, for a given skill, the dynamics of the NN is not one sigmoidal saturation but a sum of **multiple** sigmoidal dynamics with different saturation times. To express the parity function, the NN must use multiple hidden neurons, and the skill strength can be divided into the skill strength from each neuron whose dynamics follow a sigmoidal saturation. Because of the non-linearity and the function it expresses, each neuron is updated at different rates, and the slowly saturating neurons result in a longer tail compared to our multilinear model. For an example, see Fig. 8 in Appendix G.

**Data point emergence.** Our extended model (Eq. (17)) deviates from NNs when $d_k \ll D_c$ and NNs show a more abrupt change in $\mathcal{R}_k$ as a function of $D$. This is because our model asserts strict decoupling among the skills: even a few $d_k$ will contribute to learning $g_k$ from $e_{k,l}$. This differs from the NN, which lacks strict decoupling among the samples from different skills. We speculate that because NNs can perform benign [29] or tempered [30] overfitting, they treat a few data points from less frequent skills as 'noise' from more frequent skills: requiring more samples to learn the infrequent skills.

**Parameter emergence.** Note that Fig. 1(c) has high variance compared to other emergence plots in Fig. 1; this is because the NN sparsely, over many repeated trials, uses the hidden neurons to learn less frequent skills over more frequent ones (see Table 5 in Appendix I for an example of such outliers). Because NNs are less strictly biased toward frequent skills than our model, we speculate that initial conditions favoring less frequent skills may contribute to the outliers.

## 6 Discussion and conclusion

This work demonstrated scaling laws and predicted emergence in a 2-layer MLP using a tractable multilinear model. We found that representing skills as mutually exclusive functions leads to the decoupled dynamics, resulting in the scaling laws observed in a 2-layer MLP. The layerwise structure leads to emergent (sigmoidal) saturation of the skill strength, similar to what is observed in 2-layer MLPs.

Despite lacking explicit skill functions, NNs exhibit similar emergence patterns. We speculate that the model's layerwise structure and power-law frequencies of the skills induce **stage-like** dynamics

(Appendix D) in NNs. The parameters relevant for expressing more frequent skills are updated significantly faster than those for less frequent skills. When skill 'discovery' operates on different time scales with minimal interaction, the skill dynamics effectively become **decoupled**, justifying our model setup.

Our results suggest a link between feature learning and emergence [6] driven by decoupled, stage-like dynamics. The layerwise dynamics leading to sigmoidal saturation may also disentangle the problem into skills (features) of varying importance (frequencies). Then feature learning, or discovering the basis functions that describe the target function [31, 32] (for recent studies, see [33–38]), likely occurs in stages. Investigating this connection through layerwise dynamics is left for future work.

Similar to many prior works (see, e.g., [20, 18]), we studied a simple model on an idealized power-law distributed dataset. Also, our model cannot capture the complex non-linear interactions among multiple skills but can express any linear superposition of skills. In future work, we will explore 'complex skills' in language as a superposition of linearly independent skills. By validating our findings in language tasks, we aim to contribute to a broader understanding of how neural networks acquire and exhibit complex behaviors.

## 7 Related works

In this section, we review the literature on scaling laws and emergence in NNs. Focusing on data scaling, Hutter [20] develops a model with a discrete set of features. Under the assumption of a power-law distribution of features, this model demonstrates that the error decreases as a power law with increasing data size. In a related vein, Michaud et al. [18] propose a model of neural scaling laws in which the loss is decomposed into a sum over 'quanta'. Their model aims to reconcile the apparent discrepancy between loss metrics' regular power-law scaling and the abrupt development of novel capabilities in large-scale models. Various other models for neural scaling laws have been proposed in recent research, including connecting neural scaling exponents to the data manifold's dimension [39] and their relation with kernels [40], proposing solvable random-feature models [41, 21], and developing data scaling models using kernel methods [42, 43, 25].

Closely related to the study of neural scaling laws is the understanding of emergent abilities in large language models. Several studies [1–4] document examples of such emergent abilities. Arora and Goyal [44] propose a framework for the emergence of tuples of skills in language models, in which the task of predicting text requires combining different skills from an underlying set of language abilities. Okawa et al. [45] demonstrate that a capability composed of smoothly scaling skills will exhibit emergent scaling due to the multiplicative effect of the underlying skills' performance. Other works related to the skill acquisition include Yu et al. [46], who introduce a new evaluation to measure the ability to combine skills and develop a methodology for grading such evaluations, and Chen et al. [47], who formalize the notion of skills and their natural acquisition order in language models.

## Acknowledgements

NF acknowledges the UKRI support through the Horizon Europe guarantee Marie Skłodowska-Curie grant (EP/X036820/1). SL was supported by the National Research Foundation of Korea (NRF) grant funded by the Korean government (MSIT) (No.2020R1A5A1016126). We thank Charles London, Eric Michaud, Zohar Ringel, and Shuofeng Zhang for their helpful comments.

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

# A Glossary

| | |
|---|---|
| $A$ | Normalization constant for $\mathcal{P}_s$ such that $\mathcal{P}_s(k) = Ak^{-(\alpha+1)}$ |
| $T$ | Time or step |
| $D$ | Number of data points |
| $N$ | Number of parameters (skill basis functions in the model for the multilinear model; the width of hidden layer for MLP) |
| $C$ | The computation cost $T \times N$ |
| $n_s$ | The number of skills in the multitask sparse parity problem |
| $I$ | Random variable of the control bits |
| $X$ | Random variable of the skill bits |
| $\mathcal{P}_s$ | Probability of skills (control bits) |
| $\mathcal{P}_b$ | Probability of skill bits |
| $S$ | The target scale or the norm of the target function |
| $\mathcal{R}_k$ | Skill strength of the $k^{th}$ skill (Eq. (7)) |
| $\mathcal{L}$ | Total (generalization) loss |
| $\mathcal{L}^{(D)}$ | Empirical loss for $D$ samples |
| $\mathcal{L}_k$ | Skill loss of the $k^{th}$ skill (Eq. (6)) |
| $d_k$ | Number of observation of the $k^{th}$ skill (i.e. number of training points $(i, x)$ with $g_k(i, x) \neq 0$) |
| $f^*$ | Target function $f^* : \{0,1\}^{n_s+n_b} \to \{-S, S\}$ (Eq. (4)) |
| $g_k$ | The $k^{th}$ skill basis function $g_k : \{0,1\}^{n_s+n_b} \to \{-1, 0, 1\}$ (Eq. (2)) |

Table 3: Representation of the multitask sparse parity as presented in [18]. The control bits are one-hot vectors encoding a specific parity task. The frequency of the different tasks follows a power-law distribution. In this example, there are $n_s = 10$ tasks, and skill bits are length $n_b = 15$. The $y$ column is the resulting parity computed from $m = 3$ bits (highlighted in colors). The multitask dataset provides a controlled experimental setting designed to investigate skills.

| Control bits | Skill bits | $y$ |
|---|---|---|
| 10000000000 | 110001000001010 | 1 |
| 01000000000 | 010100100001000 | 0 |
| 00100000000 | 001101010110101 | 1 |
| $\vdots$ | $\vdots$ | $\vdots$ |
| 00000000001 | 100010001001100 | 1 |

# B    Background

In this section, we review the multitask sparse parity dataset, as described by Michaud et al. [18] and discuss the nonlinear dynamics of two-layer linear networks, following the work of Saxe et al. [19].

## B.1    Multitask sparse parity

The sparse parity task can be stated as follows: for a bit string of length $n_b$, the goal is to determine the parity (sum mod 2) of a predetermined subset of $m$ bits within that string. The **multitask** sparse parity [18] extends this problem by introducing $n_s$ unique sparse parity variants in the dataset. The input bit strings have a length of $n_s + n_b$. The first $n_s$ bits function as indicators by assigning a specific task. The frequency of the distinct parity tasks follows a rank-frequency distribution with an inverse power law relation (power-law distribution). The last $n_b$ bits are uniformly distributed. This sets a binary classification problem $\{0,1\}^{n_s+n_b} \to \{0,1\}$ where only a single bit of the initial $n_s$ bits is nonzero. In Table 3, the many distinct parity tasks represent different skills. [2]

The proposal in [18] aims to reconcile the regularity of scaling laws with the emergence of abilities with scale using three key hypotheses: (i) skills, represented as a finite set of computations, are distinct and separate; (ii) these skills differ in their effectiveness, leading to a ranking based on their utility to reduce the loss; and (iii) the pattern of how frequently these skills are used in prediction follows a power-law distribution. Interestingly, the multitask problem has a consistent pattern across scaling curves: each parity displays a distinct transition, characterized by a sharp decrease in loss at a specific scale of parameters, data, or training step. Such a sudden shift occurs after an initial phase of no noticeable improvement, leading to reverse sigmoid-shaped learning curves. Michaud et al. [18] empirically show that for a one-hidden-layer neural network with ReLU activation, trained using cross-entropy loss and the Adam optimizer, these transitions happen at different scales for distinct tasks. This results in a smooth decrease in the overall loss as the number of skill levels increases.

## B.2    Nonlinear dynamics of linear neural network

Saxe et al. [19] have solved the exact dynamics for two-layer linear neural networks with gradient descent under MSE loss (Fig. 5(a)). [3] The dynamics decompose into independent modes that show sigmoidal growth at different timescales (Fig. 5(c)). The setup assumes orthogonal input features $X \in \mathbb{R}^{d_1}$ and input-output correlation matrix $\Sigma \in \mathbb{R}^{d_1 \times d_3}$ for target output $f^*(X) \in \mathbb{R}^{d_3}$:

$$\mathbf{E}_X\left[X_i X_j\right] = \delta_{ij}, \qquad \Sigma = \mathbf{E}_X\left[X f^{*T}(X)\right] \tag{22}$$

---

[2]Note that here we follow the even/odd parity convention used in [18], i.e., $\{0,1\}$, instead of $\{1,-1\}$ as used in the main text.

[3]To be specific, it is under gradient flow or the continuous limit of full batch gradient descent.

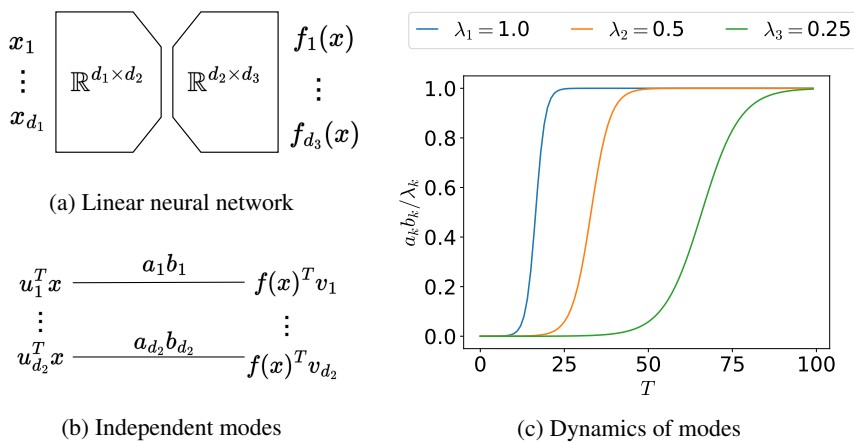

(a) Linear neural network

(b) Independent modes

(c) Dynamics of modes

Figure 5: **Nonlinear dynamics of linear neural networks. (a):** A two-layer undercomplete linear neural network, which is a multiplication of two matrices, where $d_2 < d_1$ and $d_2 < d_3$. **(b):** The $d_2$ independent modes of dynamics for linear neural network (Eq. (24)). The product of parameters $a_k b_k$ are learnable parameters and vectors $u_k, v_k$ are obtained from SVD of the input-output correlation matrix $\Sigma$ (Eq. (22)). **(c):** The temporal evolution of $a_k b_k$ under gradient descent, which follows a sigmoidal growth (Eq. (25)). Note that smaller $\lambda_k$ – the singular value of $\Sigma$ – results in a more delayed saturation of $a_k b_k$.

By performing SVD (singular value decomposition) on input-output correlation matrix $\Sigma = U \Lambda V$, the target function $f^* : \mathbb{R}^{d_1} \to \mathbb{R}^{d_3}$ becomes:

$$f^*(x) = \sum_{k=1}^{d_2} v_k \lambda_k u_k^T x, \qquad U^T \Lambda V = \mathbf{E}_X \left[ X f^*(X)^T \right] \tag{23}$$

where $u_k \in \mathbb{R}^{d_1}, v_k \in \mathbb{R}^{d_3}$ are the row vectors of $U, V$ and $\lambda_k \in \mathbb{R}$ are the singular values of $\Lambda$.

Saxe et al. [19] have shown that the dynamics of a two-layer (one-hidden-layer) undercomplete (the width of the hidden layer is smaller than the width of the input and output) linear neural network decomposes into that of the following 'modes':

$$v_k^T f(x; a, b) = a_k b_k u_k^T x \qquad k \in \{1, 2, \cdots, d_2\}. \tag{24}$$

where $a_k, b_k \in \mathbb{R}$ are the parameters. Note that Eq. (24) are $d_2$ decoupled functions $v_k^T f(x) : \mathbb{R}^{d_1} \to \mathbb{R}$ (Fig. 5(b)). Assuming small and positive initialization ($0 < a_k(0)b_k(0) \ll \lambda_k$), the dynamics of Eq. (24) under gradient descent with learning rate $\eta$ can be solved analytically; the product of parameters $a_k b_k$ grows sigmoidally with saturation time proportional to $\lambda_k^{-1}$ (Fig. 5(c)):

$$\frac{a_k(T) b_k(T)}{\lambda_k} = \frac{1}{1 + \left( \frac{\lambda_k}{a_i(0)b_i(0)} - 1 \right) e^{-2\eta \lambda_k t}}. \tag{25}$$

Using the analytic equation of the multilinear model, Saxe et al. [19] have empirically demonstrated that the dynamics of both linear and **nonlinear** neural networks closely resemble that of the multilinear model (Eq. (25)).

# C   Derivation of the multilinear model

In this section, we provide derivations of how the skill loss of our multilinear model evolves with a given resource: time (Lemma 1), data (Corollary 1), and parameters (Corollary 2). Note that two corollaries for data and parameters (Corollaries 1 and 2) follow from the decoupled dynamics (Lemma 1).

## C.1   Decoupled dynamics of the multilinear model

**Lemma 1.** *Let the multilinear model Eq. (9) be trained with gradient flow on $D$ i.i.d samples for the setup in Section 2 (input distribution: Eq. (1), target function: Eq. (4), and MSE loss: Eq. (5)). Let $k \leq N$ be a skill index in the multilinear model and the input distribution ($k \leq n_s$). Then assuming the following initialization $a_k(0) = b_k(0)$ and $0 < a_k(0)b_k(0) < S$, the dynamics of the $k^{th}$ skill strength ($\mathcal{R}_k$) is*

$$\mathcal{R}_k(T) = \frac{S}{1 + \left(\frac{S}{\mathcal{R}_k(0)} - 1\right) e^{-2\eta S \frac{d_k}{D} T}} \tag{26}$$

*and the skill loss is*

$$\mathcal{L}_k(T) = \frac{S^2}{2 \left(1 + \left(\frac{S}{\mathcal{R}_k(0)} - 1\right)^{-1} e^{2\eta S \frac{d_k}{D} T}\right)^2}, \tag{27}$$

*where $\eta$ is the learning rate and $d_k$ is the number of observations with $g_k(I = k, x^{(j_k)}) \neq 0$.*

**Proof**   For $j = 1, \cdots, D$, denote $(i^{(j)}, x^{(j)})$ be the $j^{th}$ data point in the training set. Then the empirical loss for $D$ datapoints is given as

$$\mathcal{L}^{(D)} = \frac{1}{2D} \sum_{j=1}^{D} \left(f^*(i^{(j)}, x^{(j)}) - f(i^{(j)}, x^{(j)})\right)^2. \tag{28}$$

We note that

$$\left(f^*(i^{(j)}, x^{(j)}) - f(i^{(j)}, x^{(j)})\right)^2 = \left(\sum_{k=1}^{n_s}(S - a_k b_k)g_k(i^{(j)}, x^{(j)})\right)^2$$

$$= (S - a_{i^{(j)}} b_{i^{(j)}})^2 g_{i^{(j)}}(i^{(j)}, x^{(j)})^2$$

$$= (S - a_{i^{(j)}} b_{i^{(j)}})^2,$$

as $g_i(i, j) \in \{1, -1\}$ and $g_k(i, j) = 0$ for $i \neq k$. So if we denote $d_k$ the number of data points with $i^{(j)} = k$, then we can conclude

$$\mathcal{L}^{(D)} = \frac{1}{2D} \sum_{j=1}^{D} (S - a_{i^{(j)}} b_{i^{(j)}})^2 = \frac{1}{2D} \sum_{k=1}^{n_s} d_k (S - a_k b_k)^2, \tag{29}$$

which is the decoupled loss in the main text (Eq. (11)). Using the gradient descent equation and Eq. (29), we obtain

$$\frac{da_k}{dt} = -\eta \frac{d\mathcal{L}_D}{da_k} \tag{30}$$

$$= -\eta \frac{d_k}{D} b_k (a_k b_k - S). \tag{31}$$

Likewise, we can obtain the equation for $b_k$ as

$$\frac{db_k}{dt} = -\eta \frac{d_k}{D} a_k (a_k b_k - S). \tag{32}$$

Because of symmetry between $a$ and $b$ (See Appendix B.2 or [19]), assuming $a_k(0) = b_k(0)$, and $a_k(0)b_k(0) > 0$ results in $a_k(T) = b_k(T)$ for all $T$. The equation for $\mathcal{R}_k = a_k b_k$ is

$$\frac{d\mathcal{R}_k}{dt} = -\eta \frac{da_k}{dt} b_k + a_k \frac{db_k}{dt} = -\eta \frac{d_k}{D}(b_k^2 + a_k^2)(a_k b_k - S) \tag{33}$$

$$= -2\eta \frac{d_k}{D} \mathcal{R}_k (\mathcal{R}_k - S). \tag{34}$$

Assuming $a_k(0)b_k(0) < S$, we can solve the differential equation to obtain

$$\mathcal{R}_k(T) = \frac{S}{1 + \left(\frac{S}{\mathcal{R}_k(0)} - 1\right) e^{-2\eta S \frac{d_k}{D} T}}. \tag{35}$$

The equation for $\mathcal{L}_k$ follows from Eq. (10). ■

## C.2 One-shot learner

**Corollary 1.** *For the setup in Lemma 1, the $k^{th}$ skill loss ($\mathcal{L}_k$) at $T, N \to \infty$ is*

$$\mathcal{L}_k(\infty) = \begin{cases} 0 & : d_k > 0 \\ (S - \mathcal{R}_k(0))^2/2 \approx S^2/2 & : d_k = 0, \end{cases} \tag{36}$$

*where $d_k$ is the number of $k^{th}$ skill's observations.*

**Proof** The corollary follows directly from Lemma 1. By taking $T, N \to \infty$,

$$\mathcal{R}_k(\infty) = \begin{cases} S & : d_k > 0 \\ \mathcal{R}_k(0) & : d_k = 0 \end{cases} \tag{37}$$

We obtain the result by using the relationship between $\mathcal{R}_k$ and $\mathcal{L}_k$ in Eq. (10). ■

## C.3 Equivalence between a basis function and a skill

**Corollary 2.** *Let the multilinear model Eq. (9) be trained with gradient flow on $D$ i.i.d samples for the setup in Section 3 (input distribution: Eq. (1), target function: Eq. (4), and MSE loss: Eq. (5)). Assume $a_k(0) = b_k(0)$, $0 < a_k(0)b_k(0) < S$, and that the model has the $N$ most frequent skills as basis functions. Then $\mathcal{R}_k$ for the $k^{th} \leq n_s$ skill at $T, D \to \infty$ is*

$$\mathcal{L}_k(\infty) = \begin{cases} 0 & : k \leq N \\ S^2/2 & : k > N \end{cases} \tag{38}$$

**Proof** The corollary follows directly from Lemma 1. By taking $T, D \to \infty$,

$$\mathcal{R}_k(\infty) = \begin{cases} S & : k \leq N \\ \mathcal{R}_k(0) & : k > N \end{cases} \tag{39}$$

We obtain the result by using the relationship between $\mathcal{R}_k$ and $\mathcal{L}_k$ in Eq. (10) and $\mathcal{R}_k(0) \ll S$. ■

## D  Stage-like training: intuitive derivation of the scaling laws

Even though we provide more detailed (Appendix E) and rigorous (Appendix J) derivation of the scaling laws, a less general yet more intuitive solution aids in understanding the scaling laws of our model and NNs. In this section, we define stage-like training – one skill is completely learned before the next skill initiates learning (Fig. 6(a)) – and state the conditions for it to occur. We provide an example of how stage-like training results in the time scaling law and explain how the model in Michaud et al. [18] may arise from the NN dynamics. Finally, we discuss the stage-like training's role in emergence in NNs.

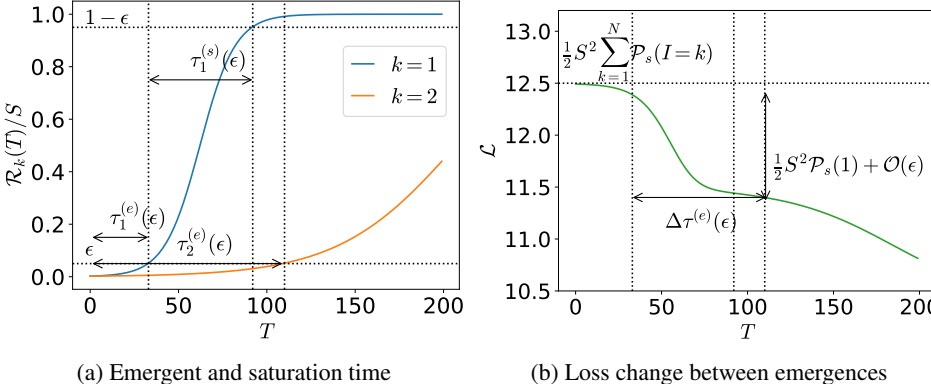

(a) Emergent and saturation time      (b) Loss change between emergences

Figure 6: **Stage-like training.** The multilinear model is trained on the multitask sparse parity problem with $\alpha = 0.6$ and $S = 5$. **(a):** Skill strength of the model as a function of time. The emergent time $\tau_k^{(e)}(\epsilon)$ is the time required for the $k^{th}$ skill to reach $\mathcal{R}_k/S = \epsilon$. The saturation time $\tau_k^{(s)}(\epsilon)$ is the time required for $\mathcal{R}_k/S$ to saturate from $\epsilon$ to $1 - \epsilon$. The model shows stage-like training if the emergent time interval $\tau_{k+1}^{(e)}(\epsilon) - \tau_k^{(e)}(\epsilon)$ is larger than the saturation time $\tau_k^{(s)}(\epsilon)$ for sufficiently small $\epsilon$ (0.05 in the figure). **(b):** The loss as a function of time for the same system as (a). For stage-like training, the change in the loss for the $k^{th}$ emergence is $\mathcal{P}_s(k)\mathcal{L}_k + \mathcal{O}(\epsilon)$ and the interval for the next emergence is $\Delta\tau^{(e)}(\epsilon) = \tau_{k+1}^{(e)}(\epsilon) - \tau_k^{(e)}(\epsilon)$.

### D.1  Stage-like training

When a model exhibits an emergence behavior – when saturation of skill occurs abruptly after a delay – and the intervals between each emergence are sufficiently large, the model admits stage-like training. The multilinear model (sigmoidal saturation of skills strength, Eq. (11)) in the multitask sparse parity dataset (power-law decay of skill frequencies, Eq. (1)) can satisfy such conditions: In Fig. 6(a), we observe the stage-like training in time in which one skill saturates (reaches $\mathcal{R}_k/S \approx 1$) before the next skill initiates its emergence. To quantify this behavior, we define two intervals for each skill (see Fig. 6(a)):

- The emergent time $\tau_k^{(e)}(\epsilon)$: the time for $\mathcal{R}_k/S$ to reach $\epsilon$;
- The saturation time $\tau_k^{(s)}(\epsilon)$: the time for $\mathcal{R}_k/S$ to saturate from $\epsilon$ to $1 - \epsilon$.

Using the dynamics equation (Eq. (11)) and that $d_k/D \to \mathcal{P}_s(k)$, the emergent time and saturation time of the $k^{th}$ skill becomes

$$\tau_k^{(e)}(\epsilon) = \frac{1}{2\eta\mathcal{P}_s(k)S}\ln\left(\frac{\frac{S}{\mathcal{R}_k(0)}-1}{\frac{1}{\epsilon}-1}\right) \propto k^{\alpha+1}, \qquad \tau_k^{(s)}(\epsilon) = \frac{1}{\eta\mathcal{P}_s(k)S}\ln\left(\frac{1}{\epsilon}-1\right) \propto k^{\alpha+1}.$$

(40)

For sufficiently small initialization ($\mathcal{R}_k(0) \ll S$), we get a **stage-like** training:

$$\tau_k^{(s)}(\epsilon) < \tau_{k+1}^{(e)}(\epsilon) - \tau_k^{(e)}(\epsilon), \qquad \epsilon \ll 1. \tag{41}$$

where the model finishes learning (saturating) the $k^{th}$ skill before starting to learn (emerging) the next skill.

## D.2 Time scaling law from stage-like training

Assuming our model satisfies the stage-like training for all $k$ of interest, we can derive the time scaling law from the stage-like training.

At $\tau_k^{(e)}(\epsilon)$, because of stage-like training, all skills with index up to but not including $k$ have saturated ($\mathcal{R}_{i<k} \approx S$), or equivalently $\mathcal{L}_{i<k} \approx 0$ (Eq. (10)). The total loss, the sum of $\mathcal{L}_j$ weighted by $\mathcal{P}_s(j) \propto j^{-(\alpha+1)}$ (Eq. (6)), becomes $\sum_{j=k}^{\infty} \mathcal{P}_s(I = j)S^2/2$ (Fig. 6(b)). The saturation of the $k^{th}$ skill results in a loss difference of $\mathcal{P}_s(I = k)S^2/2$. Thus, we obtain

$$\frac{\Delta \mathcal{L}}{\mathcal{L}} \approx \frac{\mathcal{P}_s(I = k)}{\sum_{j=k}^{\infty} \mathcal{P}_s(I = j)} = -\frac{k^{-(\alpha+1)}}{\sum_{j=k}^{\infty} j^{-(\alpha+1)}} \approx -\frac{k^{-(\alpha+1)}}{\int_k^{\infty} j^{-(\alpha+1)} dj} \tag{42}$$

$$= -\alpha k^{-1} + \mathcal{O}(k^{-2}). \tag{43}$$

Accordingly, the emergent interval between the $k$ and $k + 1$ skills relative to the $\tau_k^{(e)}(\epsilon)$ is

$$\frac{\Delta T}{T} = \frac{\tau_{k+1}^{(e)}(\epsilon) - \tau_k^{(e)}(\epsilon)}{\tau_k^{(e)}(\epsilon)} = \frac{(k+1)^{\alpha+1} - k^{\alpha+1}}{k^{\alpha+1}} \tag{44}$$

$$= (\alpha + 1)k^{-1} + \mathcal{O}(k^{-2}). \tag{45}$$

Assuming $k \gg 1$ and combining Eq. (43) and Eq. (45) to the largest order, we have the equation for the power-law with exponent $-\alpha/(\alpha + 1)$ in Fig. 2(a):

$$\frac{\Delta \mathcal{L}}{\mathcal{L}} = -\frac{\alpha}{\alpha + 1} \frac{\Delta T}{T}. \tag{46}$$

If the stage-like training holds for any resource (e.g., time, data, or parameters), the scaling law can be derived using the ratio of change in loss per skill (Eq. (43)) and the ratio of change with respect to the resource (given by the emergent time in Eq. (45)). The quanta model in Michaud et al. [18] is an example where the stage-like training holds for all resources.

## D.3 Discussion on the effective decoupling of skills in neural networks

In Section 5, we have empirically demonstrated that the multilinear model predicts the emergence of a 2-layer NN (Fig. 1). In Section 6, we briefly discussed why NNs, despite their **lack** of the decoupling among the skills, behave similarly to the decoupled model with $g_k$s as fixed basis functions: the **stage-like training** in NNs – induced by the model's layerwise structure and power-law frequencies of the skills – effectively decouples the skills. In this subsection, we extend the discussion in more detail.

In NNs, even though $g_k$s are 'discovered' (feature learned) by non-tractable dynamics, we speculate that similar stage-like dynamics also hold in 'discovering' (feature learning) $g_k$s: parameters 'useful' for expressing more frequent skills will be updated significantly faster than parameters useful for expressing less frequent skills.

If skill discovery and saturation dynamics operate at different time scales (stages), with negligible interaction among the skills, the skill dynamics become effectively **decoupled**. Because the dynamics are decoupled in stages, NNs repeat the feature learning process – using the limited resource (time, data, parameters) to express the skill – for all skills with each iteration varying only in the scale of the resource (e.g. training time, number of observations, and number of hidden layer neurons): resulting in a similar emergence to our multilinear model.

A more concrete understanding of our speculation that feature learning also occurs in stages due to a layerwise structure is left for future work.

# E Derivation of the scaling law exponents

This section provides a detailed derivation of the scaling laws up to a rigor common in physics and engineering. For example, we approximate the Riemann sum as integral or treat $k$, the number of

skills, as a differentiable parameter. For more general and rigorous derivations including the prefactor constants, see Appendix J. Instead, for more intuition and the relationship to the quanta model in Michaud et al. [18], see Appendix D.

Table 4: **Summary of the scaling laws.** The leftmost column shows the bottleneck of the scaling law. The middle three columns show the resource values in terms of the bottleneck (either taken to infinity or proportional to the bottleneck). The last column shows the scaling exponent for the loss as power-law of the bottleneck where $\alpha + 1$ is the exponent of the Zipfian input data (Eq. (1)).

| Bottleneck | Time | Data | Parameter | Exponent |
|---|---|---|---|---|
| Time ($T$) | $T$ | $\infty$ | $\infty$ | $-\alpha/(\alpha+1)$ |
| Data ($D$) | $\infty$ | $D$ | $\infty$ | $-\alpha/(\alpha+1)$ |
| Parameter ($N$) | $\infty$ | $\infty$ | $N$ | $-\alpha$ |
| Compute ($C$) | $C^{(\alpha+1)/(\alpha+2)}$ | $\infty$ | $C^{1/(\alpha+2)}$ | $-\alpha/(\alpha+2)$ |

### E.1 Time scaling law exponent

To derive the time scaling law exponent, we assume the time as the bottleneck and take $N, D \to \infty$. By using the decoupled dynamics of each skill loss (Lemma 1),

$$\mathcal{L}_k = \frac{S^2}{2\left(1 + \left(\frac{S}{\mathcal{R}_k(0)} - 1\right)^{-1} e^{2\eta \frac{d_k}{D} ST}\right)^2}. \tag{47}$$

Noting that $d_k/D \to \mathcal{P}_s(k)$ as $D \to \infty$, where $\mathcal{P}_s(k) = Ak^{-(\alpha+1)}$, we have

$$\mathcal{L}_k = \frac{S^2}{2\left(1 + \left(\frac{S}{\mathcal{R}_k(0)} - 1\right)^{-1} e^{2\eta Ak^{-(\alpha+1)} ST}\right)^2}. \tag{48}$$

This is a function of $k^{-(\alpha+1)}T$ only, suggesting the **decoupling** dynamics for each skill. Thus,

$$\frac{d\mathcal{L}_k}{dT} = -\frac{k}{(\alpha+1)T}\frac{d\mathcal{L}_k}{dk}. \tag{49}$$

Using Eq. (6) and taking $N, n_s \to \infty$ at the same rate,[4] we can approximate the loss as an integral instead of a sum over $k$:

$$\mathcal{L} \approx \lim_{N \to \infty} \int_1^N Ak^{-(\alpha+1)}\mathcal{L}_k dk, \tag{50}$$

where $A$ is the normalization constant for $\mathcal{P}_s$. We can differentiate the loss and use Eq. (49) to express the equation in terms of $k$:

$$\frac{d\mathcal{L}}{dT} = \lim_{N \to \infty} \int_1^N Ak^{-(\alpha+1)}\frac{d\mathcal{L}_k}{dT}dk = -\lim_{N \to \infty}\frac{1}{(\alpha+1)T}\int_1^N Ak^{-\alpha}\frac{d\mathcal{L}_k}{dk}dk. \tag{51}$$

Integrating by parts, we obtain

$$\frac{d\mathcal{L}}{dT} = -\lim_{N \to \infty}\frac{1}{(\alpha+1)T}\left[Ak^{-\alpha}\mathcal{L}_k\right]_1^N - \lim_{N \to \infty}\frac{\alpha}{(\alpha+1)T}\int_1^N Ak^{-(\alpha+1)}\mathcal{L}_k dk \tag{52}$$

$$= -\lim_{N \to \infty}\mathcal{O}\left(N^{-\alpha}\frac{1}{T}\right) + \mathcal{O}\left(\frac{1}{Te^T}\right) - \frac{\alpha}{(\alpha+1)T}\mathcal{L}. \tag{53}$$

The first term goes to 0 as $N \to \infty$ and the second term goes to 0 exponentially faster compared to the last term for $T \gg 1$, which leads to the scaling law with exponent $-\alpha/(\alpha+1)$:

$$\frac{d\mathcal{L}(T)}{\mathcal{L}(T)} = -\frac{\alpha}{\alpha+1}\frac{dT}{T}. \tag{54}$$

---

[4]We take $N$ and $n_s$ to $\infty$ at the same rate since we do not want the number of parameters to be a bottleneck in this setup.

**Finite $N$ correction for small $\alpha$.** In Fig. 7, we observe that our model with $\alpha = 0.1$ deviates from the expected power-law with exponent $-\alpha/(\alpha + 1)$. The deviation can be explained by the antiderivative term in Eq. (52):

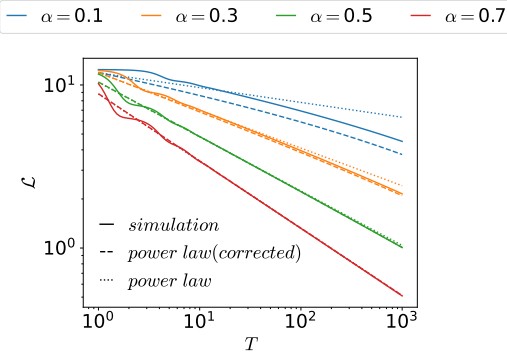

Figure 7: **Scaling law and corrected predictions.** A simulation of our multilinear model with $N = 50,000$ (solid), a scaling law with exponent $-\alpha/(\alpha + 1)$ (dotted), and a corrected scaling law considering finite $N$ (dashed, Eq. (56)). The finite $N$ corrected scaling law better predicts the dynamics, especially for smaller $\alpha$.

$$\lim_{N \to \infty} \left[ \frac{1}{2(\alpha + 1)} \frac{S^2 A}{\left(1 + \frac{1}{S/\mathcal{R}_k(0)-1}e^{2\eta SAk^{-(\alpha+1)}T}\right)^2} \frac{k^{-\alpha}}{T} \right]_1^N = \lim_{N \to \infty} \left( \mathcal{O}\left(N^{-\alpha}\frac{1}{T}\right) - \mathcal{O}\left(\frac{1}{Te^T}\right) \right).$$

(55)

The second term ($k = 1$) goes to 0 faster than $\mathcal{O}(T^{-1})$ for sufficiently larger $T$ but the first term ($k = N$) may not decay fast enough for finite $N$ and sufficiently small $\alpha$. For example, $N = 50,000$ and $\alpha = 0.1$ leads to $N^{-\alpha} \approx 0.3$, which is not negligibly small.

Assuming finite $N$ and small $\alpha$ such that the first term in Eq. (55) is non-negligible, we can rewrite Eq. (52) as

$$\frac{d\mathcal{L}}{dT} \approx -\frac{\alpha}{(\alpha + 1)}\frac{\mathcal{L} + \mathcal{L}_C}{T}, \qquad \mathcal{L}_C \approx S^2 A N^{-\alpha}/2\alpha, \tag{56}$$

where we assumed a small initialization $S/\mathcal{R}_k(0) \gg 1$ and sufficiently large number of parameters $N^{\alpha+1} \gg T$ to approximate $\mathcal{L}_C$. Because the total loss at initialization is $\mathcal{L}(0) = S^2/2$, $\mathcal{L}_C$ is non-negligible compared to the loss for sufficiently small $\alpha$. Thus considering $\mathcal{L}_C$, we obtain the corrected power-law which better approximates the time scaling law (dashed lines in Fig. 7). For a rigorous and comprehensive analysis of the time scaling law, see Theorem 2 and Theorem 3 in Appendix J.

### E.2  Data scaling law exponent

In this section, we derive the data scaling law exponent. The data scaling law assumes $T \to \infty$ and $N \to \infty$ with data as the bottleneck. From the decoupled dynamics of the multilinear model (Lemma 1), we can show that our model is a one-shot learner (Corollary 1):

**One shot learner.** *Given that $N > k$, $T \to \infty$, and $d_k$ is the number of samples from the training set with $g_k(i, x) \neq 0$, the $k^{th}$ skill loss after training is*

$$\mathcal{L}_k(\infty) = \begin{cases} 0 & : d_k > 0 \\ (S - \mathcal{R}_k(0))^2/2 \approx S^2/2 & : d_k = 0. \end{cases} \tag{57}$$

**Proof** See Appendix C.2.  ∎

Our model requires only one sample from the $k^{th}$ skill to learn such a skill, similar to how language models are few-shot learners at inference.[5] The model can one-shot learn a skill since it has $g_k$ as the basis functions, and the dynamics among different skills are decoupled. A similar one-shot learner has been studied in Hutter [20] where the error depends on a single 'observation' of a feature.

Because the $k^{th}$ skill loss **only depends** on $d_k$ (number of observations for the $k^{th}$ skill), we can calculate the expectation of the skill loss for $D$ data points from $P_{observed}(k|D)$ or the probability that $d_k > 0$:

$$P_{observed}(k|D) = 1 - (1 - \mathcal{P}_s(k))^D . \tag{58}$$

Using the one-shot learning property (Eq. (57)), the probability of observing the $k^{th}$ skill (Eq. (58)), and the decomposition of the loss into skill losses (Eq. (6)), the expected loss for $D$ datapoints is

$$\mathbf{E}_D\left[\mathcal{L}\right] = \frac{1}{2}\sum_{k=1}^{\infty} S^2 \mathcal{P}_s(k)(1 - P_{observed}(k)) \tag{59}$$

$$= \frac{1}{2}S^2 A \sum_{k=1}^{\infty} k^{-(\alpha+1)} \left(1 - \mathcal{P}_s(k)\right)^D \tag{60}$$

$$\approx \frac{1}{2}S^2 A \int_1^{\infty} k^{-(\alpha+1)} \left(1 - Ak^{-(\alpha+1)}\right)^D dk, \tag{61}$$

where the expectation $\mathbf{E}_D$ is over all possible training sets of size $D$, and $A$ is the normalization constant such that $\mathcal{P}(k) = Ak^{-(\alpha+1)}$. The difference in the loss $\Delta\mathcal{L} = \mathbf{E}_{D+1}\left[\mathcal{L}\right] - \mathbf{E}_D\left[\mathcal{L}\right]$ is

$$\Delta\mathcal{L} = \frac{1}{2}S^2 A \int_1^{\infty} k^{-(\alpha+1)} \left(1 - Ak^{-(\alpha+1)}\right)^D \left(\left(1 - Ak^{-(\alpha+1)}\right) - 1\right) dk \tag{62}$$

$$= -\frac{1}{2}S^2 A^2 \int_1^{\infty} k^{-2(\alpha+1)} \left(1 - Ak^{-(\alpha+1)}\right)^D dk. \tag{63}$$

We can integrate $\Delta\mathcal{L}$ by parts.

$$\Delta\mathcal{L} = \frac{1}{2}\left[-\frac{S^2 Ak^{-\alpha}}{(\alpha+1)(D+1)}\left(1 - Ak^{-(\alpha+1)}\right)^{D+1}\right]_1^{\infty}$$

$$-\frac{S^2 A\alpha}{2(\alpha+1)(D+1)}\int_1^{\infty} k^{-(\alpha+1)}\left(1 - Ak^{-(\alpha+1)}\right)^{D+1} dk$$

$$\approx \mathcal{O}\left((1 - \mathcal{P}_s(1))^{D+1}\right) - \frac{S^2 A\alpha}{2(\alpha+1)(D+1)}\int_1^{\infty} k^{-(\alpha+1)}\left(1 - Ak^{-(\alpha+1)}\right)^D \left(1 - Ak^{-(\alpha+1)}\right) dk$$

$$\approx -\frac{\alpha}{(\alpha+1)(D+1)}\mathbf{E}_D\left[\mathcal{L}\right] + \frac{\alpha}{(\alpha+1)(D+1)}\Delta\mathcal{L}.$$

In the second line, the first term goes to 0 for $D \gg 1$. In the last line, we used the expression for $\Delta\mathcal{L}$ (Eq. (62)) and $\mathbf{E}_D\left[\mathcal{L}\right]$ (Eq. (59)). Rearranging the equation above and using that $D \gg 1$, we obtain the scaling law with exponent $-\alpha/(\alpha+1)$:

$$\frac{\Delta\mathcal{L}}{\mathbf{E}_D\left[\mathcal{L}\right]} = -\frac{\alpha}{1 + (\alpha+1)D} \approx -\frac{\alpha}{(\alpha+1)}\frac{1}{D} \tag{64}$$

$$= -\frac{\alpha}{(\alpha+1)}\frac{\Delta D}{D}. \tag{65}$$

where in the last line, $\Delta D/D = 1/D$ as the change in the number of data points relative to $D$ is one.

### E.3 Parameter scaling law exponent

The parameter scaling law assumes $T \to \infty$ and $D \to \infty$, with the parameters $N < n_s$ as the bottleneck. Because our model is a one-shot learner (Eq. (57)), learning of the $k^{th}$ skill **only depends** on the existence of $g_k$ in the model; the model with $[g_1, \cdots, g_N]$ will learn all $k \le N$ skills with $\mathcal{L}_k = 0$.

The $\mathcal{L}_k$ dependence on $g_k$ is formalized in Corollary 2, which we repeat here.

---

[5]Few-shot learning is typically discussed in the context of models that have undergone pre-training (see, e.g. [1]). We speculate that expanding in the basis $g_k$ in our framework can model aspects of the pre-training process.

**Equivalence between a basis function and a skill.**   *Given $T, D \to \infty$ and if the multilinear model has the $N$ most frequent skill functions as a basis,*

$$\mathcal{L}_k(\infty) = \begin{cases} 0 & : k \leq N \\ S^2/2 & : k > N. \end{cases} \tag{66}$$

**Proof**  See Appendix C.3.                                                  ∎

Using Eq. (66) and Eq. (6), we can express the total loss as function of $N$:

$$\mathcal{L} \approx \frac{S^2}{2} \int_{N+1}^{\infty} Ak^{-(\alpha+1)}dk \propto (N+1)^{-\alpha}. \tag{67}$$

By approximating $N \approx N + 1$ for $N \gg 1$, we obtain the power-law with exponent $-\alpha$.

### E.4   Optimal compute scaling law

For analytical tractability, we define compute as $C := T \times N$. We start from Eq. (12) with $D \to \infty$

$$\mathcal{L} \approx \int_{1}^{N} Ak^{-(\alpha+1)}\mathcal{L}_k dk + \lim_{n_s \to \infty} \frac{S^2}{2} \int_{N}^{n_s} Ak^{-(\alpha+1)}dk. \tag{68}$$

We can use Eq. (56) to calculate the first term and integrate the last term to get

$$\mathcal{L} \approx (\mathcal{L}(0) + \mathcal{L}_C)T^{-\alpha/(\alpha+1)} - \mathcal{L}_c + \frac{S^2 A}{2\alpha}N^{-\alpha} \tag{69}$$

$$\approx \mathcal{O}(T^{-\alpha/(\alpha+1)}) + \mathcal{O}(N^{-\alpha}), \tag{70}$$

where we used that $\mathcal{L}(0) \gg \mathcal{L}_C$ and $S^2 A/(2\alpha) - \mathcal{L}_C > 0$. Intuitively, the approximation shows the tradeoff between $T$ – when increased, decreases the loss of the first $N$ skills – and $N$ – when increased, decreases the loss at sufficiently large $T$ – for fixed compute $C$. For a comprehensive analysis of the approximation above, see Appendix J.

Removing the irrelevant constant terms,

$$\mathcal{L} = T^{-\alpha/(\alpha+1)} + N^{-\alpha}. \tag{71}$$

We can use the method of Lagrangian multiplier to obtain

$$-\frac{\alpha}{\alpha+1}T^{-\alpha/(\alpha+1)-1} + \lambda N = 0, \tag{72}$$

$$-\alpha N^{-(\alpha+1)} + \lambda T = 0, \tag{73}$$

$$NT - C = 0, \tag{74}$$

where $\lambda$ is the Lagrange multiplier and $C$ is compute. We can solve the above set of equations to obtain $T^{\alpha+1} \propto N$ or equivalently

$$T \propto C^{(\alpha+1)/(\alpha+2)}, \quad N \propto C^{1/(\alpha+2)}. \tag{75}$$

We can plug it in Eq. (71) to get

$$\mathcal{L} \propto C^{-\alpha/(\alpha+2)}. \tag{76}$$

This derivation is similar to that of Bordelon et al. [21] (see Appendix N: Compute Optimal Scaling from Sum of Power-Laws in [21]). For a rigorous derivation of the optimal compute scaling law, see Corollary 4 and Appendix J.

# F  Derivation of the extended multilinear model

In this section, we show the derivation for the extended multilinear model.

## F.1  Gradient flow in the extended multilinear model

**Lemma 2.** *Let the extended multilinear model Eq. (15) be trained with gradient flow on $D$ i.i.d samples for the setup in Section 2 (input distribution: Eq. (1), target function: Eq. (4), and MSE loss: Eq. (5)). For the skill index $k \leq N$ be a skill index in the multilinear model, let the feature matrix $\Phi \in \mathbb{R}^{D_c \times d_k}$ for the $k^{th}$ skill be*

$$\Phi_{lj} = e_{k,l}(i^{(j)} = k, x^{(j)}), \tag{77}$$

*and SVD on $\Phi = USV$. Assuming that the system is overparametrized ($d_k < D_c$), the gradient on $\vec{B}_k \in \mathbb{R}^{D_c}$ ($[B_{k,1}, \cdots, B_{k,D_c}]$) is contained in the column space of semi-orthogonal matrix $U \in \mathbb{R}^{D_c \times d_k}$:*

$$UU^T \frac{d\vec{B}_k}{dt} = \frac{d\vec{B}_k}{dt}. \tag{78}$$

**Proof** Similar to Lemma 1, the total loss can be decomposed into each skill such that the dynamics of $B_{k,l}$ relies only on $d_k$ observations of the $k^{th}$ skill:

$$\mathcal{L}_D = \frac{1}{2D} \sum_{k=1}^{n_s} \sum_{j=1}^{D} \left( f^*(i^{(j)}, x^{(j)}) - f(i^{(j)}, x^{(j)}) \right)^2 \tag{79}$$

$$= \frac{1}{2D} \sum_{k=1}^{n_s} \sum_{j_k=1}^{d_k} \left( Sg_k(k, x^{(j_k)}) - \sum_{l=1}^{D_c} a_k B_{k,l} e_{k,l}(k, x^{(j_k)}) \right)^2 \tag{80}$$

$$= \frac{1}{2D} \sum_{k=1}^{n_s} \sum_{j_k=1}^{d_k} \left( \sum_{l=1}^{D_c} (\frac{S}{\sqrt{D_c}} - a_k B_{k,l}) e_{k,l}(k, x^{(j_k)}) \right)^2. \tag{81}$$

In the second line, we used Eq. (16) that $e_{k,l}(I \neq k, x) = 0$ and the orthogonality of $g_k$ (Eq. (3)). In the last line, we used Eq. (16) that $g_k = D_c^{-1/2} \sum_l e_{k,l}$. We can find the gradient descent equation of $B_{k,l}$ from Eq. (81):

$$\frac{dB_{k,l}}{dt} = -\eta \sum_{j=1}^{d_k} \frac{1}{D} \left[ a_k e_{k,l}(k, x^{(j)}) \sum_{l'=1}^{D_c} (a_k B_{k,l'} - \frac{S}{\sqrt{D_c}}) e_{k,l'}(k, x^{(j)}) \right], \tag{82}$$

which in the matrix form is

$$\frac{d\vec{B}_k}{dt} = -\frac{\eta a_k}{D} \Phi\Phi^T \left( B_k a_k - \frac{\vec{S}}{\sqrt{D_c}} \right), \tag{83}$$

where $D_c$ dimensional vectors $\vec{B}_k$ and $\vec{S}$ are $[B_{k,1}, \cdots, B_{k,D_c}]$ and $[S, \cdots, S]$ respectively. It illustrates that $\frac{d\vec{B}_k}{dt}$ is contained in $\text{im}(\Phi)$, which is contained in $\text{im}(U)$ (immediate from $\Phi = USV$). As $UU^T(Uz) = U(U^TU)z = Uz$, $UU^T$ acts as identity on image of $U$, showing that $UU^T \frac{d\vec{B}_k}{dt} = \frac{d\vec{B}_k}{dt}$.

∎

## F.2  Conserved quantity of extended multilinear model

**Lemma 3.** *In the setup of Lemma 2, $a_k^2 - |\vec{B}_k|^2$ is conserved over time.*

**Proof** We can use Eq. (81) to find the equation for $a_k$:

$$\frac{da_k}{dt} = -\eta \sum_{j=1}^{d_k} \frac{1}{D} \left[ \sum_l B_{k,l} e_{k,l}(k, x^{(j)}) \sum_{l'=1}^{D_c} (a_k B_{k,l'} - \frac{S}{\sqrt{D_c}}) e_{k,l'}(k, x^{(j)}) \right], \tag{84}$$

which in the matrix form is

$$\frac{da_k}{dt} = -\frac{\eta}{D} \vec{B}_k^T \Phi \Phi^T \left( \vec{B}_k a_k - \frac{\vec{S}}{\sqrt{D_c}} \right). \tag{85}$$

Then

$$a_k \frac{da_k}{dt} = -\frac{\eta a_k}{D} \vec{B}_k^T \Phi \Phi^T \left( \vec{B}_k a_k - \frac{\vec{S}}{\sqrt{D_c}} \right) \tag{86}$$

$$= \vec{B}_k^T \frac{d\vec{B}_k}{dt}, \tag{87}$$

where we used Eq. (83) in the last line. Thus, $a_k^2 - |\vec{B}_k|^2$ is conserved during the dynamics. ∎

### F.3 $D_c$ shot learner

**Proposition 1.** *Let the setup be as that in Lemma 2. Suppose that $a_k(T)$ is eventually bounded away from zero, i.e. there exists $\delta > 0$ and $M > 0$ such that $T > M \Rightarrow |a_k(T)| \geq \delta$. Also assume that $U^\perp$-component of $\vec{B}_k(0)a_k(0)$ and $\vec{B}_k(0)S$ is negligible. Then the skill strength $\mathcal{R}_k$ is*

$$\mathcal{R}_k(\infty) = \begin{cases} d_k < D_c : & S\left(1 - \sqrt{1 - d_k/D_c}\right) \\ d_k \geq D_c : & S \end{cases} \tag{88}$$

**Proof** First, we show that $\frac{d\mathcal{L}_k}{dt} \leq 0$ with equality only holding when the gradient is 0.

$$\frac{d\mathcal{L}_k}{dt} = \frac{d\mathcal{L}_k}{da_k}\frac{da_k}{dt} + \sum_i^{D_c} \frac{d\mathcal{L}_k}{dB_{k,i}}\frac{dB_{k,i}}{dt} \tag{89}$$

$$= -\eta \frac{d_k}{D} \left( \frac{d\mathcal{L}_k}{da_k}\frac{d\mathcal{L}_k}{da_k} + \sum_i^{D_c} \frac{d\mathcal{L}_k}{dB_{k,i}}\frac{d\mathcal{L}_k}{dB_{k,i}} \right) \leq 0. \tag{90}$$

The equality holds only when

$$\frac{d\mathcal{L}_k}{da_k} = \frac{da_k}{dt} = 0 \quad \text{and} \quad \frac{d\mathcal{L}_k}{dB_{k,i}} = \frac{dB_{k,i}}{dt} = 0. \tag{91}$$

We show that both $a_k$ and $\vec{B}_k$ are bounded throughout whole dynamics. As

$$\mathcal{L}_k = \left| \Phi \left( \vec{B}_k a_k - \frac{\vec{S}}{\sqrt{D_c}} \right) \right|^2 \geq \sigma^2 \left| UU^T \left( \vec{B}_k a_k - \frac{\vec{S}}{\sqrt{D_c}} \right) \right|^2 \tag{92}$$

for $\sigma^2$ the smallest nonzero eigenvalue of $\Phi\Phi^T$, where $\Phi = USV$. This shows that

$$UU^T \left( \vec{B}_k a_k - \frac{\vec{S}}{\sqrt{D_c}} \right) \tag{93}$$

is bounded, so $UU^T \vec{B}_k a_k$ is bounded. Meanwhile, in Lemma 2, we showed that $(1 - UU^T)\frac{d\vec{B}_k}{dt} = 0$, so $(1 - UU^T)\vec{B}_k a_k$ is bounded. This shows that $\vec{B}_k a_k$ is bounded. As $a_k^2 - |\vec{B}_k|^2$ is constant (Lemma 3) and $|\vec{B}_k a_k| = |a_k||\vec{B}_k|$ is bounded, this shows that both $a_k$ and $|\vec{B}_k|$ are bounded.

The dynamics moving in some bounded region always has at least one accumulation point, which we denote as $p$. We will show that $\frac{d\mathcal{L}_k}{dt} = 0$ at $p$. The function $\mathcal{L}_k(t)$ in $t$ is a decreasing differential function which is positive. We also note that $\frac{d^2\mathcal{L}_k(t)}{dt^2}$ is globally bounded, as it can be expressed in polynomial expression in $(a_k, \vec{B}_k)$ and we showed that $(a_k(t), \vec{B}_k(t))$ is bounded. From Taylor's theorem, one can obtain

$$\inf \mathcal{L}_k(t) \leq \mathcal{L}_k(t_1 + t_2) \leq \mathcal{L}_k(t_1) + t_2 \frac{d\mathcal{L}_k}{dt}(t_1) + \frac{t_2^2}{2}M \tag{94}$$

for $M = \sup |\frac{d^2 \mathcal{L}_k(t)}{dt^2}|$. Choosing $t_2 = -\frac{d\mathcal{L}_k}{dt}(t_1)M^{-1}$ shows that

$$\mathcal{L}_k(t_1) - \frac{1}{2M}\left(\frac{d\mathcal{L}_k}{dt}(t_1)\right)^2 \geq \inf \mathcal{L}_k(t) \tag{95}$$

and letting $t_1 \to \infty$ here gives

$$\lim_{t_1 \to \infty} \frac{1}{2M}\left(\frac{d\mathcal{L}_k}{dt}(t_1)\right)^2 \leq \lim_{t_1 \to \infty}(\mathcal{L}_k(t_1) - \inf \mathcal{L}_k(t)) = 0 \tag{96}$$

so $\frac{d\mathcal{L}_k}{dt} \to 0$ as $t \to \infty$. Meanwhile, as $p$ is accumulation point of $(a_k, B_k)$, $\frac{d\mathcal{L}_k}{dt}(p)$ is accumulation point of $\frac{d\mathcal{L}_k}{dt}(a_k(t), \vec{B}_k(t))$. As $\lim_{t \to \infty} \frac{d\mathcal{L}_k}{dt}(t) = 0$, the only accumulation point of $\frac{d\mathcal{L}_k}{dt}(t)$ is zero, which shows that $\frac{d\mathcal{L}_k}{dt}(p) = 0$.

We have seen that $a_k^2 - |\vec{B}_k|^2$ and $(I - UU^T)\vec{B}_k$ are conserved in our dynamics. A quantity conserved in dynamics should also be conserved at $p$, so $p = (a, \vec{B})$ should satisfy the following conditions:

- $a^2 - |\vec{B}|^2 = a_k(0)^2 - |\vec{B}_k(0)|^2$ (Lemma 3);
- $(I - UU^T)\vec{B} = (I - UU^T)\vec{B}_k(0)$ (Lemma 2);
- $\frac{d\mathcal{L}_k}{dt}(a, \vec{B}) = 0$, or equivalently the gradient is 0 at $p$.

We will solve for $p$ satisfying those three conditions. The third condition is equivalent to that

$$aUU^T\left(\vec{B}a - \frac{\vec{S}}{\sqrt{D_c}}\right) = 0. \tag{97}$$

As $a_k(T)$ is eventually bounded away from zero, we have $a \neq 0$, so

$$UU^T\left(\vec{B}a - \frac{\vec{S}}{\sqrt{D_c}}\right) = 0. \tag{98}$$

It follows that

$$\vec{B} = UU^T\vec{B} + (I - UU^T)\vec{B} = UU^T\frac{\vec{S}}{\sqrt{D_c}}a^{-1} + (I - UU^T)\vec{B}_k(0) \tag{99}$$

and substituting to first condition gives

$$a^2 - \frac{1}{a^2}\left|UU^T\frac{\vec{S}}{\sqrt{D_c}}\right|^2 - \left|(I - UU^T)\vec{B}_k(0)\right|^2 = a_k(0)^2 - |\vec{B}_k(0)|^2. \tag{100}$$

This is equivalent to a quadratic equation in $a^2$, and has a following solution of

$$a^2 = \sqrt{\left|UU^T\frac{\vec{S}}{\sqrt{D_c}}\right|^2 + \frac{(a_k(0)^2 - |UU^T\vec{B}_k(0)|^2)^2}{4}} + \frac{a_k(0)^2 - |UU^T\vec{B}_k(0)|^2}{2}. \tag{101}$$

This shows that there are two candidates for $p$, with $a$ given as two square roots of Eq. (101) and $B$ determined from $a$ by Eq. (99). It is impossible for $\mathcal{L}_k(t)$ to have accumulation points both in regions $a > 0$ and $a < 0$, as it would imply $a_k(t) = 0$ happens infinitely many often, contradicting that $a_k$ is eventually bounded away from zero. Thus it follows that $\mathcal{L}_k(t)$ can only have one accumulation point. As dynamics having unique accumulation point should converge, it follows that

$$(a, \vec{B}) = (a_k(\infty), \vec{B}_k(\infty)). \tag{102}$$

One can check that the $U^\perp$-component of $\vec{B}_k(\infty)a_k(\infty)$ is given as

$$(I - UU^T)\vec{B}_k(\infty)a_k(\infty) = (I - UU^T)\vec{B}_k(0)a_k(0) \tag{103}$$

and this is bounded by $|(1 - UU^T)B_k(0)|(S + a_k(0))$, so by our assumption this is negligible. Thus, we find that $\vec{B}_k(\infty)a_k(\infty)$ is the pseudo-inverse solution, which is also found by the linear

model with $e_{k,l}$ as basis functions. We can calculate $\mathcal{L}_k(\infty)$ using the result from kernel (linear) regression [23–27] (for a summary, see tables 1 and 2 in appendix A of [27]). Using the terminology in table 1 of [27], the sample size is $d_k$; the number of parameters is $D_c$; ridge and noise are absent; the eigenfunctions are $[e_{k,1}, \cdots, e_{k,D_c}]$; the eigen coefficients are $\mathbf{E}_X[e_{k,i}(x)Sg_k(x)] = SD_c^{-1/2}$ (Eq. (16)); eigenvalues are uniform; the learnability is $d_k/D_c$ for all $i$; and the overfitting coefficient is $(1 - d_k/D_c)^{-1}$. Taking into account that we have halved the MSE loss (Eq. (5)), the test loss is

$$\mathcal{L}_k(\infty) = \frac{S^2}{2}\left(1 - \frac{d_k}{D_c}\right). \tag{104}$$

We obtain the result by using Eq. (10). ∎

## F.4 $N_c$ basis functions for a skill

**Proposition 2.** *Let the extended multilinear model Eq. (18) be trained with gradient flow on $D \to \infty$ i.i.d samples for the setup in Section 3 with $n_s \to \infty$ (input distribution: Eq. (1), target function: Eq. (4), and MSE loss: Eq. (5), initialization: that of Proposition 1). For a model with the following finite $N$ basis functions*

$$[e_{1,1},\ \cdots,\ e_{1,N_c},\ e_{2,1},\ \cdots,\ e_{q,r}], \tag{105}$$

*where quotient $q = \lfloor (N-1)/N_c \rfloor + 1$ and remainder $r$ is such that $(q-1)N_c + r = N$. The skill strength at $T \to \infty$ becomes*

$$\mathcal{R}_k(\infty) = \begin{cases} k > q : & 0 \\ k = q : & S\frac{r}{N_c} \\ k < q : & S. \end{cases} \tag{106}$$

**Proof** Because we have $D \to \infty$ and because $[e_{k,1}, \cdots e_{k,N_c}]$ can express $g_k$ (Eq. (20)), it is trivial to show that $\mathcal{R}_k(\infty) = S$ for $k < q$. For $k = q$, the gradient descent dynamics (Eq. (83)) leads to

$$\frac{d\vec{B}_k}{dt} = -\frac{\eta a_k}{D}\Phi\Phi^T\left(\vec{B}_k a_k - \frac{\vec{S}}{\sqrt{N_c}}\right) \tag{107}$$

where the matrix $\Phi \in \mathbb{R}^{r \times d_k}$ and vector $\vec{B}_k \in \mathbb{R}^r$ are the feature matrix(Eq. (77)) and parameters for the $k^{th}$ skill respectively. As $D \to \infty$, the matrix $\Phi\Phi^T$ becomes a rank $r$ identity matrix scaled by the frequency of the skill:

$$\lim_{D \to \infty} \frac{1}{D}(\Phi\Phi^T)_{ll'} = \mathbf{E}_{I,X}\left[e_{k,l}(k,X)e_{k,l'}(k,X)\right] = \mathcal{P}(k)\delta_{l,l'}. \tag{108}$$

Plugging in $\Phi\Phi^T$,

$$\frac{dB_{k,l}}{dt} = -\eta\mathcal{P}(k)a_k\left(B_{k,l}a_k - \frac{S}{\sqrt{N_c}}\right). \tag{109}$$

Assuming the initialization in Proposition 1, we can show that $a_k(\infty)B_{k,l}(\infty) = S/\sqrt{N_c}$ for $l \leq r$. From Eq. (7), the skill strength $\mathcal{R}_k(\infty)$ is

$$\mathcal{R}_k(\infty) = \sum_{l=1}^{r} \frac{S}{\sqrt{N_c}}\mathbf{E}_X\left[e_{k,l}(k,X)g_k(k,X)\right] \tag{110}$$

$$= S\frac{r}{N_c}, \tag{111}$$

where we used Eq. (20) for the linear correlation between $e_{k,l}$ and $g_k$. ∎

# G  Time emergence example in NN

In this section, we discuss an example for the time emergence case (Fig. 1(a)) in which the saturation of skill in an NN consists of multiple saturating 'modes' as in Fig. 8.

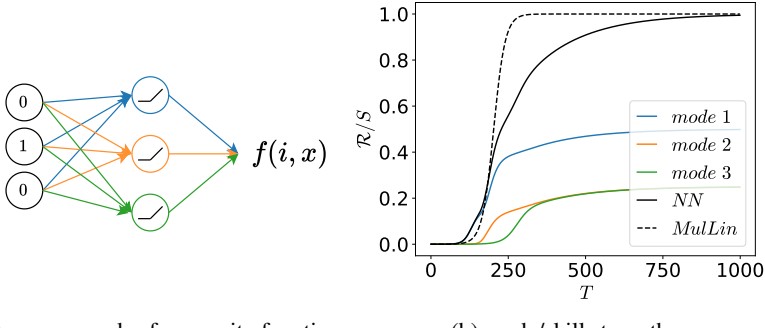

(a) neuron modes for a parity function          (b) mode/skill strength

Figure 8: **Modes in NN.** A 2-layer MLP with ReLU activations with a width of 3 and weight sharing (Eq. (114)) is trained to fit the parity function. **(a):** The skill strength $\mathcal{R}$, because of the last layer's linearity, can be decomposed into skill strength from each hidden neuron or each 'mode' (shown in different colors, Eq. (119)). **(b):** The skill strength for each mode follows a near-sigmoidal curve with different emergent/saturation times (colors) whose sum results in the total skill strength (solid black). Note that different saturation times of each mode result in a deviation from the prediction of the multilinear model with $\mathcal{B}^2 = 1/3$ (dashed black).

**Task.**   We assume an input $X \in \mathbb{R}^{3 \times 8}$ (note that we are not using $X$ as a random variable) that is all 8 possible inputs for bits with dimension 3. The target $Y$ is the parity function scaled by $S$.

$$X = \begin{pmatrix} 0 & 0 & 0 & 0 & 1 & 1 & 1 & 1 \\ 0 & 0 & 1 & 1 & 0 & 0 & 1 & 1 \\ 0 & 1 & 0 & 1 & 0 & 1 & 0 & 1 \end{pmatrix}, \qquad Y = \begin{pmatrix} S & -S & -S & S & -S & S & S & -S \end{pmatrix} \tag{112}$$

**NN.**   We assume a 2-layer width 3 NN with ReLU activation with the input dimension 3 (Fig. 8(a)). The NN has 16 parameters, but to simplify the argument, we use weight sharing so NN has only 4 parameters:

$$f(x; \alpha, \beta, \gamma, c) = w^T \sigma(Wx + b) + c \tag{113}$$

where $\sigma$ is the ReLU activation and $W, b, w$ are

$$W = \begin{pmatrix} -\alpha & \alpha & -\alpha \\ -\beta & \beta & -\beta \\ \gamma & -\gamma & \gamma \end{pmatrix}, \qquad b = \begin{pmatrix} 0 \\ \beta \\ -\gamma \end{pmatrix}, \qquad w = \begin{pmatrix} -2\alpha \\ \beta \\ \gamma \end{pmatrix}. \tag{114}$$

**Modes.**   It is easy to see that $\alpha = \beta = \gamma = \sqrt{2S}$ and $c = -S$ leads to the target parity function. We note that one parameter except $c$ (i.e. $\alpha, \beta, \gamma$) maps to one neuron or a mode (colors in Fig. 8(a)). We define the first mode $f^{(1)}$ as

$$f^{(1)}(x) = w_1 \sigma(W_1^T x + b_1) = -2\alpha^2 \sigma(x_2 - x_1 - x_3) \tag{115}$$
$$= -2\alpha^2 h_1(x), \qquad h_1(x) := \sigma(x_2 - x_1 - x_3), \tag{116}$$

where $w_1, b_1$ are the first entry of $w, b$ respectively and $W_1$ is the first row of $W$. Note that $f^{(1)}(x)$ takes a form similar to the multilinear model (Eq. (9)) but with $h_1$ as the respective basis. We define $f^{(2)}, f^{(3)}$ similarly, and the sum of modes becomes the NN:

$$f(x) = \sum_{q=1}^{3} f^{(i)}(x) + c, \tag{117}$$

which resembles the multilinear model with different skills.

**Mode strength.**  Analogous to the skill strength in Eq. (7), we define mode $q$'s strength $\mathcal{R}^{(q)}$ as

$$\mathcal{R}^{(q)} = \frac{1}{8S^2} Y^T f^{(q)}(X), \tag{118}$$

where $f^{(q)}(X) = [f^{(q)}(X_1), \cdots, f^{(q)}(X_8)]$ and $X_j$ are the $j^{th}$ column of $X$. By the linearity of the expectation,

$$\mathcal{R} = \sum_{q=1}^{3} \mathcal{R}^{(q)}. \tag{119}$$

Note that constant $c$ always has zero correlation (inner product) to the target ($Y$).

**Analysis.**  The dynamics of each mode $\mathcal{R}^{(q)}(x)$ differs from that of the multilinear model (Eq. (11)) because $h_q(x)$ often depends on the parameter, and the dynamics are no longer decoupled among each mode. Nevertheless, each mode follows a sigmoid-like growth (Fig. 8(b)). We note that each mode has a different saturation time scale or is updated at different frequencies. A mode with a longer time scale leads to a longer 'tail' of saturation as discussed in the main text.

**Update frequency.**  Because of the non-linearity, each mode differs in the gradients it receives. We can explicitly calculate the gradient for each parameter as:

$$\frac{d\alpha^2}{dt} = 2\eta\alpha^2(-S - (-2\alpha^2 + 2\beta^2 + c)) \tag{120}$$

$$\frac{d\beta^2}{dt} = -\eta\beta^2(S - (-2\alpha^2 + 5\beta^2 + 5c)) \tag{121}$$

$$\frac{d\gamma^2}{dt} = -\eta\gamma^2(S - (\gamma^2 + c)) \tag{122}$$

$$\frac{dc}{dt} = -\eta(2\alpha^2 - 5\beta^2 - \gamma^2 - 8c). \tag{123}$$

We immediately notice that $c$ will grow the fastest for small initialization ($\alpha, \beta, \gamma, c \ll 1$) because it saturates exponentially while other parameters saturate sigmoidally. Considering that $S$ is always the largest term and $c$ saturate to $S$ quickly, we notice that the saturation is in the order of $\alpha^2$ ($\approx 2S + 2c \approx 4S$), $\beta^2$ ($\approx -S + 5c \approx 4S$), and $\gamma^2 (\approx 2S)$. We observe that our crude approximation holds in Fig. 8(b): the first ($\alpha$) and the second ($\beta$) modes saturate at similar timescale, while the third mode ($\gamma$) requires approximately twice the time for saturation.

# H Details of the multilinear model

The multilinear model (Fig. 9(a)) has two identifying properties: 1) the layerwise structure and 2) $g_k$ as the basis functions. In this section, we discuss the role of each property in more detail.

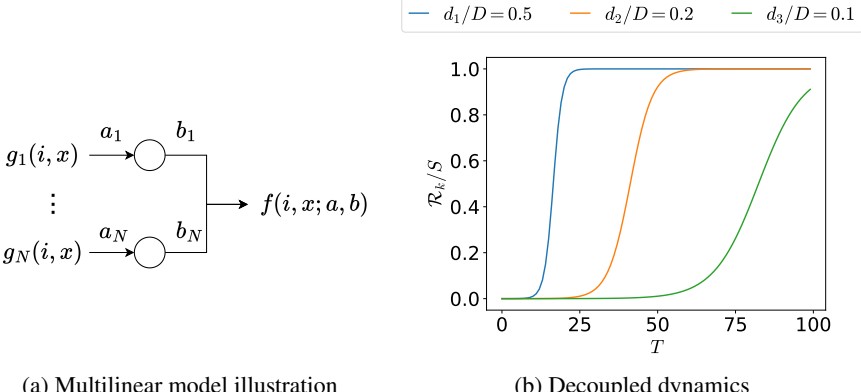

(a) Multilinear model illustration        (b) Decoupled dynamics

Figure 9: **Multilinear model. (a):** An illustration of the multilinear model which is multilinear in terms of parameters, generating a layerwise structure. The model also has the skill functions $g_k$s as basis functions. **(b):** The dynamics of the multilinear model are decoupled and each skill strength ($\mathcal{R}_k$) shows a sigmoidal growth in time. Note that less frequent skills have a more delayed growth.

**Multilinearity.** The product of two parameters ($a_k b_k$) creates the layerwise structure (Fig. 9(a)) that gives rise to the emerging dynamics (sudden saturation or sigmoidal growth) in Fig. 9(b). The time emergence of NN is well-described by the sigmoidal dynamics (Fig. 1(a)); a non-sigmoidal saturation dynamics, for example, that of linear models (Fig. 10(a)), would inadequately describe the time emergence. Such dynamics have first been studied by Saxe et al. [19] (See Appendix B.2 for an overview).

Assuming a sufficiently fast decay of $d_k$ for the skills, the sigmoidal growth results in a stage-like training (Appendix D) where one skill fully saturates before the next skill emerges. In Appendix D, we discuss how the stage-like training can describe the quanta model [18] and how NNs, without explicit $g_k$s, decouple each skill.

Finally, note that even though sigmoidal saturation has a resemblance to the test accuracy in grokking [48], our model is irrelevant to grokking because $\mathcal{R}_k$ – which is defined over the expectation over the $k^{th}$ skill (Eq. (7)) – appears both in the empirical loss (Eq. (11)) and the test loss: failing to describe the discrepancy between train and test accuracy in grokking.

**Connection to linear models.** In Section 4 and Appendix E, we have shown how the scaling laws follow from the basis functions $g_k$ that decouples the loss. To analyze the role of $g_k$, we can ask whether a simpler linear model with $g_k$ as basis functions (Eq. (124)) also recovers the scaling laws. The answer is yes and we outline how a linear model can recover all scaling laws. In addition, we also outline how extended linear models – extended similar to Section 5 such that skills are decoupled – can recover all emergence behaviors shown in Appendix F except the time emergence.

By replacing $a_k b_k$ with $w_k$, we obtain the linear model with skill basis functions:

$$f_T(i, x; w) = \sum_{k=1}^{N} w_k(T) g_k(i, x). \tag{124}$$

The dynamics of the linear model under gradient flow is

$$\mathcal{R}_k(T) = w_k(T) = S(1 - e^{-\eta \frac{d_k}{D} T}), \tag{125}$$

where we assumed $w_k(0) = 0$. The linear model follows an exponential saturation of the skill strength in contrast to the sigmoidal saturation of the multilinear model (Fig. 10).

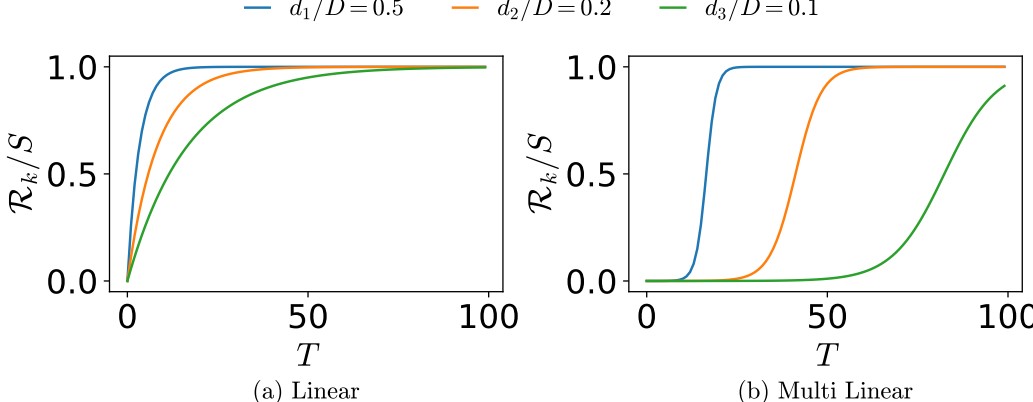

(a) Linear        (b) Multi Linear

Figure 10: **Dynamics of linear and multilinear model. (a):** Skill strength dynamics of the linear model (Eq. (125)) **(b):** Skill strength dynamics of the multilinear model (Eq. (11)). For the linear model, $\mathcal{R}_k$ emerges from $T = 0$ for all $d_k/D > 0$: obstructing the stage-like training. For the multilinear model, $\mathcal{R}_k$ shows a delayed emergence depending on $d_k/D$: allowing the stage-like training and describing the sigmoidal time emergence in Fig. 1(a).

Nevertheless, the linear model Eq. (125) results in the same scaling laws in Section 4. For the time scaling law, we recover the relationship between $d\mathcal{L}_k/dT$ and $d\mathcal{L}_k/dk$ in Appendix E.1 because $\mathcal{R}_k(T)$ is a function of $\frac{d_k}{D}T$ only (where $d_k/D = \mathcal{P}_s(k)$ for $D \to \infty$). For the data scaling law, we recover Corollary 1 because each $w_k$ (i.e. $\mathcal{R}_k$) is decoupled. For the parameter scaling law, we recover Corollary 2 trivially as the linear model shares the same basis functions.

The data and parameter emergence in Section 5 can be obtained from the linear model in Eq. (124) if we extend the model analogous to Eqs. (15) and (18). For example, we can extend the model for data emergence as

$$f_T(i, x; W) = \sum_{k=1}^{N} \sum_{l=1}^{D_c} W_{k,l}(T) e_{k,l}(i, x), \tag{126}$$

where the matrix $W \in \mathbb{R}^{N \times D_c}$ is an extension of $w \in \mathbb{R}^N$ in Eq. (124), $D_c$ is a fixed scalar, and $e_{k,l}(i, x) : \{0, 1\}^{n_s + n_b} \to \mathbb{R}$ are functions with the following properties:

$$\mathbf{E}_{X|I=k}\left[e_{k,l}e_{k,l'}\right] = \delta_{ll'}, \quad e_{k,l}(I \neq k, x) = 0, \quad \sum_{l=1}^{D_c} \frac{1}{\sqrt{D_c}} e_{k,l} = g_k. \tag{127}$$

The equivalence can be shown by Lemma 2 which states that the multilinear model finds the minimum norm solution: the solution that the linear model finds in a ridgeless regression setup.

Thus, for our setup, the basis functions play a critical role in the scaling laws and data/parameter emergences. The choice of basis functions, also known as the task-model alignment (see [23, 27]), determines the linear model's scaling laws and emergence behaviors. See Bordelon et al. [21] for a study of the scaling laws in linear models.

# I   Additional plots and tables

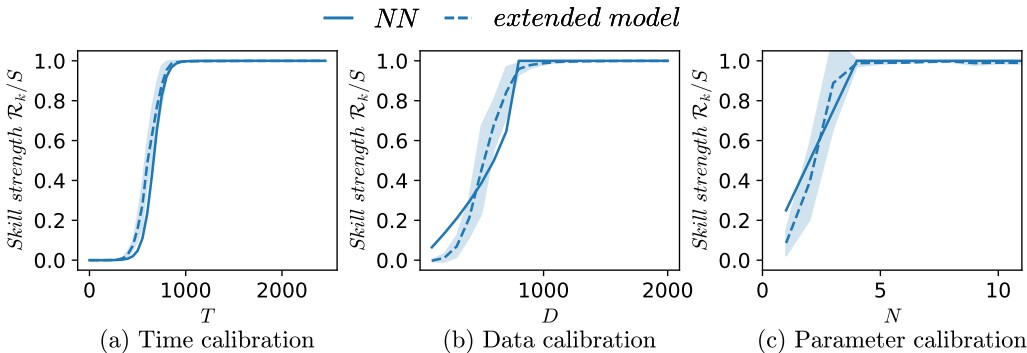

(a) Time calibration       (b) Data calibration       (c) Parameter calibration

Figure 11: **Calibration and prediction on emergence.** The calibration of the extended multilinear model (solid) on the 2-layer NN (dashed) for $n_s = 1$ system. For the calibrated parameters, we have $\mathcal{B}^2 = 1/22$ for time (Eq. (14)), $D_c = 800$ for data (Eq. (17)), and $N_c = 4$ for hidden layer width (Eq. (21)).

Table 5: **Samples of skill strength $\mathcal{R}_k/S$.** The table shows the skill strength at $N = 10$ for 10 different runs of the parameter emergence experiment (Fig. 1(c)). Note that the variance of $\mathcal{R}_k/S$ is amplified by the outliers – shaded columns – that learn a less frequent skill at the cost of a more frequent skill (second column) or fail to learn a skill (seventh column).

| | | | | | | | | | | |
|---|---|---|---|---|---|---|---|---|---|---|
| $k = 1$ | 0.98 | 0.98 | 0.98 | 0.98 | 0.98 | 0.98 | 0.98 | 0.98 | 0.98 | 0.98 |
| $k = 2$ | 4.5 | 0.95 | 0.95 | 0.95 | 0.96 | 0.96 | 0.04 | 0.96 | 0.96 | 0.95 |
| $k = 3$ | 0.6 | 0.0 | 0.72 | 0.90 | 0.92 | 0.64 | 0.88 | 0.8 | 0.58 | 0.52 |
| $k = 4$ | 0.0 | 0.78 | 0.0 | 0.0 | 0.0 | 0.0 | 0.0 | 0.0 | 0.0 | 0.0 |
| $k = 5$ | 0.0 | 0.0 | 0.0 | 0.0 | 0.0 | 0.0 | 0.0 | 0.0 | 0.0 | 0.0 |

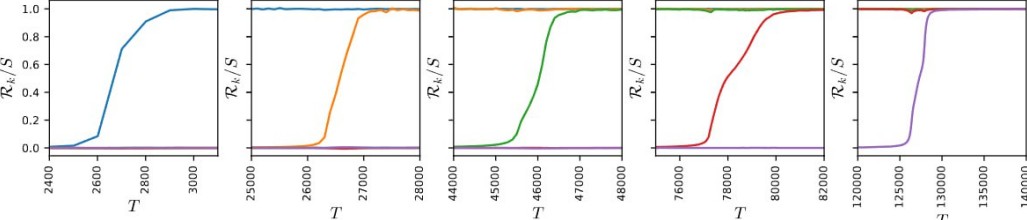

Figure 12: **Enlarged emergence.** Enlarged view of skill emergence from Fig. 4, showing that saturations also follow a sigmoidal pattern. The $x$-axis is measured in steps.

## J Rigorous derivation of the scaling laws

In Appendix E, we discussed the scaling laws in simplified settings, favoring intuition over mathematical rigor. Building upon the intuitive understanding developed in Appendix E, we now turn our attention to a rigorous analysis of the scaling laws. In this section, we will derive general scaling laws by considering a comprehensive set of parameters and variables. Our goal is to establish the conditions under which these scaling laws hold and to quantify the associated error terms. By explicitly analyzing the error terms, this section aims to provide a rigorous assessment of the validity and limitations of our scaling law estimates.

Table 6: **Scaling laws and their conditions.** The leftmost column indicates the condition for the 'large resource' – large enough to be treated as infinity, while the second column is the condition between the other two resources for the scaling law (third column). The last two columns show where the statement for the prefactor constant (e.g. $\mathcal{A}_N$ for scaling law $\mathcal{L} = \mathcal{A}_N N^{-\alpha}$) and the scaling law (with the assumptions and explicit error terms) are given. Note that whenever $T$ appears in theorems and corollaries, $\eta S$ is multiplied to make it dimensionless.

| Large resource | Condition | Scaling law | Constant | Statement |
|---|---|---|---|---|
| $D \gg T^3$ | $N^{\alpha+1} = o(T)$ | $\mathcal{L} = \mathcal{A}_N N^{-\alpha}$ | Theorem 1 | Theorem 1 |
| $D \gg NT^2, T^3$ | $N^{\alpha+1} \gg T$ | $\mathcal{L} = \mathcal{A}_T T^{-\alpha/(\alpha+1)}$ | Theorem 4 | Theorems 2 and 3 |
| $D \gg T^3$ | $N^{\alpha+1} \approx T$ | $\mathcal{L} = \mathcal{A}_C C^{-\alpha/(\alpha+2)}$ | Corollary 5 | Corollary 4 |
| $T \gg D(\log D)^{1+\epsilon}$ | $N^{\alpha+1} = o(D)$ | $\mathcal{L} = \mathcal{A}_N N^{-\alpha}$ | Theorem 5 | Theorem 5 |
| $T \gg D(\log D)^{1+\epsilon}$ | $N^{\alpha+1} \gg D$ | $\mathcal{L} = \mathcal{A}_D D^{-\alpha/(\alpha+1)}$ | Theorem 5 | Theorem 5 |

### J.1 General set up, repeated

We go back to the most general settings possible. Our starting point is Eq. (27), which describes the dynamics of $\mathcal{R}_k$ and $\mathcal{L}_k$ valid for $k \le N$:

$$\mathcal{L}_k = \frac{S^2}{2 \left( 1 + \left( \frac{S}{\mathcal{R}_k(0)} - 1 \right)^{-1} e^{2\eta \frac{d_k}{D} ST} \right)^2} \tag{27}$$

We do not use skills for indices $k > N$ in our model, but we can still denote

$$\mathcal{R}_k = 0 \quad \text{and} \quad \mathcal{L}_k = \frac{S^2}{2}. \tag{128}$$

For $\mathcal{P}_s(k) = Ak^{-\alpha-1}$, the total loss is given as

$$\mathcal{L} = \sum_{k=1}^{n_s} \mathcal{P}_s(k) \mathcal{L}_k = \sum_{k=1}^{N} \mathcal{P}_s(k) \mathcal{L}_k + \sum_{k=N+1}^{n_s} \mathcal{P}_s(k) \frac{S^2}{2}. \tag{129}$$

When $n_s, N, T$ are all set, their dependency with the data is only determined by the statistics $d_k$, the number of data with $i^{(j)} = k$. We assumed that $(i, x) \in I \times \{0, 1\}^{n_d}$ was collected as random samples with $i$ following the Zipfian distribution of size $n_s$ and exponent $\alpha + 1$, or equivalently $P(i = k) = \mathcal{P}_s(k) = Ak^{-\alpha-1}$ for $1 \le k \le n_s$. Then $(d_1, \cdots, d_{n_s})$ is a vector denoting the number of occurrences in $D$ independent sampling from that distribution. It follows that $d_i$ follows binomial distribution $B(D, \mathcal{P}_s(k))$.

In this complete perspective, our loss is dependent on all of those parameters and variables

$$\mathcal{L} = \mathcal{L}(n_S, \mathcal{D}, \mathcal{R}_{init}, N, T) \tag{130}$$

where $\mathcal{R}_{init} = (\mathcal{R}_1(0), \cdots, \mathcal{R}_N(0))$ denotes the vector representing initial condition. We will also simply denote $r_k = \mathcal{R}_k(0)$. We will not assume much on $r_k$, but we absolutely need $0 < r_k < S$ for dynamics to hold, and we also should have

$$\sum_{k=1}^{n_s} \mathcal{P}_s(k) r_k^2 = \mathbf{E}[f(0)^2] \ll S^2. \tag{131}$$

We will not impose any particular distribution on $\mathcal{R}_{init}$. Instead, we will try to identify sufficient conditions on $r_k$ for our desired result to hold, and those conditions will differ by the situation we are considering. For example, in Theorems 2 and 3 where we prove time scaling law $\mathcal{L} = \Theta(T^{-\alpha/(\alpha+1)})$ for large enough $D$ and bottleneck $T$, we only require $\epsilon < r_k < S/2$ for some $\epsilon > 0$. However, the exact constant depends on the distribution of $r_k$, and figuring out the explicit constant seems to be only feasible when we fix $r_k = r$ as in Theorem 4.

## J.2 Estimates for large $D$

We will first consider the situation where $D$ becomes the 'large resource' so that its effect on the loss function is negligible. The number of data $d_k$ follows binomial distribution $B(D, \mathcal{P}_s(k))$, so $d_k/D$ converges to $\mathcal{P}_s(k)$ for large enough $D$. So taking the limit of $\mathcal{L}$ when we let $D \to \infty$ has the effect of replacing $d_k/D$ by $\mathcal{P}_s(k)$ in the expression of $\mathcal{L}$. We will establish an explicit inequality comparing the difference between $\mathcal{L}$ and this limit.

**Lemma 4.** *For a function $F : \mathbb{R} \to \mathbb{R}$ with its total variation $V(F)$ bounded, we have*

$$\left| \mathbf{E}_{\mathcal{D}} \left[ F(\frac{d_k}{D}) \right] - \mathbf{E}_{z \sim \mathcal{N}(\mathcal{P}_s(k), \mathcal{P}_s(k)(1-\mathcal{P}_s(k))/D)} \left[ F(z) \right] \right| < \frac{V(F)}{\sqrt{D}\sqrt{\mathcal{P}_s(k)(1-\mathcal{P}_s(k))}} \tag{132}$$

*where $\mathcal{N}(\mu, \sigma^2)$ denotes normal distribution of mean $\mu$ and variance $\sigma^2$.*

**Proof** This is just an application of the Berry-Esseen inequality (with constant 1, see [49] for modern treatment) applied to $d_k$ following binomial distribution $B(D, \mathcal{P}_s(k))$. ∎

**Lemma 5.** *Let $F : \mathbb{R} \to \mathbb{R}$ be a $C^2$ function such that $F''$ is bounded. Then we have*

$$\left| \mathbf{E}_{z \sim \mathcal{N}(\mathcal{P}_s(k), \mathcal{P}_s(k)(1-\mathcal{P}_s(k))/D)} \left[ F(z) \right] - F(\mathcal{P}_s(k)) \right| \leq \frac{\mathcal{P}_s(k)(1-\mathcal{P}_s(k))}{2D} \sup |F''|. \tag{133}$$

**Proof** First, we apply Taylor's theorem to show that

$$|F(z) - F(\mathcal{P}_s(k)) - F'(\mathcal{P}_s(k))(z - \mathcal{P}_s(k))| \leq \frac{(z - \mathcal{P}_s(k))^2}{2} \sup |F''|. \tag{134}$$

Taking expectation when $z$ follows normal distribution $\mathcal{N}(\mathcal{P}_s(k), \frac{\mathcal{P}_s(k)(1-\mathcal{P}_s(k))}{D})$ gives

$$|\mathbf{E}_z \left[ F(z) - F(\mathcal{P}_s(k)) \right]| = |\mathbf{E}_z \left[ F(z) - F(\mathcal{P}_s(k)) - F'(\mathcal{P}_s(k))(z - \mathcal{P}_s(k)) \right]| \tag{135}$$

$$\leq \mathbf{E}_z \left[ |F(z) - F(\mathcal{P}_s(k)) - F'(\mathcal{P}_s(k))(z - \mathcal{P}_s(k))| \right] \tag{136}$$

$$\leq \mathbf{E}_z \left[ \frac{(z - \mathcal{P}_s(k))^2}{2} \sup |F''| \right] \tag{137}$$

$$= \frac{\mathcal{P}_s(k)(1-\mathcal{P}_s(k))}{2D} \sup |F''|. \tag{138}$$

∎

**Proposition 3.** *We have*

$$\left| \mathbf{E}_{\mathcal{D}} \left[ \mathcal{L}_k \right] - \frac{S^2}{2 \left( 1 + \left( \frac{S}{r_k} - 1 \right)^{-1} e^{2\eta \mathcal{P}_s(k)ST} \right)^2} \right| < \frac{2^\alpha S^2}{\sqrt{D\mathcal{P}_s(k)}} + \frac{4S^4 \eta^2 T^2 \mathcal{P}_s(k)}{D}. \tag{139}$$

**Proof** Consider the function $F : \mathbb{R} \to \mathbb{R}$ given as

$$F(z) = \frac{S^2}{2 \left( 1 + \left( \frac{S}{r_k} - 1 \right)^{-1} e^{2\eta STz} \right)^2}. \tag{140}$$

This function is monotone decreasing and $C^2$ on the whole domain, and its supremum and infimum are given as

$$\sup F = \lim_{z \to -\infty} F(z) = \frac{S^2}{2} \quad \text{and} \quad \inf F = \lim_{z \to \infty} F(z) = 0. \tag{141}$$

This implies that

$$V(F) = \sup F - \inf F = \frac{S^2}{2}. \tag{142}$$

Also, we will show that $F''$ is globally bounded. We first calculate

$$F''(z) = -4S^3 r_k (1 - \frac{r_k}{S})^2 \eta^2 T^2 \frac{e^{2\eta STz}(1 - \frac{r_k}{S} - \frac{2r_k}{S}e^{2\eta STz})}{\left(1 - \frac{r_k}{S} + \frac{r_k}{S}e^{2\eta STz}\right)^4}. \tag{143}$$

We consider the following inequalities

$$e^{2\eta STz} \leq \frac{S}{r_k}\left(1 - \frac{r_k}{S} + \frac{r_k}{S}e^{2\eta STz}\right) \tag{144}$$

$$\left|1 - \frac{r_k}{S} - \frac{2r_k}{S}e^{2\eta STz}\right| \leq \left|1 - \frac{r_k}{S}\right| + \frac{2r_k}{S}e^{2\eta STz} < 2\left(1 + \frac{r_k}{S}(e^{2\eta STz} - 1)\right) \tag{145}$$

to show that

$$|F''(z)| < 4S^3 r_k (1 - \frac{r_k}{S})^2 \eta^2 T^2 \frac{\frac{2S}{r_k}\left(1 - \frac{r_k}{S} + \frac{r_k}{S}e^{2\eta STz}\right)^2}{\left(1 - \frac{r_k}{S} + \frac{r_k}{S}e^{2\eta STz}\right)^4} < 8S^4 \eta^2 T^2 \tag{146}$$

for all $z$. Thus we can apply both Lemma 4 and Lemma 5 to this function $F$ and we have

$$\left|\mathbf{E}_{\mathcal{D}}\left[F(\frac{d_k}{D})\right] - F(\mathcal{P}_s(k))\right| < \frac{V(F)}{\sqrt{D}\sqrt{\mathcal{P}_s(k)(1 - \mathcal{P}_s(k))}} + \frac{\mathcal{P}_s(k)(1 - \mathcal{P}_s(k))}{2D}\sup|F''|$$

$$< \frac{S^2}{2\sqrt{D}\sqrt{\mathcal{P}_s(k)(1 - \mathcal{P}_s(k))}} + \frac{4\mathcal{P}_s(k)S^4\eta^2 T^2}{D}$$

$$< \frac{2^\alpha S^2}{\sqrt{D\mathcal{P}_s(k)}} + \frac{4\mathcal{P}_s(k)S^4\eta^2 T^2}{D} \tag{147}$$

where the last line follows from that we always have

$$1 - \mathcal{P}_s(k) \geq 1 - \mathcal{P}_s(1) = \frac{2^{-(\alpha+1)} + \cdots + n_s^{-(\alpha+1)}}{1 + 2^{-(\alpha+1)} + \cdots + n_s^{-(\alpha+1)}} > \frac{2^{-(\alpha+1)}}{1 + 2^{-(\alpha+1)}} > \frac{1}{2^{2(\alpha+1)}}. \tag{148}$$

∎

**Lemma 6.** *For any integer $N$ and $\sigma \geq 1/2$ and $\sigma \neq 1$, we have*

$$\sum_{k=1}^{N} k^{-\sigma} = \zeta(\sigma) + \frac{N^{1-\sigma}}{1-\sigma} + O(N^{-\sigma}) \tag{149}$$

*where $\zeta$ is the Riemann zeta function (defined over the whole complex plane except $1$ via analytic continuation). In addition,*

$$\sum_{k=1}^{N} k^{-1} = \log N + \gamma + O(N^{-1}) \tag{150}$$

*where $\gamma = 0.5772156649...$ is Euler's constant.*

**Proof** See Corollary 1.15 of [50], or other analytic number theory textbooks. ∎

**Proposition 4.** *(Large D approximation) We have*

$$\mathbf{E}_{\mathcal{D}}[\mathcal{L}] - \sum_{k=1}^{N} \mathcal{P}_s(k)\frac{S^2}{2\left(1 + \left(\frac{S}{r_k} - 1\right)^{-1}e^{2\eta\mathcal{P}_s(k)ST}\right)^2} - \sum_{k=N+1}^{n_s} \mathcal{P}_s(k)\frac{S^2}{2} \tag{151}$$

$$= O\left(S^2 D^{-1/2} f_\alpha(N) + S^4\eta^2 T^2 D^{-1}\right) \tag{152}$$

*where*

$$f_\alpha(N) = \begin{cases} 1 & \text{if } \alpha > 1 \\ \log N & \text{if } \alpha = 1 \\ N^{(1-\alpha)/2} & \text{if } \alpha < 1. \end{cases} \tag{153}$$

*The constant on the $O$ term only depends on $\alpha$.*

**Proof** From the description of $\mathcal{L}$ in Eq. (129), we have

$$\mathbf{E}_\mathcal{D}[\mathcal{L}] - \sum_{k=1}^N \mathcal{P}_s(k) \frac{S^2}{2\left(1 + \left(\frac{S}{r_k} - 1\right)^{-1} e^{2\eta \mathcal{P}_s(k)ST}\right)^2} - \sum_{k=N+1}^{n_s} \mathcal{P}_s(k)\frac{S^2}{2} \qquad (154)$$

$$= \sum_{k=1}^N \mathcal{P}_s(k) \left( \mathbf{E}_\mathcal{D}[\mathcal{L}_k] - \frac{S^2}{2\left(1 + \left(\frac{S}{r_k} - 1\right)^{-1} e^{2\eta \mathcal{P}_s(k)ST}\right)^2} \right). \qquad (155)$$

We apply Proposition 3 to give

$$\sum_{k=1}^N \mathcal{P}_s(k) \left( \mathbf{E}_\mathcal{D}[\mathcal{L}_k] - \frac{S^2}{2\left(1 + \left(\frac{S}{r_k} - 1\right)^{-1} e^{2\eta \mathcal{P}_s(k)ST}\right)^2} \right) < \sum_{k=1}^N \mathcal{P}_s(k)\left(\frac{2^\alpha S^2}{\sqrt{D\mathcal{P}_s(k)}} + \frac{4S^4\eta^2 T^2\mathcal{P}_s(k)}{D}\right).$$

$$(156)$$

Each of these sum involving $\mathcal{P}_s(k)$ is bounded as

$$\sum_{k=1}^N \mathcal{P}_s(k)^2 < \left(\sum_{k=1}^N \mathcal{P}_s(k)\right)^2 < 1 \qquad (157)$$

and

$$\sum_{k=1}^N \sqrt{\mathcal{P}_s(k)} < \sum_{k=1}^N k^{-(\alpha+1)/2} = O(f_\alpha(N)) \qquad (158)$$

which follows from Lemma 6. Combining those two gives

$$\sum_{k=1}^N \mathcal{P}_s(k)\left(\frac{2^\alpha S^2}{\sqrt{D\mathcal{P}_s(k)}} + \frac{S^4\eta^2 T^2\mathcal{P}_s(k)}{D}\right) = O\left(S^2 D^{-1/2} f_\alpha(N) + S^4\eta^2 T^2 D^{-1}\right). \qquad (159)$$

∎

While Proposition 4 holds for any $D$, it becomes only meaningful if the resulting error terms are less than the main term we desire. We will revisit this when the exact main term is found, and determine the sufficient size of $D$ for error terms to become small enough.

### J.3   Estimates for not too small $n_s$

We next discuss the effect of $n_s$. When $n_s \to \infty$ heuristically, then intuitively we have $\mathcal{P}_s(k) \to k^{-(\alpha+1)}/\zeta(\alpha+1)$. We will discuss the difference between when we regard $n_s$ as $\infty$ and when we do not.

**Proposition 5.** *The following equations hold:*

$$A^{-1} = \sum_{k=1}^{n_s} k^{-(\alpha+1)} = \zeta(\alpha+1) - \frac{n_s^{-\alpha}}{\alpha} + O(n_s^{-\alpha-1}) \qquad (160)$$

$$\mathcal{P}_s(k) = \frac{k^{-\alpha-1}}{\zeta(\alpha+1)}\left(1 + \frac{n_s^{-\alpha}}{\alpha\zeta(\alpha+1)}O(n_s^{-\alpha-1})\right) \qquad (161)$$

$$\sum_{k=N+1}^{n_s} \mathcal{P}_s(k) = \frac{N^{-\alpha} - n_s^{-\alpha}}{\alpha\zeta(\alpha+1)} + O(N^{-\min(\alpha+1,2\alpha)}) \qquad (162)$$

*All implied constants on $O$ only depend on $\alpha$.*

**Proof** The first statement Eq. (160) follows from substituting $\sigma = \alpha + 1$ in Lemma 6. As $\mathcal{P}_s(k) = Ak^{-(\alpha+1)}$, the second statement Eq. (161) immediately follows. If we substitute $n_s = N$ into Eq. (160) and calculate differences between them, we obtain

$$\sum_{k=N+1}^{n_s} k^{-\alpha-1} = \frac{N^{-\alpha} - n_s^{-\alpha}}{\alpha} + O(N^{-\alpha-1}). \tag{163}$$

Thus we have

$$\sum_{k=N+1}^{n_s} \mathcal{P}_s(k) = A \sum_{k=N+1}^{n_s} k^{-(\alpha+1)} = \frac{N^{-\alpha} - n_s^{-\alpha}}{\alpha\zeta(\alpha+1)} + O\left(N^{-\alpha-1} + (N^{-\alpha} - n_s^{-\alpha})n_s^{-\alpha}\right). \tag{164}$$

Regardless of the size of $n_s$, We always have

$$(N^{-\alpha} - n_s^{-\alpha})n_s^{-\alpha} \le \left(\frac{N^{-\alpha}}{2}\right)^2 = \frac{N^{-2\alpha}}{4} \tag{165}$$

so the third statement Eq. (162) follows. ∎

We go back to the description of total loss given in Eq. (129) as

$$\mathcal{L} = \sum_{k=1}^{N} \mathcal{P}_s(k)\mathcal{L}_k + \sum_{k=N+1}^{n_s} \mathcal{P}_s(k)\frac{S^2}{2} \tag{129}$$

and we take its expectation in $\mathcal{D}$. Proposition 4 suggests that its limit when $D \to \infty$ is given as

$$\lim_{D\to\infty} \mathbf{E}_{\mathcal{D}}[\mathcal{L}] = \sum_{k=1}^{N} \mathcal{P}_s(k) \frac{S^2}{2\left(1 + \left(\frac{S}{r_k} - 1\right)^{-1} e^{2\eta\mathcal{P}_s(k)ST}\right)^2} + \sum_{k=N+1}^{n_s} \mathcal{P}_s(k)\frac{S^2}{2}. \tag{166}$$

Denote

$$\mathscr{L}_1 = \sum_{k=1}^{N} \mathcal{P}_s(k) \frac{S^2}{2\left(1 + \left(\frac{S}{r_k} - 1\right)^{-1} e^{2\eta\mathcal{P}_s(k)ST}\right)^2} \tag{167}$$

$$\mathscr{L}_2 = \sum_{k=N+1}^{n_s} \mathcal{P}_s(k)\frac{S^2}{2}. \tag{168}$$

We discuss the effect of $n_s$ in $\mathscr{L}_1$ and $\mathscr{L}_2$, by comparing limit of $\mathscr{L}_1$ and $\mathscr{L}_2$ when $n_s \to \infty$ and their original values.

- For the term $\mathscr{L}_1$, the change of letting $n_s$ as finite value from $n_s \to \infty$ has effect of multiplying $T$ by $1 + n_s^{-\alpha}/(\alpha\zeta(\alpha+1))$, and multiplying whole $\mathscr{L}_1$ by $1 + n_s^{-\alpha}/(\alpha\zeta(\alpha+1))$. It can be equivalently put as

$$\mathscr{L}_1(n_s, N, T) = \left(1 + \frac{n_s^{-\alpha}}{\alpha\zeta(\alpha+1)} + O(n_s^{-\alpha-1})\right) \mathscr{L}_1\left(\infty, N, T\left(1 + \frac{n_s^{-\alpha}}{\alpha\zeta(\alpha+1)} + O(n_s^{-\alpha-1})\right)\right).$$
$$\tag{169}$$

  We always have $n_s > N$ and $N \to \infty$ eventually, so if dependency of $\mathscr{L}_1$ with respect to $T$ is at most polynomial order, then change of main term of $\mathscr{L}_1$ is negligible. We can't establish exact statements yet without the descriptions of size of $\mathcal{L}_1$.

- The term $\mathscr{L}_2$ only depends on $N$ and $n_s$, not on $T$. Applying Proposition 5 (especially Eq. (162)) gives

$$\mathscr{L}_2(n_s, N, T) = \frac{N^{-\alpha} - n_s^{-\alpha}}{\alpha\zeta(\alpha+1)} \frac{S^2}{2} + O(N^{-\min(\alpha+1,2\alpha)}S^2) \tag{170}$$

  When $n_s$ grows faster than $N$ then $n_s^{-\alpha}$ part is totally negligible, and when $n_s$ has same order as $N$ then $n_s^{-\alpha}$ affects the constant for main term of $\mathscr{L}_2$. Things might get little complicated when $n_s = N + o(N)$, where $N^{-\alpha} - n_s^{-\alpha} = o(N^{-\alpha})$ can happen then.

- Comparing size of $\mathscr{L}_1$ and $\mathscr{L}_2$ mainly depends on time. The term $\mathscr{L}_2$ is fixed, and $\mathscr{L}_1$ decreases as $T$ increases. For $T = \infty$ we have $\mathscr{L}_1 = 0$, so $\mathscr{L}_2$ having order $N^{-\alpha}$ dominates (this proves scaling law for $N$ of exponent $\alpha$), so restriction on $n_s$ becomes quite substantial. For small $T$ and large $N$ where the size of $\mathscr{L}_2$ is small, we can expect the restriction on $n_s$ to be less substantial. For example, in the extreme case $N = \infty$, we have $\mathscr{L}_2 = 0$, and $n_s$ does not matter at all (except that, of course, it should satisfy $n_s \geq N$).

For such reasons, it is hard to quantify exact conditions for $n_s$ such that error terms are controlled, unless we specify relative growth of $(N, T)$. However, $n_s = \omega(N)$ suffices to assure that setting $n_s = \infty$ has zero effect on the main term. We will not worry about $n_s$ in this setting anymore too, and come back to this at the very end to determine enough $n_s$.

### J.4 Estimating main terms

We assume $D = \infty$ and $n_s = \infty$ – virtually implying that $d_k/D = \mathcal{P}_s(k)$ and $\mathcal{P}_s(k) = k^{-\alpha-1}/\zeta(\alpha + 1)$ (calculated by rule of $n_s = \infty$). We decomposed our main term into

$$\lim_{n_s \to \infty} \lim_{D \to \infty} \mathbf{E}_{\mathcal{D}}[\mathcal{L}] = \mathscr{L}_1 + \mathscr{L}_2 \tag{171}$$

where

$$\mathscr{L}_1 = \sum_{k=1}^{N} \mathcal{P}_s(k) \frac{S^2}{2\left(1 + \left(\frac{S}{r_k} - 1\right)^{-1} e^{2\eta \mathcal{P}_s(k)ST}\right)^2} \tag{172}$$

and

$$\mathscr{L}_2 = \sum_{k=N+1}^{\infty} \mathcal{P}_s(k) \frac{S^2}{2}. \tag{173}$$

By Proposition 5, $\mathcal{L}_2$ is determined almost completely as

$$\mathscr{L}_2 = \frac{S^2 N^{-\alpha}}{2\alpha\zeta(\alpha + 1)} + O(N^{-\alpha-1}). \tag{174}$$

Now focus on $\mathscr{L}_1$. For

$$F(z) = \frac{S^2}{2\left(1 + \left(\frac{S}{r_k} - 1\right)^{-1} e^{2\eta STz}\right)^2} \tag{175}$$

(note: it really depends on $r_k$ so it is correct to write $F_k$, but for convenience we will keep using $F$.) one can express $\mathscr{L}_1$ as

$$\mathscr{L}_1 = \sum_{k=1}^{N} \mathcal{P}_s(k) F(\mathcal{P}_s(k)). \tag{176}$$

**Lemma 7.** *Let $F(z)$ be defined as Eq. (175).*

1. *(Estimate for large $z$) We have*

$$0 \leq F(z) \leq \frac{(S - r_k)^2}{2} \min\left(1, \frac{S^2}{r_k^2} e^{-4\eta STz}\right). \tag{177}$$

2. *(Estimate for small $z$) For $z \geq 0$, we have*

$$\frac{(S - r_k)^2}{2} - \frac{8\eta S^3 T}{27} z \leq F(z) \leq \frac{(S - r_k)^2}{2}. \tag{178}$$

**Proof**

1. The left side is obvious. For the right side, $F(z) \leq (S - r_k)^2/2$ follows from noting that $F(0) = \frac{(S-r_k)^2}{2}$ and proving $F'(z) \leq 0$, and $F(z) \leq \frac{(S-r_k)^2}{2} \frac{S^2}{r_k^2} e^{-4\eta STz}$ follows from just replacing $1 + \left(\frac{S}{r_k} - 1\right)^{-1} e^{2\eta STz}$ in the denominator of $F$ by $\left(\frac{S}{r_k} - 1\right)^{-1} e^{2\eta STz}$.

2. For the left side, it suffices to show $-F'(z) \le \frac{8\eta S^3 T}{27}$. One can calculate

$$F'(z) = -2S^2 r_k (1 - \frac{r_k}{S})^2 \eta T \frac{e^{2\eta STz}}{\left(1 + \frac{r_k}{S}(e^{2\eta STz} - 1)\right)^3} \tag{179}$$

and

$$F''(z) = -4S^3 r_k (1 - \frac{r_k}{S})^2 \eta^2 T^2 \frac{e^{2\eta STz}(1 - \frac{r_k}{S} - \frac{2r_k}{S}e^{2\eta STz})}{\left(1 + \frac{r_k}{S}(e^{2\eta STz} - 1)\right)^4} \tag{180}$$

so $F$ has unique inflection point at

$$1 - \frac{r_k}{S} - \frac{2r_k}{S}e^{2\eta STz} = 0 \quad \Rightarrow \quad e^{2\eta STZ} = \frac{1}{2}\left(\frac{S}{r_k} - 1\right) \tag{181}$$

and this point is where $-F'(z)$ obtains maximum. Substituting this to the expression of $F'(z)$ gives $-F'(z) = \frac{8\eta S^3 T}{27}$.

∎

Our threshold for distinguishing two approximation methods will be set as $z = z_0 = (\zeta(\alpha + 1)\eta ST)^{-1}$, where both two error terms are bounded by $O(S^2)$. The constant $\zeta(\alpha + 1)$ is set to make later calculations much easier. Applying Lemma 7 gives

$$\mathscr{L}_1 = \sum_{k=1}^{N} \mathcal{P}_s(k) F(\mathcal{P}_s(k)) \tag{182}$$

$$= \sum_{1 \le k \le N, \mathcal{P}_s(k) < z_0} \frac{(S - r_k)^2}{2} \mathcal{P}_s(k)$$

$$+ O\left(\eta S^3 T \sum_{1 \le k \le N, \mathcal{P}_s(k) < z_0} \mathcal{P}_s(k)^2 + S^2 \sum_{1 \le k \le N, \mathcal{P}_s(k) > z_0} \mathcal{P}_s(k) \min\left(1, \frac{S^2}{r_k^2}e^{-4\eta ST\mathcal{P}_s(k)}\right)\right). \tag{183}$$

Denote

$$\mathscr{M} = \sum_{1 \le k \le N, \mathcal{P}_s(k) < z_0} \frac{(S - r_k)^2}{2} \mathcal{P}_s(k) \tag{184}$$

$$\mathscr{E}_1 = \eta S^3 T \sum_{1 \le k \le N, \mathcal{P}_s(k) < z_0} \mathcal{P}_s(k)^2 \tag{185}$$

$$\mathscr{E}_2 = S^2 \sum_{1 \le k \le N, \mathcal{P}_s(k) > z_0} \mathcal{P}_s(k) \min\left(1, \frac{S^2}{r_k^2}e^{-4\eta ST\mathcal{P}_s(k)}\right). \tag{186}$$

**Proposition 6.** *Suppose that there exists $0 < r < \sqrt{S}$ such that $r \le r_k < S/2$ for all k. In the decomposition of*

$$\lim_{n_s \to \infty} \lim_{D \to \infty} \mathbf{E}_{\mathcal{D}}[\mathcal{L}] = \mathscr{M} + \mathscr{L}_2 + O(\mathscr{E}_1 + \mathscr{E}_2) \tag{187}$$

*given as above, we have the following bound.*

1. *If $(\eta ST)^{1/(\alpha+1)} > N$, then*

$$\mathscr{L}_2 = \frac{S^2 N^{-\alpha}}{2\alpha\zeta(\alpha + 1)} + O(S^2 N^{-\alpha-1}) \tag{188}$$

$$\mathscr{M} = \mathscr{E}_1 = 0 \tag{189}$$

$$\mathscr{E}_2 = O\left(S^2 (\log(S/r))^{\alpha/(\alpha+1)}(\eta ST)^{-\alpha/(\alpha+1)}\right) \tag{190}$$

2. If $(\eta ST)^{1/(\alpha+1)} < N$, then

$$\mathscr{L}_2 + \mathscr{M} = \Theta\left(S^2 \sum_{k>(\eta ST)^{1/(\alpha+1)}} \mathcal{P}_s(k)\right) = \Theta(S^2(\eta ST)^{-\alpha/(\alpha+1)}) \tag{191}$$

$$\mathscr{E}_1 = O\left(S^2(\eta ST)^{-\alpha/(\alpha+1)}\right) \tag{192}$$

$$\mathscr{E}_2 = O\left(S^2(\log(S/r))^{\alpha/(\alpha+1)}(\eta ST)^{-\alpha/(\alpha+1)}\right) \tag{193}$$

*Here all constants in $O$ and $\Theta$ terms are absolute with respect to $\eta, S, T, N$. (They may depend on $\alpha$.)*

**Proof** We first note that the condition $\mathcal{P}_s(k) < z_0 = (\zeta(\alpha+1)\eta ST)^{-1}$ is equivalent to

$$\mathcal{P}_s(k) < z_0 = (\zeta(\alpha+1)\eta ST)^{-1} \Leftrightarrow k^{-\alpha-1} < \frac{1}{\eta ST} \Leftrightarrow k > (\eta ST)^{1/(\alpha+1)}. \tag{194}$$

Thus we can rephrase the descriptions of terms as

$$\mathscr{M} = \sum_{(\eta ST)^{1/(\alpha+1)}<k\leq N} \frac{(S-r_k)^2}{2}\mathcal{P}_s(k) \tag{195}$$

$$\mathscr{E}_1 = \eta S^3 T \sum_{(\eta ST)^{1/(\alpha+1)}<k\leq N} \mathcal{P}_s(k)^2 \tag{196}$$

$$\mathscr{E}_2 = S^2 \sum_{k\leq \min((\eta ST)^{1/(\alpha+1)},N)} \mathcal{P}_s(k)\min\left(1, \frac{S^2}{r_k^2}e^{-4\eta ST\mathcal{P}_s(k)}\right). \tag{197}$$

Applying Proposition 5 easily shows that

$$\mathscr{L}_2 = \frac{S^2 N^{-\alpha}}{2\alpha\zeta(\alpha+1)} + O(S^2 N^{-\alpha-1}). \tag{198}$$

For $\mathscr{M}$ and $\mathscr{E}_1$, we will consider them by dividing two cases depending on whether $(\eta ST)^{1/(\alpha+1)} > N$ or $(\eta ST)^{1/(\alpha+1)} < N$. If $(\eta ST)^{1/(\alpha+1)} > N$, then the condition $(\eta ST)^{1/(\alpha+1)} < k \leq N$ is never satisfied, so $\mathscr{M} = \mathscr{E}_1 = 0$. Now suppose $(\eta ST)^{1/(\alpha+1)} < N$. We first note that

$$\mathscr{L}_2 + \mathscr{M} = \sum_{(\eta ST)^{1/(\alpha+1)}<k\leq N} \frac{(S-r_k)^2}{2}\mathcal{P}_s(k) + \sum_{k>N} \frac{S^2}{2}\mathcal{P}_s(k). \tag{199}$$

As $(S-r_k)^2 = \Theta(S^2)$, we can let

$$\mathscr{L}_2 + \mathscr{M} = \Theta\left(S^2 \sum_{k>(\eta ST)^{1/(\alpha+1)}} \mathcal{P}_s(k)\right) \tag{200}$$

and using Proposition 5 gives the desired estimate $\mathscr{L}_2 + \mathscr{M} = \Theta(S^2(\eta ST)^{-\alpha/(\alpha+1)})$. For $\mathscr{E}_1$, estimating sum of $\mathcal{P}_s(k)^2$ using Lemma 6 gives

$$\mathscr{E}_1 = O\left(\eta S^3 T \sum_{k>(\eta ST)^{1/(\alpha+1)}} k^{-2(\alpha+1)}\right) = O\left(S^2(\eta ST)^{-\alpha/(\alpha+1)}\right). \tag{201}$$

For $\mathscr{E}_2$ we always have

$$\mathscr{E}_2 \leq S^2 \sum_{k\leq(\eta ST)^{1/(\alpha+1)}} \mathcal{P}_s(k)\min\left(1, \frac{S^2}{r^2}e^{-4\eta ST\mathcal{P}_s(k)}\right) \tag{202}$$

regardless of the size of $N$, so it suffices to bound this sum. If we denote $l = (\eta ST)^{1/(\alpha+1)}$ and define

$$F_2(z) = \min\left(1, \frac{S^2}{r^2}e^{-4\eta STz}\right), \tag{203}$$

it suffices to show the bound

$$\sum_{k \leq l} \mathcal{P}_s(k) F_2(\mathcal{P}_s(k)) = O\left( (\log(S/r))^{\alpha/(\alpha+1)} (\eta ST)^{-\alpha/(\alpha+1)} \right). \tag{204}$$

We will approximate this sum as

$$\sum_{k \leq l} \mathcal{P}_s(k) F_2(\mathcal{P}_s(k)) = \sum_{k \leq l} (\mathcal{P}_s(k) - \mathcal{P}_s(k+1)) \frac{\mathcal{P}_s(k)}{\mathcal{P}_s(k+1) - \mathcal{P}_s(k)} F_2(\mathcal{P}_s(k)) \tag{205}$$

$$= \sum_{k \leq l} (\mathcal{P}_s(k) - \mathcal{P}_s(k+1)) \frac{k^{-\alpha-1}}{(\alpha+1)k^{-\alpha-2}(1 + O(k^{-1}))} F_2(\mathcal{P}_s(k)) \tag{206}$$

$$= O\left( \sum_{k \leq l} (\mathcal{P}_s(k) - \mathcal{P}_s(k+1)) \mathcal{P}_s(k)^{-1/(\alpha+1)} F_2(\mathcal{P}_s(k)) \right). \tag{207}$$

to obtain the form of Riemann sum approximation for the integral of

$$\int_{z=\mathcal{P}_s(l)}^{\infty} z^{-1/(\alpha+1)} F_2(z) dz \tag{208}$$

at $\mathcal{P}_s(l) < \mathcal{P}_s(l-1) < \cdots < \mathcal{P}_s(1)$. As $F_2(z)$ is decreasing function, this Riemann sum is always less than the integral, so we obtain

$$\sum_{k \leq l} \mathcal{P}_s(k) F_2(\mathcal{P}_s(k)) = O\left( \int_{z=\mathcal{P}_s(l)}^{\infty} z^{-1/(\alpha+1)} F_2(z) dz \right). \tag{209}$$

We note that $\mathcal{P}_s(l) = (\zeta(\alpha+1)\eta ST)^{-1}$. The threshold for $F_2(z)$ to become 1 is given at

$$\frac{S^2}{r^2} e^{-4\eta STz} = 1 \quad \Leftrightarrow \quad z = \frac{1}{2\eta ST} \log \frac{S}{r}. \tag{210}$$

As $r < \sqrt{S}$, this value is always greater than $\mathcal{P}_s(l)$. Thus we can divide our integral as

$$\int_{(\zeta(\alpha+1)\eta ST)^{-1}}^{\infty} z^{-1/(\alpha+1)} F_2(z) dz \tag{211}$$

$$= \int_{(\zeta(\alpha+1)\eta ST)^{-1}}^{(2\eta ST)^{-1} \log(S/r)} z^{-1/(\alpha+1)} dz + \int_{(2\eta ST)^{-1} \log(S/r)}^{\infty} z^{-1/(\alpha+1)} \frac{S^2}{r^2} e^{-4\eta STz} dz. \tag{212}$$

The first part is bounded by

$$\int_{(\zeta(\alpha+1)\eta ST)^{-1}}^{(2\eta ST)^{-1} \log(S/r)} z^{-1/(\alpha+1)} dz = O\left( ((2\eta ST)^{-1} \log(S/r))^{\alpha/(\alpha+1)} \right) \tag{213}$$

which can be shown to be $O\left( (\log(S/r))^{\alpha/(\alpha+1)} (\eta ST)^{-\alpha/(\alpha+1)} \right)$. For the second part, we apply substitution of $w = 4\eta STz$ to show

$$\int_{(2\eta ST)^{-1} \log(S/r)}^{\infty} z^{-1/(\alpha+1)} \frac{S^2}{r^2} e^{-4\eta STz} dz = \frac{S^2}{r^2} (4\eta ST)^{-\alpha/(\alpha+1)} \int_{2\log(S/r)}^{\infty} w^{-1/(\alpha+1)} e^{-w} dw \tag{214}$$

$$= \frac{S^2}{r^2} (4\eta ST)^{-\alpha/(\alpha+1)} \Gamma\left( \frac{\alpha}{\alpha+1}, 2\log \frac{S}{r} \right) \tag{215}$$

and applying the asymptotic $\Gamma(s, x) = O(x^{s-1} e^{-x})$ suggests that this is bounded by

$$\ll \frac{S^2}{r^2} (4\eta ST)^{-\alpha/(\alpha+1)} \left( \log \frac{S}{r} \right)^{-1/(\alpha+1)} e^{-2\log(S/r)} = O\left( (\eta ST)^{-\alpha/(\alpha+1)} \right). \tag{216}$$

$$\blacksquare$$

**Theorem 1.** *(Parameter scaling law) Assume the following conditions: $n_s > N$ with $\lim(N/n_s) = \gamma < 1$ ($\gamma$ can be zero), and there exists $0 < r < \sqrt{S}$ such that $r < \mathcal{R}_k(0) < S/2$ for all $k$. If $N, T \to \infty$ while satisfying $N^{\alpha+1} = o(T)$, the expected loss $\mathbf{E}_{\mathcal{D}}[\mathcal{L}]$ for all datasets $\mathcal{D}$ of size $D$ satisfies*

$$
\begin{aligned}
\mathbf{E}_{\mathcal{D}}[\mathcal{L}] = {}& \frac{S^2(1-\gamma^\alpha)}{2\alpha\zeta(\alpha+1)} N^{-\alpha} \\
& + O\left(S^2 N^{-\min(\alpha+1,2\alpha)} + S^2 \left(\log(S/r)\right)^{\alpha/(\alpha+1)} (\eta ST)^{-\alpha/(\alpha+1)}\right) \\
& + O\left(S^2 D^{-1/2} f_\alpha(N) + S^4 \eta^2 T^2 D^{-1}\right),
\end{aligned}
\tag{217}
$$

*where*

$$
f_\alpha(N) = \begin{cases} 1 & \text{if } \alpha > 1 \\ \log N & \text{if } \alpha = 1 \\ N^{(1-\alpha)/2} & \text{if } \alpha < 1. \end{cases}
\tag{218}
$$

*The constant on the $O$ term only depends on $\alpha$. When $D \gg T^3$, then all the error terms involving $D$ are negligible.*

**Proof** In the situation $n_s = \infty$ and $D = \infty$, Proposition 6 shows that

$$
\mathbf{E}_{\mathcal{D}}[\mathcal{L}] = \frac{S^2}{2\alpha\zeta(\alpha+1)} N^{-\alpha} + O\left(S^2 N^{-(\alpha+1)} + S^2 \left(\log(S/r)\right)^{\alpha/(\alpha+1)} (\eta ST)^{-\alpha/(\alpha+1)}\right).
\tag{219}
$$

We consider the effect of $n_s$ first. As $\mathscr{L}_1$ becomes an error term in this estimation, letting $n_s$ as a finite value has no effect on overall estimation. The term $\mathscr{L}_2$ accounts for the main term, and letting $n_s$ as finite value changes it to

$$
\frac{N^{-\alpha} - n_s^{-\alpha}}{\alpha\zeta(\alpha+1)} \frac{S^2}{2} + O(N^{-\min(\alpha+1,2\alpha)} S^2).
\tag{220}
$$

This accounts for the factor $(1 - \gamma^\alpha)$ on the main term and $O(N^{-\min(\alpha+1,2\alpha)} S^2)$ added to the error term. The effect of $D$ is exactly described in Proposition 4, contributing the error term of $O\left(S^2 D^{-1/2} f_\alpha(N) + S^4 \eta^2 T^2 D^{-1}\right)$. Regarding the sufficient condition for $D$, if $D \gg T^3$ then we have

$$
S^4 \eta T^2 D^{-1} \ll T^{-\alpha/(\alpha+1)}, \quad S^2 D^{-1/2} f_\alpha(N) \ll T^{-3/2} N^{1/2} \ll T^{-1}
\tag{221}
$$

so all error terms involving $D$ are less than $O(T^{-\alpha/(\alpha+1)})$. $\blacksquare$

For the situation $T = O(N^{\alpha+1})$ however, the error terms $\mathscr{E}_1$ and $\mathscr{E}_2$ are of same size, so we can only say that the main term is of $O(S^2(\eta ST)^{-\alpha/(\alpha+1)})$.

**Theorem 2.** *(Upper bound for the time scaling law) Assume the following conditions: $n_s > N$, and there exists there exists $0 < r < \sqrt{S}$ such that $r < \mathcal{R}_k(0) < S/2$ for all $k$. If $N, T \to \infty$ while satisfying $\eta ST = O(N^{\alpha+1})$, the expected loss $\mathbf{E}_{\mathcal{D}}[\mathcal{L}]$ is*

$$
\mathbf{E}_{\mathcal{D}}[\mathcal{L}] = O\left(S^2 \left(\log(S/r)\right)^{\alpha/(\alpha+1)} (\eta ST)^{-\alpha/(\alpha+1)} + S^2 D^{-1/2} f_\alpha(N) + S^4 \eta^2 T^2 D^{-1}\right)
\tag{222}
$$

*with constant on $O$ only depending on $\alpha$ and $\limsup((\eta ST)^{1/(\alpha+1)}/N)$, with $f_\alpha$ defined as in Theorem 1. If $D \gg NT^2$ and $D \gg T^3$, then all the error terms involving $D$ are negligible.*

**Proof** The error term regarding $D$ can be obtained in the same way as Theorem 1, so we will let $D = \infty$ for the rest of the proof. Also, we can let $n_s = \infty$, as we observed that it contributes at most to the constant factor of the upper bound and does not change the scaling.

In the decomposition of Proposition 6, we always have

$$
\mathscr{E}_2 = O\left(S^2 \left(\log(S/r)\right)^{\alpha/(\alpha+1)} (\eta ST)^{-\alpha/(\alpha+1)}\right)
\tag{223}
$$

and

$$
\mathscr{E}_1 = O\left(S^2 (\eta ST)^{-\alpha/(\alpha+1)}\right)
\tag{224}
$$

holding regardless of $N$, so it only remains to consider $\mathscr{L}_2 + \mathscr{M}$. If $(\eta ST)^{1/(\alpha+1)} < N$, then $\mathscr{L}_2 + \mathscr{M}$ is of size $O\left(S^2 (\eta ST)^{-\alpha/(\alpha+1)}\right)$. If $(\eta ST)^{1/(\alpha+1)} \geq N$, then $N$ and $(\eta ST)^{1/(\alpha+1)}$ has same order, so $\mathscr{L}_2 + \mathscr{M} = \mathscr{L}_2 = \Theta(S^2 N^{-\alpha})$ is $O\left(S^2 (\eta ST)^{-\alpha/(\alpha+1)}\right)$. Thus in either cases we have the desired bound. $\blacksquare$

**Theorem 3.** *(Lower bound for the time scaling law) Assume the following conditions:* $n_s > N$ *and* $0 < \mathcal{R}_k(0) < S/2$. *If* $N, T \to \infty$ *while satisfying* $(8\zeta(\alpha+1)^{-1}\eta ST)^{1/(\alpha+1)} < N$, *the expected loss* $\mathbf{E}_\mathcal{D}[\mathcal{L}]$ *is*

$$\mathbf{E}_\mathcal{D}[\mathcal{L}] \geq \kappa S^2 (\eta ST)^{-\alpha/(\alpha+1)} + O\left(\eta^{-1}ST^{-1} + S^2 D^{-1/2}f_\alpha(N) + S^4\eta^2 T^2 D^{-1}\right) \quad (225)$$

*for* $\kappa$ *and constant on* $O$ *only depending on* $\alpha$, *with* $f_\alpha$ *defined as in Theorem 1. If* $D \gg NT^2$ *and* $D \gg T^3$, *then all the error terms involving* $D$ *are negligible.*

**Proof** The error term regarding $D$ can be obtained in the same way as Theorem 1, so we will let $D = \infty$ for the rest of the proof. We only show the lower bound for $\mathcal{L}_1$, holding regardless of $N$ and $n_s$. In Lemma 7 (Eq. (178)) we have

$$F(z) \geq \frac{(S-r_k)^2}{2} - \frac{8\eta S^3 T}{27}z \geq \frac{S^2}{8} - \frac{8\eta S^3 T}{27}z \quad (226)$$

for $z \geq 0$, so if $z \leq (4\eta ST)^{-1}$ then $F(z) \geq S^2/8 - 2S^2/27 > S^2/20$. The condition $\mathcal{P}_s(k) \leq (4\eta ST)^{-1}$ is equivalent to that $k \geq (4\zeta(\alpha+1)^{-1}\eta ST)^{1/(\alpha+1)}$. In evaluating $\mathcal{L}_1 = \sum_{k=1}^N \mathcal{P}_s(k)F(\mathcal{P}_s(k))$, we will only add over $k$ in range of

$$(4\zeta(\alpha+1)^{-1}\eta ST)^{1/(\alpha+1)} < k < (8\zeta(\alpha+1)^{-1}\eta ST)^{1/(\alpha+1)}. \quad (227)$$

From the assumption, this interval sits inside $1 < k < N$. For such $k$ we use upper bound of $F(\mathcal{P}_s(k)) > S^2/20$. Then by using Proposition 5 we can obtain

$$\mathcal{L}_1 \geq \frac{S^2}{20} \sum_{(4\zeta(\alpha+1)^{-1}\eta ST)^{1/(\alpha+1)}<k<(8\zeta(\alpha+1)^{-1}\eta ST)^{1/(\alpha+1)}} \mathcal{P}_s(k) \quad (228)$$

$$= \frac{S^2}{20}\left(\frac{(\zeta(\alpha+1)^{-1}\eta ST)^{-\alpha/(\alpha+1)}}{\alpha\zeta(\alpha+1)}(4^{-\alpha/(\alpha+1)} - 8^{-\alpha/(\alpha+1)}) + O\left((\eta ST)^{-1}\right)\right). \quad (229)$$

The possible effect of $n_s$ on the main term is to multiply both the main term by and $T$ by $(1 + n_s^{-\alpha})$, so it increases the bound. ■

The condition $(8\zeta(\alpha+1)^{-1}\eta ST)^{1/(\alpha+1)} < N$ is not absolutely necessary for lower bound. The condition $(\eta ST)^{1/(\alpha+1)} = \Theta(N)$ and $n_s \geq 2N$ would suffice and one can formulate a similar theorem, although the constant of lower bound might be much smaller if $(\eta ST)^{1/(\alpha+1)}/N$ is small.

Lastly, we provide a simpler version of those results combined and discuss the special case where the optimal compute $C = NT$, or the given engineering budget, is specified.

**Corollary 3.** *(Summary of the large data estimation) Assuming* $D \gg NT^2, T^3$ *and* $n_s \gg N^{1+\epsilon}$ *such that effects of* $n_s$ *and* $D$ *are negligible, then for* $N, T \to \infty$ *we have*

$$\mathbf{E}_\mathcal{D}[\mathcal{L}] = \Theta_{\eta,S,r}\left(\max(N^{-\alpha}, T^{-\alpha/(\alpha+1)})\right), \quad (230)$$

*where* $\Theta_{\eta,S,r}$ *denotes that the implied constant depends on* $\eta, S, \alpha$ *and* $r = \min \mathcal{R}_k(0) > 0$. *In particular, we have*

$$N^{\alpha+1} = O(T) \quad \Rightarrow \quad \mathbf{E}_\mathcal{D}[\mathcal{L}] = \Theta_{\eta,S,r}(N^{-\alpha}) \quad (231)$$

*and*

$$T = O(N^{\alpha+1}) \quad \Rightarrow \quad \mathbf{E}_\mathcal{D}[\mathcal{L}] = \Theta_{\eta,S,r}(T^{-\alpha/(\alpha+1)}). \quad (232)$$

**Proof** Apply Theorem 1 if $N^{\alpha+1} = o(T)$ and Theorem 2 and Theorem 3 if $N^{\alpha+1} \gg T$. ■

**Corollary 4.** *(The 'computationally optimal' case) Denote* $C = NT$ *and assume the conditions in Corollary 3. Then we have*

$$\mathbf{E}_\mathcal{D}[\mathcal{L}] \gg C^{-\alpha/(\alpha+2)}. \quad (233)$$

*When* $N = \Theta(C^{1/(\alpha+2)})$ *and* $T = \Theta(C^{(\alpha+1)/(\alpha+2)})$, *we achieve* $\mathbf{E}_\mathcal{D}[\mathcal{L}] = \Theta(C^{-\alpha/(\alpha+2)})$. *(Its implied constant may depend on implied constant for growth of* $N$ *and* $T$.)

**Proof** The first part follows from

$$\mathbf{E}_\mathcal{D}[\mathcal{L}] \gg \max(N^{-\alpha}, T^{-\alpha/(\alpha+1)}) \quad (234)$$

and

$$\max(N^{-\alpha}, T^{-\alpha/(\alpha+1)}) \geq (N^{-\alpha})^{1/(\alpha+2)}(T^{-\alpha/(\alpha+1)})^{(\alpha+1)/(\alpha+2)} = (NT)^{-\alpha/(\alpha+2)}. \quad (235)$$

The second part can be checked by substituting $(N, T) = (C^{1/(\alpha+2)}, C^{(\alpha+1)/(\alpha+2)})$ (or their constant multiples) to Corollary 3. ■

## J.5 Computing the constant for time scaling law

While we have found the time scaling law $\mathbf{E}[\mathcal{L}] = O(T^{-\alpha/(\alpha+1)})$ holding for $T = O(N^{\alpha+1})$, bounds in Theorem 2 and Theorem 3 were chosen rather lazily and do not depict the correct picture. We will find the constant using a more refined estimation, but we require additional assumptions on parameters. We will focus on the setting where $D$ and $n_s$ are large enough to be negligible, $\mathcal{R}_k(0) = r$ is fixed, and $T = O(N^{\alpha+1})$ with fixed constant such that time scaling law holds.

**Theorem 4.** *(Constant for time scaling law) Denote $\mathcal{L}^\infty$ as the loss when $D, n_s \to \infty$ so that their effect is negligible:*

$$\mathcal{L}^\infty = \mathcal{L}^\infty(T, N) = \sum_{k=1}^{N} \mathcal{P}_s(k) \frac{S^2}{2 \left(1 + \left(\frac{S}{r} - 1\right)^{-1} e^{2\eta \mathcal{P}_s(k) S T}\right)^2} + \frac{S^2 N^{-\alpha}}{2\alpha\zeta(\alpha+1)}. \tag{236}$$

*When $T, N \to \infty$ and $\lim N/(\eta S T)^{1/(\alpha+1)} = \lambda$ for a fixed constant $\lambda \in (0, \infty]$, the following limit exists:*

$$\mathcal{A}(\lambda) = \lim_{T, N \to \infty} (\eta S T)^{\alpha/(\alpha+1)} \mathcal{L}^\infty(T, N). \tag{237}$$

*The prefactor constant $\mathcal{A}$ as the a function of $\lambda$ (when $\lambda = \infty$ then let $\lambda^{-\alpha} = \lambda^{-(\alpha+1)} = 0$) is*

$$\mathcal{A}(\lambda) = \frac{\zeta(\alpha+1)^{-1/(\alpha+1)}}{\alpha+1} \int_{\lambda^{-(\alpha+1)}/\zeta(\alpha+1)}^{\infty} u^{-1/(\alpha+1)} \Phi_{S,r}(u) du + \frac{S^2}{2\alpha\zeta(\alpha+1)} \lambda^{-\alpha}, \tag{238}$$

*where*

$$\Phi_{S,r}(u) = \frac{S^2}{2 \left(1 + \left(\frac{S}{r} - 1\right)^{-1} e^{2u}\right)^2}. \tag{239}$$

**Proof** We first observe

$$\mathcal{L}^\infty = \sum_{k=1}^{N} \mathcal{P}_s(k) \Phi_{S,r}(\eta S T \mathcal{P}_s(k)) + \frac{S^2 N^{-\alpha}}{\alpha\zeta(\alpha+1)}. \tag{240}$$

We will seek to convert it into Riemann sum form of certain integral. We start by noting that

$$\mathcal{P}_s(k) = (\mathcal{P}_s(k) - \mathcal{P}_s(k+1)) \frac{k}{\alpha+1} (1 + O(k^{-1})) \tag{241}$$

$$= \frac{\zeta(\alpha+1)^{-1/(\alpha+1)}}{\alpha+1} (\mathcal{P}_s(k) - \mathcal{P}_s(k+1)) \mathcal{P}_s(k)^{-1/(\alpha+1)} (1 + O(k^{-1})) \tag{242}$$

Denote $u_k = \eta S T \mathcal{P}_s(k)$, then the sum can be approximated to

$$\sum_k \mathcal{P}_s(k) \Phi_{S,r}(\eta S T \mathcal{P}_s(k)) \tag{243}$$

$$\approx \sum_k (\mathcal{P}_s(k) - \mathcal{P}_s(k+1)) \mathcal{P}_s(k)^{-1/(\alpha+1)} \Phi_{S,r}(\eta S T \mathcal{P}_s(k)) \tag{244}$$

$$= (\eta S T)^{-\alpha/(\alpha+1)} \sum_k (u_k - u_{k+1}) u_k^{-1/(\alpha+1)} \Phi_{S,r}(u_k) \tag{245}$$

if we ignore small $k$. As $\Phi_{S,r}$ is decreasing, this corresponds to Riemann sum taking minimum in the interval $[u_{k+1}, u_k]$. So integral provides an upper bound for this sum. Similarly, we can approximate it with Riemann sum taking maximum in $[u_k, u_{k-1}]$ if we use

$$\mathcal{P}_s(k) = \frac{\zeta(\alpha+1)^{-1/(\alpha+1)}}{\alpha+1} (\mathcal{P}_s(k-1) - \mathcal{P}_s(k)) \mathcal{P}_s(k-1)^{-1/(\alpha+1)} (1 + O(k^{-1})) \tag{246}$$

instead. As $\Phi_{S,r}$ shows exponential decay, we can ignore values at small $k$, so this shows

$$(\eta S T)^{-\alpha/(\alpha+1)} \sum_k (u_k - u_{k+1}) u_k^{-1/(\alpha+1)} \Phi_{S,r}(u_k) \approx \int_{u_N}^{\infty} u^{-1/(\alpha+1)} \Phi_{S,r}(u) du \tag{247}$$

and from that
$$u_N = \eta ST N^{-(\alpha+1)}\zeta(\alpha+1)^{-1} = \lambda^{-(\alpha+1)}\zeta(\alpha+1)^{-1} \tag{248}$$
we obtain our desired result. ∎

Theorem 4 basically tells that for $N = \lambda(\eta ST)^{1/(\alpha+1)}$ and $D, n_s$ large enough, we have
$$\mathcal{L} \sim \mathcal{A}(\lambda)(\eta ST)^{-\alpha/(\alpha+1)} \tag{249}$$
with $\mathcal{A}(\lambda)$ given as Eq. (238), thus specifying the constant for time scaling law. For finite $\lambda$, this theorem covers the computationally optimal case of $(N, T) = (\lambda_1 C^{1/(\alpha+2)}, \lambda_2 C^{(\alpha+1)/(\alpha+2)})$ for some nonzero constant $\lambda_1, \lambda_2$. For $\lambda = \infty$, it describes the case $T = o(N^{\alpha+1})$ where effect of $N$ is negligible.

**Corollary 5.** *Denote $\mathcal{L}^\infty$ as $\mathcal{L}^\infty$ as the loss when $D, n_s \to \infty$ same as Eq. (236). Denote $C = NT$ and suppose that*
$$(N, \eta ST) = (\lambda(\eta SC)^{1/(\alpha+2)}, \lambda^{-1}(\eta SC)^{(\alpha+1)/(\alpha+2)}) \tag{250}$$
*for a fixed constant $0 < \lambda < \infty$. Then as $C \to \infty$, we have*
$$\mathcal{L}^\infty = \mathcal{A}\left(\lambda^{(\alpha+2)/(\alpha+1)}\right)\lambda^{\alpha/(\alpha+1)}(\eta SC)^{-\alpha/(\alpha+2)}(1 + o(1)) \tag{251}$$
*where $\mathcal{A}$ is given as Eq. (238) of Theorem 4.*

**Proof** As $\lim N/(\eta ST)^{1/(\alpha+1)} = \lambda^{(\alpha+2)/(\alpha+1)}$ under above conditions, we can apply Theorem 4 and substituting Eq. (250) into Eq. (249) gives the desired result. ∎

Technically we can optimize $\mathcal{L}^\infty$ for a given fixed value of $C = NT$ by letting $\lambda$ as argument of minimum of $\mathcal{A}\left(\lambda^{(\alpha+2)/(\alpha+1)}\right)\lambda^{-\alpha/(\alpha+1)}$, although it seems almost impossible to obtain any form of formula for such $\lambda$.

Lastly, we provide the following estimate for the time scale constant $(\mathcal{A}(\lambda))$ when $r$ is small, especially the first term in Eq. (238).

**Proposition 7.** *As $r \to 0$, we have ($\Lambda > 0$ fixed)*
$$\int_\Lambda^\infty u^{-1/(\alpha+1)}\Phi_{S,r}(u)du \approx \left(\log\frac{S-r}{r}\right)^{\alpha/(\alpha+1)}\frac{2^{1/(\alpha+1)}S^2(\alpha+1)}{4\alpha}. \tag{252}$$

**Proof** Denote $M = (\frac{S}{r} - 1)$, and replace $u$ by $(\log M)v$. Then we have
$$\int_\Lambda^\infty u^{-1/(\alpha+1)}\Phi_{S,r}(u)du = (\log M)^{\alpha/(\alpha+1)}\frac{S^2}{2}\int_{\Lambda/\log M}^\infty \frac{v^{-1/(\alpha+1)}dv}{(1+M^{2v-1})^2} \tag{253}$$
$$= (\log M)^{\alpha/(\alpha+1)}\frac{S^2}{2}\int_0^\infty 1_{v \geq \Lambda/\log M}\frac{v^{-1/(\alpha+1)}dv}{(1+M^{2v-1})^2}. \tag{254}$$

As $M \to \infty$, the integrand converges to
$$\lim_{M\to\infty} 1_{v \geq \Lambda/\log M}\frac{v^{-1/(\alpha+1)}dv}{(1+M^{2v-1})^2} = \begin{cases} v^{-1/(\alpha+1)} & \text{if } v \leq 1/2 \\ 0 & \text{if } v > 1/2. \end{cases} \tag{255}$$

The integrand is bounded by $v^{-1/(\alpha+1)}$ if $v \leq 1/2$ and $v^{-1/(\alpha+1)}e^{-2(2v-1)}$ if $v > 1/2$, those of which are all integrable. So we can apply Lebesgue's dominated convergence theorem to show
$$\lim_{M\to\infty}\int_{\Lambda/\log M}^\infty \frac{v^{-1/(\alpha+1)}dv}{(1+M^{2v-1})^2} = \int_0^\infty \left(\lim_{M\to\infty} 1_{v \geq \Lambda/\log M}\frac{v^{-1/(\alpha+1)}dv}{(1+M^{2v-1})^2}\right) \tag{256}$$
$$= \int_0^{1/2} v^{-1/(\alpha+1)}dv. \tag{257}$$

Thus we have
$$\lim_{r\to 0}\left(\log\frac{S-r}{r}\right)^{-\alpha/(\alpha+1)}\int_\Lambda^\infty u^{-1/(\alpha+1)}\Phi_{S,r}(u)du = \frac{S^2}{2}\int_0^{1/2} v^{-1/(\alpha+1)}dv \tag{258}$$
$$= \frac{2^{1/(\alpha+1)}S^2(\alpha+1)}{4\alpha} \tag{259}$$

which can be observed to be equivalent to the desired expression of Eq. (252). ∎

## J.6 Estimates for large $T$ and threshold between data/parameter scaling

The estimates for small $D$ require different techniques from estimates for large $D$. We will consider the situation $T$ grows much faster than $D$ and $N$, and discuss when data scaling law of $\mathcal{L} = \Theta(D^{-\alpha/(\alpha+1)})$ happens. We will consider a simpler setting of '$n_s = \infty$' or equivalently that effects of $n_s$ are negligible ($n_s = \omega(N)$ seems to suffice) and $\mathcal{R}_k(0) = r < S$ is fixed, although it won't be impossible to discuss their subtle effects.

First we single out effect of $T$ by comparing $\mathcal{L}(T)$ and $\mathcal{L}(\infty)$. We remind

$$\mathcal{L}_k(T) = \frac{S^2}{2\left(1 + \left(\frac{S}{r} - 1\right)^{-1} e^{2\eta d_k ST/D}\right)^2} \tag{27}$$

and its limit when $T \to \infty$ is given as

$$\mathcal{L}_k(\infty) = \lim_{T \to \infty} \mathcal{L}_k(T) = \begin{cases} \frac{(S-r)^2}{2} & \text{if } d_k = 0 \\ 0 & \text{if } d_k > 0. \end{cases} \tag{260}$$

**Proposition 8.** *Suppose that $\mathcal{R}_k(0) = r < S$ is fixed. For large $T$, we have*

$$\mathbf{E}_{\mathcal{D}}[\mathcal{L}(T)] - \mathbf{E}_{\mathcal{D}}[\mathcal{L}(\infty)] = O\left(S^4 r^{-2} D e^{-4\eta ST/D}\right). \tag{261}$$

**Proof** As $\mathcal{L}_k(T)$ is decreasing in $T$, we always have $\mathcal{L}_k(T) \geq \mathcal{L}_k(\infty)$ so therefore

$$\mathbf{E}_{\mathcal{D}}[\mathcal{L}(T)] - \mathbf{E}_{\mathcal{D}}[\mathcal{L}(\infty)] \geq 0. \tag{262}$$

So we only need to establish an upper bound for $\mathcal{L}_k(T) - \mathcal{L}_k(\infty)$. We note that $\mathcal{L}_k(T) - \mathcal{L}_k(\infty)$ when $d_k = 0$, so one can write

$$\mathcal{L}_k(T) - \mathcal{L}_k(\infty) = 1_{d_k > 0} \mathcal{L}_k(T) \tag{263}$$

where $1_{d_k > 0}$ denotes the characteristic function

$$1_{d_k > 0} = \begin{cases} 1 & \text{if } d_k > 0 \\ 0 & \text{if } d_k = 0. \end{cases} \tag{264}$$

We use simple bound of

$$\mathcal{L}_k(T) < \frac{S^2}{2\left(\left(\frac{S}{r} - 1\right)^{-1} e^{2\eta d_k ST/D}\right)^2} < \frac{S^4}{2} r^{-2} e^{-4\eta d_k ST/D}. \tag{265}$$

As $d_k$ follows binomial distribution $B(D, \mathcal{P}_s(k))$, considering its moment generating function gives

$$\mathbf{E}_{d_k}[e^{-4\eta d_k ST/D}] = \left(1 - \mathcal{P}_s(k) + \mathcal{P}_s(k) e^{-4\eta ST/D}\right)^D \tag{266}$$

so thus

$$\mathbf{E}_{d_k}[1_{d_k > 0} e^{-4\eta d_k ST/D}] = \left(1 - \mathcal{P}_s(k) + \mathcal{P}_s(k) e^{-4\eta ST/D}\right)^D - (1 - \mathcal{P}_s(k))^D. \tag{267}$$

Meanwhile, for $0 \leq u, v \leq 1$ real numbers, we have

$$|u^D - v^D| = |u - v||u^{D-1} + u^{D-2}v + \cdots + v^{D-1}| \leq D|u - v| \tag{268}$$

so, applying this inequality to above gives

$$\mathbf{E}_{d_k}[1_{d_k > 0} e^{-4\eta d_k ST/D}] \leq D\mathcal{P}_s(k) e^{-4\eta ST/D}. \tag{269}$$

Thus, we can deduce

$$\mathbf{E}_{d_k}[\mathcal{L}_k(T)] - \mathbf{E}_{d_k}[\mathcal{L}_k(\infty)] = \mathbf{E}_{d_k}[1_{d_k > 0} \mathcal{L}_k(T)] \tag{270}$$

$$< \frac{S^4 r^{-2}}{2} \mathbf{E}_{d_k}[1_{d_k > 0} e^{-4\eta d_k ST/D}] \tag{271}$$

$$\leq \frac{S^4 r^{-2}}{2} D e^{-4\eta ST/D} \mathcal{P}_s(k) \tag{272}$$

and thus

$$0 \le \mathbf{E}_{\mathcal{D}}[\mathcal{L}(T)] - \mathbf{E}_{\mathcal{D}}[\mathcal{L}(\infty)] < \frac{S^4 r^{-2}}{2} D e^{-4\eta ST/D} \sum_{k=1}^{\infty} \mathcal{P}_s(k)^2 = O\left(S^4 r^{-2} D e^{-4\eta ST/D}\right). \tag{273}$$

■

This provides an almost complete account for the effect of very large $T$. We will let $T = \infty$ from this point. We have

$$\mathbf{E}_{\mathcal{D}}[\mathcal{L}(\infty)] = \frac{(S-r)^2}{2} \sum_{k=1}^{N} \mathcal{P}_s(k)(1 - \mathcal{P}_s(k))^D + \frac{S^2}{2} \sum_{k=N+1}^{\infty} \mathcal{P}_s(k). \tag{274}$$

Applying Lemma 6 gives

$$\sum_{k=N+1}^{\infty} \mathcal{P}_s(k) = \frac{N^{-\alpha}}{\alpha\zeta(\alpha+1)} + O(N^{-\alpha-1}) \tag{275}$$

so it suffices to focus on the first sum. We will divide the range of $k$ into two $1 \le k \le M$ and $M < k \le N$. For the sum over $1 \le k \le M$, we will apply the following simple bound (in the last part, we used $1 - x \le e^{-x}$)

$$0 \le \sum_{k=1}^{M} \mathcal{P}_s(k)(1 - \mathcal{P}_s(k))^D \le (1 - \mathcal{P}_s(M))^D \le e^{-\mathcal{P}_s(M)D}. \tag{276}$$

For the sum over $M < k \le N$, we will approximate the sum into some integral, which happens to be incomplete gamma function.

**Proposition 9.** *For $2 < M < N$ integers, we have*

$$\sum_{k=M+1}^{N} \mathcal{P}_s(k)(1 - \mathcal{P}_s(k))^D \tag{277}$$

$$= D^{-\alpha/(\alpha+1)} \frac{\zeta(\alpha+1)^{-1/(\alpha+1)}}{\alpha+1} \left( \Gamma\left(\frac{\alpha}{\alpha+1}, D\mathcal{P}_s(N)\right) - \Gamma\left(\frac{\alpha}{\alpha+1}, D\mathcal{P}_s(M)\right) \right) \tag{278}$$

$$+ O\left( D^{-(2\alpha+1)/(\alpha+1)} + D^{-\alpha/(\alpha+1)} M^{-1} \right). \tag{279}$$

*Here $\Gamma$ denotes the incomplete gamma function*

$$\Gamma(s, x) = \int_x^{\infty} y^{s-1} e^{-y} dy. \tag{280}$$

**Proof** Consider the interval $[\mathcal{P}_s(N), \mathcal{P}_s(M)]$ and its partition $\mathscr{P} = \{\mathcal{P}_s(N) < \mathcal{P}_s(N-1) < \cdots < \mathcal{P}_s(M)\}$. For a function $f(x) = x^{-1/(\alpha+1)}(1-x)^D$, we will consider its upper and lower Darboux sums with respect to $\mathcal{P}$. As $f$ is decreasing in $(0, 1]$, its upper and lower Darboux sums are given respectively as

$$U(f, \mathscr{P}) = \sum_{k=M}^{N-1} (\mathcal{P}_s(k) - \mathcal{P}_s(k+1)) \mathcal{P}_s(k+1)^{-1/(\alpha+1)} (1 - \mathcal{P}_s(k+1))^D \tag{281}$$

$$L(f, \mathscr{P}) = \sum_{k=M}^{N-1} (\mathcal{P}_s(k) - \mathcal{P}_s(k+1)) \mathcal{P}_s(k)^{-1/(\alpha+1)} (1 - \mathcal{P}_s(k))^D. \tag{282}$$

and those give bound of the integral of $f$ as

$$L(f, \mathscr{P}) \le \int_{\mathcal{P}_s(N)}^{\mathcal{P}_s(M)} f(x) dx \le U(f, \mathscr{P}). \tag{283}$$

Meanwhile, by noting that

$$\mathcal{P}_s(k) = \frac{\zeta(\alpha+1)^{-1/(\alpha+1)}}{\alpha+1} (\mathcal{P}_s(k) - \mathcal{P}_s(k+1)) \mathcal{P}_s(k)^{-1/(\alpha+1)} (1 + O(k^{-1})) \tag{284}$$

one can show

$$\sum_{k=M}^{N} \mathcal{P}_s(k)(1 - \mathcal{P}_s(k))^D \tag{285}$$

$$=\frac{\zeta(\alpha+1)^{-1/(\alpha+1)}}{\alpha+1}\left(\sum_{k=M}^{N-1}(\mathcal{P}_s(k)-\mathcal{P}_s(k+1))\mathcal{P}_s(k)^{-1/(\alpha+1)}(1-\mathcal{P}_s(k))^D\right)(1+O(M^{-1})) \tag{286}$$

$$=\frac{\zeta(\alpha+1)^{-1/(\alpha+1)}}{\alpha+1}L(f,\mathscr{P})(1+O(M^{-1})). \tag{287}$$

Applying a similar argument for upper Darboux sum gives

$$\sum_{k=M}^{N}\mathcal{P}_s(k)(1-\mathcal{P}_s(k))^D = \frac{\zeta(\alpha+1)^{-1/(\alpha+1)}}{\alpha+1}U(f,\mathscr{P})(1+O(M^{-1})) \tag{288}$$

and from Eq. (283) it follows

$$\sum_{k=M}^{N}\mathcal{P}_s(k)(1-\mathcal{P}_s(k))^D = \frac{\zeta(\alpha+1)^{-1/(\alpha+1)}}{\alpha+1}\left(\int_{\mathcal{P}_s(N)}^{\mathcal{P}_s(M)}x^{-1/(\alpha+1)}(1-x)^D dx\right)(1+O(M^{-1})). \tag{289}$$

From now we will estimate the integral

$$\int_{\mathcal{P}_s(N)}^{\mathcal{P}_s(M)}x^{-1/(\alpha+1)}(1-x)^D dx. \tag{290}$$

We replace $x = y/D$ in the integral inside, then it becomes

$$\int_{\mathcal{P}_s(N)}^{\mathcal{P}_s(M)}x^{-1/(\alpha+1)}(1-x)^D dx = D^{-\alpha/(\alpha+1)}\int_{D\mathcal{P}_s(N)}^{D\mathcal{P}_s(M)}y^{-1/(\alpha+1)}\left(1-\frac{y}{D}\right)^D dy. \tag{291}$$

We want to approximate $\left(1-\frac{y}{D}\right)^D$ by $e^{-y}$, so we will estimate difference between them. We have

$$D\log(1-y/D) = -y - \sum_{k=2}^{\infty}\frac{y^k}{kD^{k-1}} \tag{292}$$

so if $D > 2y$ then

$$-y > D\log(1-y/D) = -y - \frac{1}{D}\sum_{k=2}^{\infty}\frac{y^k}{kD^{k-2}} > -y - \frac{1}{D}\sum_{k=2}^{\infty}\frac{y^k}{2(2y)^{k-2}} = -y - \frac{y^2}{D} \tag{293}$$

so

$$e^{-y}\left(1-\frac{y^2}{D}\right) < e^{-y}e^{-y^2/D} < \left(1-\frac{y}{D}\right)^D < e^{-y}, \tag{294}$$

where we used the inequality $1 - x \le e^{-x}$. As $\mathcal{P}_s(M) < 1/2$ if $M > 2$ (obvious from $\mathcal{P}_s(M) < (\mathcal{P}_s(1) + \mathcal{P}_s(2))/2 < 1/2$), any $y$ in the interval $[D\mathcal{P}_s(N), D\mathcal{P}_s(M)]$ satisfies $D > 2y$. So, we can apply this approximation in every $y$. It follows that

$$\int_{D\mathcal{P}_s(N)}^{D\mathcal{P}_s(M)}y^{-1/(\alpha+1)}\left(1-\frac{y}{D}\right)^D dy \tag{295}$$

$$=\int_{D\mathcal{P}_s(N)}^{D\mathcal{P}_s(M)}y^{-1/(\alpha+1)}e^{-y}dy + O\left(\int_{D\mathcal{P}_s(N)}^{D\mathcal{P}_s(M)}y^{-1/(\alpha+1)}e^{-y}\frac{y^2}{D}dy\right) \tag{296}$$

$$=\int_{D\mathcal{P}_s(N)}^{D\mathcal{P}_s(M)}y^{-1/(\alpha+1)}e^{-y}dy + O\left(D^{-1}\int_0^{\infty}y^{-1/(\alpha+1)}e^{-y}y^2 dy\right) \tag{297}$$

$$=\Gamma\left(\frac{\alpha}{\alpha+1},D\mathcal{P}_s(N)\right) - \Gamma\left(\frac{\alpha}{\alpha+1},D\mathcal{P}_s(M)\right) + O(D^{-1}). \tag{298}$$

Combining this with Eq. (289) and Eq. (291) gives the desired result. ∎

We combine Proposition 8 and Proposition 9 together to obtain this final estimation result.

**Theorem 5.** *(Scaling laws for large time estimation) Suppose that $N, D \to \infty$ and $n_s \gg N^{1+\epsilon}$ for some $\epsilon > 0$ so that effect of $n_s$ is negligible. Suppose that $\mathcal{R}_k(0) = r$ for all $1 \le k \le N$.*

1. *(Parameter scaling law) If $N = o(D^{1/(\alpha+1)})$, then we have*

$$\mathbf{E}_{\mathcal{D}}[\mathcal{L}] = \frac{S^2}{2\alpha\zeta(\alpha+1)} N^{-\alpha} + O\left(S^2 D^{-\alpha/(\alpha+1)} + S^2 N^{-\alpha-1} + S^4 r^{-2} D e^{-4\eta ST/D}\right). \tag{299}$$

2. *(Data scaling law) If $D = O(N^{\alpha+1})$ and $\mu = \lim(D/N^{\alpha+1})$ exists (it can be zero), then*

$$\mathbf{E}_{\mathcal{D}}[\mathcal{L}] = D^{-\alpha/(\alpha+1)} \left(\frac{(S-r)^2\zeta(\alpha+1)^{-1/(\alpha+1)}}{2(\alpha+1)}\Gamma\left(\frac{\alpha}{\alpha+1}, \frac{D}{N^{\alpha+1}\zeta(\alpha+1)}\right) + \frac{S^2(D/N^{\alpha+1})^{\alpha/(\alpha+1)}}{2\alpha\zeta(\alpha+1)}\right)$$
$$+ O\left(S^2 D^{-(2\alpha+1)/(2\alpha+2)} + S^4 r^{-2} D e^{-4\eta ST/D}\right) \tag{300}$$

*Here $\Gamma$ denotes the incomplete gamma function*

$$\Gamma(s, x) = \int_x^\infty y^{s-1} e^{-y} dy. \tag{301}$$

*In particular, if $D = o(N^{\alpha+1})$ such that $\mu = 0$, we have*

$$\mathbf{E}_{\mathcal{D}}[\mathcal{L}] = D^{-\alpha/(\alpha+1)} \frac{(S-r)^2\zeta(\alpha+1)^{-1/(\alpha+1)}}{2(\alpha+1)}\Gamma\left(\frac{\alpha}{\alpha+1}\right)(1 + o(1))$$
$$+ O\left(S^4 r^{-2} D e^{-4\eta ST/D}\right). \tag{302}$$

*In either case, $T \gg D(\log D)^{1+\epsilon}$ for some $\epsilon > 0$ implies that error terms involving $T$ are negligible.*

**Proof** Proposition 8 states

$$\mathbf{E}_{\mathcal{D}}[\mathcal{L}(T)] - \mathbf{E}_{\mathcal{D}}[\mathcal{L}(\infty)] = O\left(S^4 r^{-2} D e^{-4\eta ST/D}\right) \tag{261}$$

and we showed

$$\mathbf{E}_{\mathcal{D}}[\mathcal{L}(\infty)] = \frac{(S-r)^2}{2}\sum_{k=1}^N \mathcal{P}_s(k)(1-\mathcal{P}_s(k))^D + \frac{S^2}{2}\sum_{k=N+1}^\infty \mathcal{P}_s(k) \tag{274}$$

and

$$\sum_{k=N+1}^\infty \mathcal{P}_s(k) = \frac{N^{-\alpha}}{\alpha\zeta(\alpha+1)} + O(N^{-\alpha-1}). \tag{275}$$

For the sum of $\mathcal{P}_s(k)(1-\mathcal{P}_s(k))^D$ over $1 \le k \le N$, we use the estimate (see Eq. (276)) of

$$\sum_{k=1}^M \mathcal{P}_s(k)(1-\mathcal{P}_s(k))^D = O\left(e^{-\mathcal{P}_s(M)D}\right) \tag{303}$$

and the estimate of Proposition 9. Combining all those gives

$$\mathbf{E}_{\mathcal{D}}[\mathcal{L}] \tag{304}$$

$$= \frac{S^2 N^{-\alpha}}{2\alpha\zeta(\alpha+1)} \tag{305}$$

$$+ D^{-\alpha/(\alpha+1)} \frac{(S-r)^2\zeta(\alpha+1)^{-1/(\alpha+1)}}{2(\alpha+1)}\left(\Gamma\left(\frac{\alpha}{\alpha+1}, D\mathcal{P}_s(N)\right) - \Gamma\left(\frac{\alpha}{\alpha+1}, D\mathcal{P}_s(M)\right)\right) \tag{306}$$

$$+ O\left(S^2(D^{-(2\alpha+1)/(\alpha+1)} + D^{-\alpha/(\alpha+1)}M^{-1} + N^{-\alpha-1} + e^{-\mathcal{P}_s(M)D}) + S^4 r^{-2} e^{-4\eta ST/D}\right). \tag{307}$$

We will prove our main statement by choosing appropriate $M$ depending on size comparison between $D$ and $N$.

1. If $N = o(D^{1/(\alpha+1)})$, then we let $M = 3$, and also regard all incomplete gamma function values as $O(1)$. Then it follows

$$\mathbf{E}_\mathcal{D}[\mathcal{L}] = \frac{S^2 N^{-\alpha}}{2\alpha\zeta(\alpha+1)} + O\left(S^2 D^{-\alpha/(\alpha+1)} + S^2 N^{-\alpha-1} + S^4 r^{-2} e^{-4\eta ST/D}\right) \quad (308)$$

and thus obtaining the parameter scaling law.

2. Suppose $D = O(N^{\alpha+1})$ and $\mu = \lim(D/N^{\alpha+1})$ exists. We want

$$D^{-\alpha/(\alpha+1)}\frac{(S-r)^2\zeta(\alpha+1)^{-1/(\alpha+1)}}{2(\alpha+1)}\Gamma\left(\frac{\alpha}{\alpha+1}, D\mathcal{P}_s(N)\right) + \frac{S^2 N^{-\alpha}}{2\alpha\zeta(\alpha+1)} \quad (309)$$

to be our main term, and set $M < N$ such that the term

$$S^2 D^{-\alpha/(\alpha+1)}\Gamma\left(\frac{\alpha}{\alpha+1}, D\mathcal{P}_s(M)\right) \quad (310)$$

and error terms not depending on $T$ given as

$$O\left(S^2(D^{-(2\alpha+1)/(\alpha+1)} + D^{-\alpha/(\alpha+1)}M^{-1} + N^{-\alpha-1} + e^{-\mathcal{P}_s(M)D})\right) \quad (311)$$

are all bounded by $O(D^{-(2\alpha+1)/(2\alpha+2)})$. Set $M = D^{1/(2\alpha+2)}$. Then $\mathcal{P}_s(M) = D^{-1/2}/\zeta(\alpha+1)$, so applying the asymptotic $\Gamma(s,x) = O(x^{s-1}e^{-x})$ gives

$$\Gamma\left(\frac{\alpha}{\alpha+1}, D\mathcal{P}_s(M)\right) = O\left(D^{-1/2(\alpha+1)}e^{-\sqrt{D}/\zeta(\alpha+1)}\right). \quad (312)$$

This term and $e^{-\mathcal{P}_s(M)D} = e^{-\sqrt{D}/\zeta(\alpha+1)}$ are less than $D^{-\alpha/(\alpha+1)}M^{-1} = O(D^{-(2\alpha+1)/(2\alpha+2)})$, and obviously $D^{-(2\alpha+1)/(\alpha+1)}$ is less than $D^{-(2\alpha+1)/(2\alpha+2)}$. Thus it follows that

$$\mathbf{E}_\mathcal{D}[\mathcal{L}] = D^{-\alpha/(\alpha+1)}\frac{(S-r)^2\zeta(\alpha+1)^{-1/(\alpha+1)}}{2(\alpha+1)}\Gamma\left(\frac{\alpha}{\alpha+1}, \frac{D}{N^{\alpha+1}\zeta(\alpha+1)}\right) + \frac{S^2 N^{-\alpha}}{2\alpha\zeta(\alpha+1)}$$
$$+ O\left(S^2 D^{-(2\alpha+1)/(2\alpha+2)} + S^4 r^{-2} D e^{-4\eta ST/D}\right). \quad (313)$$

Regarding the final statement regarding sufficient condition for large $T$, $T \gg D(\log D)^{1+\epsilon}$ implies

$$De^{-4\eta ST/D} < De^{-4\eta S(\log D)^{1+\epsilon}} < D \cdot D^{-4\eta S(\log D)^\epsilon} \ll D^{-K} \quad (314)$$

for any $K > 0$, showing that the error term $O\left(S^4 r^{-2} D e^{-4\eta ST/D}\right)$ is negligible compared to all other error terms of Eq. (299) and Eq. (300). $\blacksquare$

We also provide a summary of all large time estimation results.

**Corollary 6.** *(Summary of large time estimation) Assuming $T \gg D(\log D)^{1+\epsilon}$ and $n_s \gg N^{1+\epsilon}$ such that effects of $n_s$ and $T$ are negligible, and $\mathcal{R}_k(0) = r$ for all $1 \le k \le N$. Then for $D, N \to \infty$, we have*

$$\mathbf{E}_\mathcal{D}[\mathcal{L}] = \Theta_{\eta,S,r}\left(\max(N^{-\alpha}, D^{-\alpha/(\alpha+1)})\right), \quad (315)$$

*where $\Theta_{\eta,S,r}$ denotes that the implied constant depends on $\eta, S, r$ and $\alpha$. In particular, we have*

$$N^{\alpha+1} = O(D) \quad \Rightarrow \quad \mathbf{E}_\mathcal{D}[\mathcal{L}] = \Theta_{\eta,S,r}(N^{-\alpha}) \quad (316)$$

*and*

$$D = O(N^{\alpha+1}) \quad \Rightarrow \quad \mathbf{E}_\mathcal{D}[\mathcal{L}] = \Theta_{\eta,S,r}(D^{-\alpha/(\alpha+1)}). \quad (317)$$

**Proof** Just summarize the results of Theorem 5. $\blacksquare$

## K    Methods

In this section, we present the methods used in our experiments.

### K.1    2-layer MLP

We trained a 2-layer fully connected neural network (MLP) with ReLU activations. All parameters of the MLP were initialized with a Gaussian distribution with a standard deviation of $0.001$. The input dimension of the model was $n_s + n_b = 5 + 32$ where $n_s$ is the length of control bits (number of skills) and $n_b$ is the length of the skill bits. Each skill has $m = 3$ mutually exclusive sparse bits that are used to express the skill function. The target scale was $S = 5$. The model was trained with SGD without momentum and no weight decay (the exception is the parameter emergence experiment where Adam with learning rate $0.001$ and weight decay of $5 \times 10^{-5}$ was used to escape the local minima).[6] For the data emergence experiment, the learning rate was halved every $50,000$ step.

The skill strength $\mathcal{R}_k(T)$ (Eq. (7)) was measured using $20,000$ i.i.d samples from the $k^{th}$ skill.[7] For the time emergence, the skill strengths were measured every 50 steps, while for other experiments, they were measured after training. To mimic the infinite parameter $N \to \infty$, we used the model of width 1000 (for the hidden layer). To mimic the infinite time $T \to \infty$, we trained for $5 \times 10^5$ steps ($3 \times 10^4$ steps for time emergence) where each step had the batch size of $4000$ ($2000$ for the data emergence experiment). To mimic $D \to \infty$, we sampled new data points for every batch. The details are given in the following table.

| Name | Values |
|---|---|
| width | 1000 |
| learning rate | 0.05 |
| initialization standard deviation | 0.01 |
| activation | ReLU |
| batch size | 4000 |
| steps | 500,000 |
| target scale | 5 |
| number of skill bits | 32 |
| number of skills | 5 |

### K.2    Transformer

This section outlines the transformer architecture used in Fig. 4. Data is encoded as for the 2-layer MLP, but with one-hot positional encoding appended to the data. We use a basic decoder transformer with 1 block, an initial embedding layer with output dimension 512, and a final linear layer. For the attention mechanism, we used 4 attention heads. For non-linearity, we used ReLU. A batch size of 5000 was used with a target scale $S = 1$ and default Pytorch initialization. The model was trained with SGD with a learning rate of $5 \times 10^{-5}$, weight decay of $10^{-5}$, and momentum with $\beta = 0.9$. At every 100 steps, the skill strength $\mathcal{R}_k(T)$ (Eq. (7)) was measured using $20,000$ i.i.d samples from the $k^{th}$ skill.

### K.3    Measurement of skill strength

The skill strength $\mathcal{R}_k$ is a simple linear correlation between the learned function $f$ – function expressed by NN – and $g_k$ for $\mathcal{P}_b$ given $I = k$. We approximate the expectation over $X$ by taking the mean over $20,000$ i.i.d samples from $\mathcal{P}_b$ for the $k^{th}$ skill:

$$\mathcal{R}_k = \mathbf{E}_X[f(k, X)g_k(k, X)] \approx \frac{1}{20000} \sum_{j=1}^{20000} f(k, x^{(j)})g_k(k, x^{(j)}), \tag{318}$$

---

[6]We are free to choose any optimizer as long as it preserves the order in which the skills are learned. Additionally, the parameter emergence experiment uses infinite data; we expect the same solution for Adam and SGD.

[7]Note that except the data scaling law experiment, the training set size is infinite.

where the notation $x^{(j)}$ denotes the $j^{th}$ sample.

### K.4 Details of the scaling law experiment

For the loss of the model (solid lines) in Fig. 2, we used the analytic equation for the model (Eq. (12)) under suitable assumptions such as sufficiently large $n_s$ (Table 2). For the scaling laws (dotted lines) in Fig. 2, we used the exponents from Appendix E or Appendix J and the prefactor constants from Theorem 4 (time scaling law), Theorem 5 (data scaling law), and Theorem 1 (parameter scaling law). For the hyperparameters of the simulation, we used $n_s = 10^5$ such that $n_s$ is large compared to other resources; $S = 1$ and $\mathcal{R}_k(0) = 0.01$ such that $S - \mathcal{R}_k(0) \approx S$; and $\eta = 1$.

**Time scaling law.** The total loss as a function of $T$ for $D, N \to \infty$ (Fig. 2(a), solid) is

$$\mathcal{L} = \frac{S^2}{2} \sum_{k=1}^{n_s} \mathcal{P}_s(k) \frac{1}{\left(1 + \left(\frac{S}{\mathcal{R}_k(0)} - 1\right)^{-1} e^{2\eta \mathcal{P}_s(k) ST}\right)^2}, \tag{319}$$

which follows by taking $D \to \infty$ and $N = n_s$ on Eq. (12). The scaling law (Fig. 2(a), dotted) is

$$\mathcal{L} = \mathcal{A}_T T^{-\alpha/(\alpha+1)}, \tag{320}$$

where the exponent is derived in Appendix E.1 or Theorems 2 and 3. The prefactor constant is

$$\mathcal{A}_t = \frac{S^2}{2} \frac{\zeta(\alpha+1)^{-1/(\alpha+1)}}{(\alpha+1)(\eta S)^{\alpha/(\alpha+1)}} \int_0^\infty \frac{u^{-1/(\alpha+1)}}{\left(1 + \left(\frac{S}{r} - 1\right)^{-1} e^{2u}\right)^2} du, \tag{321}$$

which we obtained by taking $D \to \infty$ on Eq. (238).

**Data scaling law.** The total loss as a function of $D$ when $N, T \to \infty$ (Fig. 2(b), solid) is

$$\mathbf{E}_D\left[\mathcal{L}\right] = \frac{S^2}{2} \sum_{k=1}^{n_s} \left(1 - \mathcal{P}_s(k)\right)^D \mathcal{P}_s(k), \tag{322}$$

which follows from Eq. (58). The scaling law (Fig. 2(b), dotted) is

$$\mathcal{L} = \mathcal{A}_D D^{-\alpha/(\alpha+1)}, \tag{323}$$

where the exponent follows from Appendix E.2 or Theorem 5. The prefactor constant is

$$\mathcal{A}_D = \frac{S^2}{2} \frac{\zeta(\alpha+1)^{-1/(\alpha+1)}}{\alpha+1} \Gamma\left(\frac{\alpha}{\alpha+1}\right) \tag{324}$$

which we obtained by taking $N \to \infty$ in Eq. (302).

**Parameter scaling law.** The total loss as a function of $N$ when $T, D \to \infty$ (Fig. 2(c), solid) is

$$\mathcal{L} = \frac{S^2}{2} \sum_{k=N+1}^{n_s} \mathcal{P}_s(k), \tag{325}$$

which follows from taking $T, D \to \infty$ on Eq. (12). The scaling law (Fig. 2(c), dotted) is

$$\mathcal{L} = \mathcal{A}_N N^{-\alpha}, \tag{326}$$

where the exponent follows from Theorem 1. The prefactor constant is

$$\mathcal{A}_N = \frac{S^2}{2}, \tag{327}$$

which we obtained by taking $D, T \to \infty$, $N/n_s \to 0$, and $\zeta(\alpha+1) \approx \alpha^{-1}$ in Eq. (217).

**Compute scaling law.** The total loss as a function of $T$ and $N$ for $D \to \infty$ (Fig. 3, solid) is

$$\mathcal{L} = \frac{S^2}{2} \sum_{k=1}^{N} \mathcal{P}_s(k) \frac{1}{\left(1 + \left(\frac{S}{\mathcal{R}_k(0)} - 1\right)^{-1} e^{2\eta \mathcal{P}_s(k) S T}\right)^2} + \sum_{k=N+1}^{n_s} \mathcal{P}_s(k), \tag{328}$$

which follows by taking $D \to \infty$ in Eq. (12). In Fig. 3, we plotted for $N \in \{10, 20, 50, 70, 100, 200, 500, 700, 1000, 2000, 5000, 10000\}$ and $T \in [1, 1000]$ as examples of different tradeoff between $T$ and $N$ for fixed $C$.

The scaling law (Fig. 3, dotted) is

$$\mathcal{L} = \mathcal{A}_c C^{-\alpha/(\alpha+2)}, \tag{329}$$

where the exponent is derived in Appendix E.4 or Corollary 4. Using Corollary 5, the prefactor constant is

$$\mathcal{A}_c = \mathcal{A}\left(\lambda^{(\alpha+2)/(\alpha+1)}\right) \lambda^{\alpha/(\alpha+1)} (\eta S)^{-\alpha/(\alpha+2)} \tag{330}$$

where $\mathcal{A} : \mathbb{R} \to \mathbb{R}$ is defined in Eq. (238). We used the minimum value of $\mathcal{A}_c$ for $\lambda \in (0, \infty]$.

### K.5 Estimates of the compute use

On CPU, our emergence experiments on the 2-layer MLP (Fig. 1) take $2 \sim 5$ hours for a single run of time emergence experiments and $20 \sim 40$ hours for a single run of other experiments depending on the CPU. All experiments were repeated 10 times (except for parameter emergence where we repeated the experiment 50 times). Each experiment requires memory of at most 5GB. The CPU cluster in which we experimented contained the following CPUs: Intel(R) Core(TM) i5-7500, i7-9700K, i7-8700; and Intel(R) Xeon(R) Silver 4214R, Gold 5220R, Silver 4310, Gold 6226R, E5-2650 v2, E5-2660 v3, E5-2640 v4, Gold 5120, Gold 6132. The transformer experiment (Fig. 4) takes $48 \sim 72$ hours for each run; we used an RTX4090 with 24GB RAM, with 1 CPU from the list above.

