# OpenReview forum: "An exactly solvable model for emergence and scaling laws in the multitask sparse parity problem"
_NeurIPS.cc/2024/Conference — NeurIPS 2024 poster_

### Official Review · Reviewer_TV9v · 2024-07-11

**Soundness:** 4
**Presentation:** 4
**Contribution:** 3
**Rating:** 7
**Confidence:** 3

**Summary:**

The paper presents a model capable of predicting the appearance of emergent abilities, using only information on the emergence of the first ability. The model successfully predicts emergence in a 2-layer MLP solving the multitask sparse parity problem, a toy-model problem constructed with a power-law skill distribution, to mimic the theorized distribution of language datasets. The model also exhibits scaling laws known to exist in this problem.

**Strengths:**

The paper expands on recent works, in a significant and active field of research. The authors present convincing results when evaluating the proposed emergence model against MLP training data. Extensive derivations are provided in the appendix, although I did not carefully check them. The paper is written in a clear manner and results are visualized in clear formats.

**Weaknesses:**

- The scope of the experiments is too small in my opinion, specifically Figure 1. It would be nice to see results for a lot more than 5 skills, since it seems possible that predicting the emergence of less-frequent skills becomes harder at higher orders of magnitude. My impression is that the experimental setup is light enough that scaling up will not require unreasonable resources.

- The general scope of the paper is a bit small, focusing only on the multitask sparse parity problem. It would have been nice to see results on more natural settings. That said, the current results on the chosen problem setting are still interesting on their own.

- Regarding the scaling laws presented in Table 2, glossing over the paper gives the impression that these are new scaling laws found by the authors, when in fact they are reproductions of the scaling laws calculated by Michaud et. al. It might help to clarify the relation to Michaud et. al. in the main section of the paper.

**Questions:**

The main section of the paper does not explain why the input data is distributed as a power law (Zipf's law), making it hard to understand the justification for readers unfamiliar with previous literature, i.e. Michaud et. al. I would suggest adding a short explanation when introducing the problem.

**Limitations:**

Limitations are adequately discussed, including scope limitations.

---

> ### Author Rebuttal · Authors · 2024-08-05
>
> Dear reviewer, thank you for taking the time to review our paper. We appreciate your comments and suggestions.
>
>  * Scope of the experiments
>
> Additional learning of the skills, as correctly spotted, requires another order of magnitude of computation which challenges our computational budget. The predictive power of our model indeed relies on the effective-decoupling assumption (second paragraph of Section 6 and Appendix D.3) of the skills in MLP, which is weakened at larger $k$ because of the noise in SGD. Please see “Intuition and limitations of our model”  in our global rebuttal for details.
>
>
>  * Scope of the paper
>
> Indeed, we agree that we focused on a more idealized, abstract, and more feasible setup over more complicated but practical scenarios requiring significant computation budgets. We believe a more specific title correctly informs the scope and motivation of the paper. Please see “Scope of the paper and its title”  in the global rebuttal.
>
>  * Scaling laws presented in Table 2
>
> In fact, as described in the second paragraph of Section 4, we arrive at the same scaling law as Hutter 2021 for data and Michaud et al. 2023 for time, data, parameters. Table 2 was to emphasize our novel contribution: how the proposed model leads to rigorous derivation of the scaling laws (please see “Contributions beyond Michaud et al. 2023” in our global rebuttal for details). That said, we are happy to incorporate this information in the caption of Table 2 for clarification.
>
>  * Question: the power-law input data
>
> We appreciate the reviewer's suggestion to provide more context on the power-law distribution of input data. We agree that a brief explanation in the main text would benefit readers less familiar with the literature. We will update our paper to include this explanation.
>
> Regarding the justification for the power-law assumption: As the reviewer recognizes, real-world tasks often exhibit a spectrum of skill frequencies (such as Zipf's law). For instance, in language tasks, placing common words like 'the' or 'is' in a sentence occurs frequently, while correctly using specialized technical terms or rare words occurs much less often.
>
> We again thank the reviewer for helpful comments and for taking the time to review our paper.

---

> > ### Comment · Reviewer_TV9v · 2024-08-11
> >
> > I thank the authors for their detailed response. In light of the changes proposed, I will change my score to accept (7).
> >
> > One question about the experimental setup: What were the hardware requirements for the 2-layer MLP experiments in figure 1?

---

> > > ### Author Response · Authors · 2024-08-12
> > >
> > > Dear reviewer, we deeply appreciate updating the score.
> > >
> > > The specification of setup is detailed in Appendix K.5. Each run of the experiment – one point in the figure – requires 2 to 5 hours for time emergence and 20 to 50 hours for other experiments on a CPU. We ran the experiments on a CPU cluster because the small size of the MLP allows less pronounced difference in the running on GPU and CPU (running on RTX 4090 GPU was typically only 3X faster than an average CPU in the cluster) while we benefited from parallelism in a larger CPU cluster.
> > >
> > > The most demanding experiment was Fig.1(b) with data emergence. The figure has 30 different numbers of datapoints ($30$ runs), repeated $10$ times for the error bars. We ran for $5 \times 10^5$ steps for data and parameter emergence (compared to $3 \times 10^4$ for time emergence) to remove the potential effect from early stopping (i.e. to assure $T \gg D$).
> > > To observe the emergence of an additional skill, we require a magnitude increase in $D$ which leads to a magnitude increase in $T$ – to assert $T \gg D$ – and a magnitude increase in the batch size – to mitigate the SGD noise.

---

### Official Review · Reviewer_D3CL · 2024-07-13

**Soundness:** 4
**Presentation:** 4
**Contribution:** 3
**Rating:** 7
**Confidence:** 5

**Summary:**

I’ll write two summaries: one to state my “moral” understanding of the work and another to state the specific contributions

**“Moral” Summary:** The authors propose an analytically solvable model to study scaling laws and emergent abilities by combining the problem of Barak et al 2022 + the data distributional assumptions of Michaud et al. 2023/2024 + the model of Saxe et al. 2024 + their own innovations.

**Specific Summary:**

- The authors study the multi-task sparse parity problem studied by Barak & Michaud
- The authors propose a multilinear model that identifies each sparse parity problem as a “skill” and then define the learned function as a “multilinear” (i.e. two independent linear parameters) function of the “skill” basis functions
- In this model, one can then exactly compute scaling and emergent abilities as a function of the typical scaling parameters (compute, data, parameters)
- The authors also train 2 layer MLPs and transformers to test how closely their maths match empirical results

**Strengths:**

- Overall, I think this is a really well done paper (although I’m concerned about Figure 1 - see weaknesses). What it does, it does thoroughly.
- Table 1 is useful in helping explain the multi-task parity problem
- Table 2 is a great way to summarize and organize both the results as well as the conditions of each result

**Weaknesses:**

- Emergence is a phenomenon studied at scale, and while I strongly support trying to find simplified models of large-scale phenomena, I feel that there needs to be an attempt to connect back to the original phenomena of interest. For instance, Michaud et al. 2023/2024 (Citation 17 in this paper) - on which this work most closely connects - at least attempts to look into real language model pretraining data. In contrast, this paper makes no such attempt, which I think strongly limits it. This is the #1 reason I don't feel comfortable giving a higher score.
- Following the above point, I consequently think a more focused title (e.g., “An exactly solvable model for emergence and scaling laws in the multitask sparse parity problem”) would better represent this paper.
- For an exactly solvable model, Figure 1 shows that the predictions only roughly match the experimental results. This becomes especially pronounced for higher k. For instance, look at Figure 1(c) orange. The prediction is that there should be a rapid leap from 0 to 1, but instead, there is a sigmoid-like transition with long tapering tails. Green, red and purple are all similar. The inability to predict 2-layer MLPs makes me think that this analytically solvable model is already only an approximation of incredibly simple networks

**Questions:**

Figure 4: What is the timescale of each skill’s emergence for the transformer? I can’t tell if the higher k lines are step functions or sigmoidal functions that are compressed by the log scaling of the x axis. To be clear, I’m not asking for when the skills emerge, but how long it takes for each skill to emerge.

---

> ### Author Rebuttal · Authors · 2024-08-05
>
> Dear reviewer, thank you for your comments on the paper and for appreciating our tables. Your moral summary represents our motivation in the paper well.
>
>  * Realistic setup and the title of the paper
>
> We fully appreciate your comment on the scope/limitation of the paper and thank you for suggesting to change the title to specify its scope and motivation. Please see “Scope of the paper and its title” in the global rebuttal.
>
>  * Theory and experiment matching in Fig.1(c)
>
> We appreciate the reviewer's careful observation of Figure 1. The discrepancies noted, particularly for higher $k$, are indeed due to the abstraction of our model. We expect the deviation to enlarge at higher $k$ as the SGD noise becomes more comparable to the difference in the skill frequencies and the effective-decoupling assumption (second paragraph of Section 6) weakens. Please see “Intuition and limitations of our model” in the global rebuttal.
>
> We acknowledge that our model is an approximation as its purpose is to serve as a baseline for understanding more complicated dynamics in NNs. The strength of our approach lies in its ability to provide analytical predictions and an intuitive understanding of key phenomena, which would be challenging to achieve with more complex models. We believe this trade-off between simplicity and predictive power is valuable for advancing theoretical understanding in the field. Please see “Scope of the paper and its title” in the global rebuttal.
>
> We discuss the limitations of our model in Section 5.5 and share our intuition on why it closely approximates an MLP in Section 6. Regarding the specific case of Figure 1(c), the discrepancy observed in MLPs is partly due to initialization-dependent learning outcomes, which our model doesn't explicitly incorporate. As discussed in the last paragraph of Section 5.5 and in Table 5 in Appendix I, the initialization-dependent learning failures in MLPs affect $\mathcal{R}_k$, especially when MLP has only a handful of hidden neurons to learn the skills: Even when an MLP can express all skill functions, it may fail to learn due to unfavorable initialization or inefficient use of its hidden neurons. We have argued how such outliers increase the standard deviation of the overall performance metric $\mathcal{R}_k$, but the argument also explains why the mean of $\mathcal{R}_k$ fails to saturate to $S$.
>
>
>  * Question: The time scale of each skill’s emergence in transformers
>
> The timescale of the transformer is in optimization steps. Analogous to the assumption in the stage-like training (Figure 6 in Appendix D), the saturation time is typically significantly smaller compared to the emergence time (when it starts to emerge) even in linear time scale. Please see Figure 1 of the attached pdf in the global rebuttal.
>
> The saturation time (how long it takes to emerge) is approximately 500 steps for the first skill and 5,000 for the fifth skill. All saturations show a sigmoid-like saturation, but are more distorted compared to that of the 2-layer MLP. Please see Figure 2 of the attached pdf in the global rebuttal. If the reviewer has additional comments or questions, please do not hesitate to let us know.
>
>
> We again thank the reviewer for the suggestion of the title and for dedicating the time to review our work.

---

> > ### Comment · Reviewer_D3CL · 2024-08-12
> > **Response to Authors' Rebuttal**
> >
> > Thank you for your comments! I appreciate the additional figures in the global response and your answers to my questions.
> >
> > I'm going to increase my confidence but keep the score. I don't feel comfortable increasing my score because while I feel this paper is very thorough, I feel it is limited in its general applicability without strong connections back to more realistic (larger, more data, real data) models.

---

> > > ### Author Response · Authors · 2024-08-12
> > >
> > > Dear reviewer, we thank you for increasing confidence in your assessment. We are glad that additional plots have clarified your question.

---

### Official Review · Reviewer_owz9 · 2024-07-15

**Soundness:** 4
**Presentation:** 3
**Contribution:** 2
**Rating:** 6
**Confidence:** 3

**Summary:**

This paper provides an in-depth study of a toy dataset and model, both in terms of scaling laws and emergent skills. They study the ‘multitask sparse parity’ synthetic problem introduced by Michaud et al. (target is the parity function of a string of random bits, each task indicates a different subset of bit ids, task frequency follows a power law). New in this work, they consider a ‘multilinear’ model (i.e. $y=ab x$ rather than simply $y=cx$) to incorporate the dynamics of a two layer neural net (as per Saxe et al. in a different setting).

The paper recovers scaling laws (which agree with Michaud et al.’s coefficients for T, D & N, and additionally adds for C=TN). The model also gives rise to emergence of each skill with a sigmoidal shape. Rarer skills are learned later on.

**Strengths:**

- From a technical perspective, the paper is a very strong and complete piece of work. It exhaustively sweeps through results and surrounding analysis of the setup studied in the paper (even, for example, having proofs of the scaling laws in increasing resolution).

- I have confidence in the claims and analysis. I spot checked several parts in depth, however, I have not been able to interrogate all claims and analysis in the paper (the appendix pushes the paper to 54 pages).

- Providing a model that unites sigmoidal-shaped skill emergence with scaling laws, is a tantalizing prospect, as these are two major properties of LLMs. A theoretical model combining the two aspects would be of high interest to several parts of the community.

**Weaknesses:**

- The paper contains a huge amount of material. A lot of the good stuff is buried in the appendix (related work is important, the contrast with a linear $y=cx$ model, the scaling law proof sketches, the stage-like training discussion). The reading experience suffers from this, and a conference paper struggles to do it justice. It's possible it would be better suited to a long-form journal format (and would allow reviewers more time to comb through the details).

- The scaling laws derived by the paper match the coefficients found in Michaud et al. (with the addition of $C$). There is a slight difference in that the setup now uses $y=abx$, but I generally felt that given the repeated outcome, providing scaling laws derived with varying resolutions need not be such a major focus of the paper and appendix (maybe some nuance has been lost on me?). Reducing this would free up bandwidth to allow readers to absorb the other more interesting parts.

- A concern I have is in how realistic the setup of the dataset and model is. The emergence and scaling laws come from a lightweight two-parameter linear regressor which receives ideal task indicators as input. The closest interpretation in a real setting I can think of, is as a two-layer linear MLP head, placed on top of a deep pretrained model that is frozen with powerful representations for each task already learned. This paper is still valuable, but I might suggest a more scoped title because of this – ‘an exactly solvable model for emergence and scaling laws’ is not inaccurate, but might be a little broad.

**Questions:**

See weaknesses.

**Limitations:**

Fine.

---

> ### Author Rebuttal · Authors · 2024-08-05
>
> Dear reviewer, thank you for your kind comments and especially for dedicating the time to read through the appendix in depth.
>
>  * Material in the appendices
>
> We appreciate your suggestion regarding the suitability of our paper for a journal format. We fully agree that a journal-style paper would allow for a more detailed narrative and potentially provide an easier reading experience. At the same time, we believe that presenting this work at a conference like NeurIPS is valuable: scaling laws and emergence are of great current interest to the broad ML community, and the conference format enables us to share our findings with a general audience and receive valuable community feedback, potentially inspiring further research.
>
> We have strived to balance the conference format constraints with the depth of our work by including substantial appendices. These appendices serve to clarify our intuitions and make connections for readers from various backgrounds, while keeping the main paper focused and concise. We appreciate your thorough review of both the main paper and the appendices, and we thank you for recognizing the value of the additional material we provided.
>
>
>  * Focus on the scaling laws
>
> Even though we wanted to put only the formal derivations in the appendix (mainly C, E, F, and J)  and present all our intuition in the main text, some intuitive arguments had to be added to the appendix (mainly D, G, and H) for an integrated presentation of our contributions. That said, we still think there is room for further emphasizing the effective decoupled dynamics of MLPs and emergence in our work as suggested by the reviewer.
>
>  * Realistic setups
>
> We appreciate your comment regarding the connection between our model and MLPs.
> While our current paper provides a focused exploration of our theoretical model, we acknowledge that a comprehensive justification of its applicability to general setups, encompassing (both theoretical and empirical), would constitute a significant body of work meriting its own dedicated study.
>
> We, however, share our intuition on why a simple model with prebuilt powerful representation approximates the properties of an MLP in the following paragraphs: in the second paragraph of Section 6 in the main text, in the discussion of the effective-decoupling in Appendix D.3, in an example scenario of MLP in Appendix G. Please see “Intuition and limitations of our model” in the global rebuttal for details.
>
> Regarding the scope and the title, we fully appreciate your concerns and plan to change the title of the paper as suggested. Please see “Scope of the paper and its title” in the global rebuttal.
>
> We again thank the reviewer for investing the time to thoroughly read our work.

---

### Official Review · Reviewer_NXT5 · 2024-07-16

**Soundness:** 1
**Presentation:** 2
**Contribution:** 1
**Rating:** 3
**Confidence:** 3

**Summary:**

The paper proposes to use a certain generalization of the well-known sparse parity problem, as a theoretical framework for neural scaling laws.

The theoretical setup doesn't seem to make sense (I'm open to changing my mind). Indeed, equations (2) and (4) taken together give
\begin{equation*}
    f^*(i,x) = S\sum_{k=1}^{n_s} g_k(i,x) = S\sum_{k=1} \delta_{ki}g_i(i,x) = S g_i(i,x).
\end{equation*}
This is definitely not "a sum of $n_s$ skills" (as claimed by the authors); it is a single skill. The same issue repeats itself in (9) when the authors introduced their so-called multi-linear model. Indeed, that model simplifies to (again thanks to (2))
\begin{equation*}
    f_T(i,x) = \sum_{k=1}^{n_s} a_k(T)b_k(T)g_k(i,x) = \ldots = a_i(T)b_i(T)g_i(i,x).
\end{equation*}
From this point onward, it is not clear what the paper is trying to do.

Second, it is not clear what the paper is trying to achieve beyond what was already done in Michaud et al's "Quantization Hypothesis" paper (for the record, the paper proposed the multi-task sparse parity problem as a example exhibiting scaling laws w.r.t sample size and model size, within the framework of their quantization hypothesis).

Finally, the paper seems to be missing some important literature, for example Cabannes et al. "Scaling Laws for Associative Memories" (ICLR 2024), which proposes finite-capacity extension of Hutter's model, and establishes an array of different scaling laws for different learning algorithms.

**Strengths:**

As explained already, my low score is because I think the theoretical setup of the paper is unclear. I'm open to changing my mind.

**Weaknesses:**

As explained already, my low score is because I think the theoretical setup of the paper is unclear. I'm open to changing my mind.

**Questions:**

In what way do (3) and (9) represent "a sum of skills" ?

**Limitations:**

Yes (in Section 5.4)

---

> ### Author Rebuttal · Authors · 2024-08-05
>
> Dear reviewer, thank you for being open to discussing what was unclear to you in the paper. Please see our response below.
>
>  * Theoretical setup clarification
>
> Regarding the sum of skill functions, we believe the confusion arose from interpreting the input variable $i$ – which depends on the data points – as a fixed index such as $k$: note that $(i,x)$ as a pair form the input.
>
> The $k^{th}$ skill function $g_k$ and the target function $f^*$ are different functions. For example, a datapoint $(i,x)$ with $i=k+1$ and any $x$ will result in $|g_k(k+1,x)| = 0$ (Eq. (2)) while $|f^*(k+1,x)| = |Sg_{k+1}(k+1,x)| = S$. Of course, a datapoint $(i,x)$ with $i=k$ will lead to $S|g_k(k,x)| = f^*(k,x)$.
>
> The Kronecker delta in your equation appears only if $i=k$, which is true only for datapoints from the $k^{th}$ skill (the skill bits are one-hot with the $k^{th}$ entry equal to 1), but not for other datapoints. We believe the clarification will also remove the confusion regarding Eq. (3) and Eq. (9).
> If still in doubt, please look at our example below using Table 1.
>
>
>  * Contributions beyond Michaud et al.
>
> Our goal was to provide the simplest quantitative model that captures the intuition of Michaud et al. 2023, to provide a rigorous derivation of the scaling laws, and to predict the emergence of an MLP. Please see “Contribution beyond Michaud et al. 2023” in the global rebuttal for details.
>
>  * Recent literature
>
> Finally, we thank the reviewer for suggesting some related literature, which we will add to our related work section (in particular, the work: Cabannes et al. "Scaling Laws for Associative Memories" (ICLR 2024)).
>
>
>  * Table 1 Example
>
> In our Table 1, note that the first column represents $i$ for a given datapoint, the second column represents $i$ in control bits, and the third column the skill bits $x$. The $(i,x)$ as a pair form the input to the model. The fourth column, which is noted as $y$ is equivalent to the target function $f^*(i,x)$. From the sixth to the last columns are the skill functions from $g_1(i,x)$ to $g_{n_s}(i,x)$.
>
> It is indeed true that for a single datapoint (a row), $f^*(i,x)$ (column 4) is a multiple of $g_i(i,x)$ (column $i+5$). However, no single skill function $g_k$ (a single column between $6$ and $n_s + 5$) can represent $f^*$ (column 4) for **all datapoints (all rows)**. Thus, we need the target function $f^*$ to be a sum of all skill functions (scaled by $S$) in Eq.(4).
>
> In order for the relationship provided by the reviewer to hold, we must assume that the data distribution is generated by a single skill only (justifying that the Kronecker delta $\delta_{ik}$ holds for all datapoints), making the problem a sparse parity and not a `multitask’ sparse parity problem.
>
> If the reviewer finds our clarification insufficient or unclear, please do not hesitate to let us know.

---

> ### Comment · Reviewer_NXT5 · 2024-08-13
>
> - Thanks for the clarification; a notation problem indeed.
>
> - I have a good understanding of the paper now. I think the contribution is interesting but still incremental (based on current literature on provable scaling laws).
>
> I'm increasing my score to 6.

---

> > ### Author Response · Authors · 2024-08-13
> >
> > Dear reviewer,
> >
> > We are glad that we clarified the confusion and thank you for updating the score.

---

> > ### Comment · Area_Chair_bVPQ · 2024-08-13
> >
> > Dear reviewer,
> >
> > Thank you for your review and efforts. Please update your score on the original review as well.

---

### Author Rebuttal · Authors · 2024-08-05

Dear reviewers and AC, we present a global rebuttal to address the overlapping reviews.


## Scope of the paper and its title

As most reviewers have pointed out, the primary strength of our paper is the detailed theoretical analysis of the scaling laws and emergence with a concrete model while the main weakness lies in the lack of empirical work on larger models in more natural setups.

Even though we fully appreciate the importance of gaining predictive power in larger models with real-world datasets, we mainly focused on theoretical tractability over realism, and believe the work is meaningful on its own by providing 1. clear isolation of key phenomena (emergence and scaling laws) without confounding factors present in more complex systems and 2. rigorous mathematical analysis and derivation from fundamental principles for future works.

On that note, we fully appreciate the comments on how the title is less focused and potentially suggests a scope beyond our intentions. As commented, a more detailed title such as **“An exactly solvable model for emergence and scaling laws in the multitask sparse parity problem”** better represents the scope and motivation of the paper. We thank the reviewers for pointing this out.

## Contributions beyond Michaud et al. 2023

As acknowledged by some reviewers, our work can be viewed (though not limited to) as a theoretical formalization of the quanta hypothesis Michaud et al. 2023 in which we wish to emphasize our contribution in more detail.

The quanta hypothesis from Michaud et al. 2023 lacks mathematical formalism (i.e., both quanta and skills are not mathematical objects) and lacks the derivation of scaling laws under “gradient descent dynamics” (it was stated that the skills were learned by some criteria, see e.g., the Quantization Hypotheses on page 3 of Michaud et. al, https://openreview.net/pdf?id=3tbTw2ga8K).

We provide a formal model trained on gradient flow that 1. analytically reproduces the scaling laws (including the prefactor constants) and 2. predicts emergence in 2-layer MLPs. The quantitative prediction of emergence in 2-layer MLPs is novel and no modeling or prediction for the emergence phenomenon was studied in Michaud et al. 2023. We believe such formalism serves as a baseline for future more complex models for understanding scaling laws and emergence in more practical NNs.

 ## Intuition and limitations of our model

Our work presents a simplified model that approximates complicated feature learning dynamics of an MLP with a decoupled basis functions and a product of parameters. Even though our work suggests that our assumptions effectively capture the emergence in MLP (e.g. Fig.(1)), the exact conditions for the assumptions to hold and whether it holds in larger models with realistic dataset requires further study.

We, however, extensively share our intuition on why our model approximates the emergence in MLPs (Second paragraph of Section 6, Appendix D.3, Appendix G, and Appendix I) and also the limits of our model of why it may diverge from the dynamics of MLPs (Section 5.5).

For example, we give some evidence that our model well-approximates the MLP because of the effective decoupling in MLPs or that skills are feature-learned in stages (Second paragraph of Section 6 and Appendix D.3). The large difference in skill frequencies, analogous to stage-like training (Appendix D) of parameters in multilinear models, allows each $g_k$ to be learned at different time scales: justifying why the decoupled dynamics of our multilinear model is a good approximation. Unfortunately, the absolute difference in skill frequencies becomes comparable to SGD noise as $k$ increases: creating a larger divergence for larger $k$. The noise from SGD can be mitigated with a larger batch size, but we then require another magnitude in training time and batch size that challenges our limited computational budget.

---

> ### Comment · Reviewer_D3CL · 2024-08-08
> **Brief Comment on Contributions beyond Michaud et al. 2023**
>
> I'll respond to the authors tomorrow, but briefly, on this overall comment, I'd like to clarify one point concerning "Contributions beyond Michaud et al. 2023" today in case other reviewers look at this in the interim.
>
> I personally think that this work extends far beyond the Michaud et al. 2023 paper. If any other reviewers feel differently, please explain why or please point me towards where you might have already explained in your review.
>
> Thank you!

---

### Decision · Program_Chairs · 2024-09-25

**Decision:**

Accept (poster)

**Comment:**

This paper introduces an analytically solvable model to explain neural scaling laws and presents a thorough analysis of the model. All reviewers gave positive reviews and were supportive of acceptance.